# Review article: Rethinking Preparedness for Coastal Compound Flooding: Insights from a Systematic Review.

Dina Vanessa Gomez Rave<sup>1</sup>, Anna Scolobig<sup>2</sup>, Manuel del Jesus<sup>1</sup>

<sup>1</sup>HCantabria- Instituto de Hidráulica Ambiental de la Universidad de Cantabria, Santander, 390011, Spain

<sup>2</sup>Institute for Environmental Sciences, University of Geneva, Geneva, 1211, Switzerland

Correspondence to: Dina Vanessa Gomez Rave (dinavanesa.gomez@unican.es)

#### Abstract.

Tackling the growing risks of Compound Flooding (CF) requires transformative preparedness strategies, particularly in estuarine and coastal regions, where the interaction of drivers such as storm surges, rainfall, and river discharge exacerbates impacts. Despite progress, fragmented governance, weak cross-sectoral coordination, and the limited integration of scientific insights hinder effective responses.

This systematic review draws on 49 studies to explore how preparedness strategies are evolving to integrate technical, environmental, and social dimensions while evaluating the role of governance and collaboration in enhancing adaptive approaches. Hybrid Early Warning Systems combining statistical and hydrodynamic models with real-time data are critical for forecast accuracy and timely decision-making. Similarly, balanced implementation of green, blue, and gray infrastructure provides sustainable responses, with Nature-based Solutions complementing traditional engineering. Our results also show that strengthening governance and communication is essential to improve preparedness. Involving communities in land-use planning, building regulations, and communication ensures that measures are both actionable and context-specific. Incorporating psychological and behavioural data into preparedness frameworks and models helps strengthen the link between awareness and behaviours. Enhanced coordination across sectors and levels of government is also vital to addressing the systemic nature of CF risks, moving beyond siloed, single-hazard responses.

## Rethinking Preparedness for Coastal Compound Flooding: Insights from a Systematic Review

#### 25 1. Introduction

The greatest risks from a changing climate may not arise from single hazards, but from the interaction of multiple climatic drivers and/or hazards combined with diverse forms of exposure, intersectional socio-economic and geopolitical vulnerabilities, often challenging the capacity of institutions and communities to respond effectively (Simpson et al., 2023). Drivers that occur simultaneously or in close succession can intensify the hazard and expand its spatial and temporal extent, resulting in more severe and prolonged events than those associated with single drivers (AghaKouchak et al., 2020; Brett et al., 2024). Hazards alone do not lead to disasters, but when combined with factors such as vulnerability and limited response capacity, their impacts can escalate rapidly, threatening both communities and ecosystems (Eze and Siegmund, 2024). To understand how these interactions give rise to high-impact situations, it is useful to distinguish the roles played by different components along the causal chain. Risk is commonly conceptualised as the potential for adverse consequences for human or ecological systems resulting from the interaction between *hazard*, *exposure*, and *vulnerability* (IPCC, 2023). Within this framework, *compound events* are defined as the combination of climatic *drivers* and/or *hazards* that jointly contribute to

societal or environmental *risk* (Zscheischler et al., 2018). *Drivers* encompass processes, variables, and phenomena in the climate and weather domain—such as precipitation, temperature, river flow, coastal water levels, atmospheric humidity, soil moisture or wind speed—that may operate across multiple spatial and temporal scales. *Hazards*, in contrast, denote the immediate physical phenomena—such as floods, heatwaves, or landslides—that may trigger *impacts* when they coincide with *exposure*—the presence of people, infrastructure, or ecosystems in harm's way—and *vulnerability*—their propensity to suffer damage or loss due to limited capacity to anticipate, cope with, or recover from the event (Koks et al., 2015; Zscheischler et al., 2020; IPCC, 2023). The interplay of multiple drivers and/or hazards can lead to *compound events*—such as co-occurring extremes—and their intersection with exposed and vulnerable systems may result in *compound risk* (IPCC, 2023). This conceptual framing provides a basis for analysing how interacting climatic conditions can evolve into complex events—and how their consequences ripple through interconnected systems.

At a more structural level, the concepts of *systemic vulnerability* and *systemic risk* offer a complementary lens. Systemic vulnerability refers to the susceptibility of interdependent systems—such as infrastructure networks, governance structures, or social services—to suffer disruption under external stress, due to the cascading effects that arise from their internal linkages (Weir et al., 2024). A recently proposed definition of systemic vulnerability highlights the persistent core of vulnerability that endures over time despite mitigation efforts, societal and technological progress, leading to reinforced impacts (Armaş et al., 2025). This core can be depicted only by studying vulnerability dynamics across space and time, using new operational tools such as the Enhanced Impact Chains (Albulescu and Armaş, 2024). Systemic risk, in turn, captures the potential for these disruptions to propagate across sectors and scales, resulting in widespread and unforeseen consequences. Such a perspective situates compound risk within the broader dynamics of interdependence, where systemic conditions shape not only the onset of these impacts but their amplification and persistence.

Flooding is among the most common and destructive natural hazards, expected to intensify in frequency and severity as a result of climate change (Xu et al., 2023). Particularly, in coastal environments, the combined action of oceanographic, hydrological, and meteorological drivers—such as rainfall, river discharge, winds, tides, and wave action—can produce complex compound events (Lucey and Gallien, 2022). While each of these drivers may individually trigger localized damages, their simultaneous or sequential occurrence often results in *Coastal Compound Flooding (CCF)* hazard, leading to more severe impacts than would be expected from any driver acting in isolation (Eilander et al., 2023). These interactions become especially critical in low-lying and estuarine regions, where the transitional character of these ecosystems intensifies the complexity of flood risk (Green et al., 2025). The joint occurrence of heavy rainfall and storm surge, for example, can more easily overwhelm standard protection thresholds (Couasnon et al., 2020). This type of compound hazard is particularly relevant for Flood Risk Management (FRM), emergency planning, and the insurance sector, as it challenges assumptions built around isolated events (Catto and Dowdy, 2021; Green et al., 2025).

In particular, FRM practices must address the limitations of traditional single-hazard assumptions when concurrent drivers occur. Managing CCF events involves strategies that account for interactions across spatial and temporal scales, while

remaining responsive to local conditions (Mishra et al., 2022). Since flood risk cannot be entirely eliminated, attention has increasingly shifted toward mechanisms that enhance the ability to cope with CCF events when they occur (Thieken et al., 2022). *Preparedness* plays a central role in this shift. As defined by the UNDRR, preparedness refers to the knowledge and capacities developed by institutions, communities, and individuals to anticipate, respond to, and recover from likely, imminent, or ongoing hazard events (UNDRR, 2017). It includes Early Warning Systems (EWS), contingency planning, and the institutional arrangements required to support timely and coordinated action. However, in the presence of CCF, the conditions under which preparedness operates become less predictable, and its effectiveness increasingly contingent on how well such complexity is accounted for (Simpson et al., 2023; Van Den Hurk et al., 2023).

Rather than replacing structural defences, strengthening preparedness serves as a complementary strategy in line with the increasing focus on non-structural measures that mitigate impacts and safeguard vulnerable communities (Scolobig et al., 2015; Fox-Rogers et al., 2016). It involves building capacities, developing tools, and enhancing coordination mechanisms to enable timely response and recovery. It is shaped by individual attributes, socioeconomic conditions, risk perception, and previous disaster experiences (Eze and Siegmund, 2024). Beyond its operational dimension, preparedness is inherently social, relying on inclusive processes that empower those at risk as active contributors to their own safety. As Maidl and Buchecker (2015) underline, its effectiveness hinges on genuine engagement and trust among local actors. Such principles are echoed in the Sendai Framework for Disaster Risk Reduction (SFDRR), which calls for the involvement of affected populations in designing and implementing risk reduction strategies (UNDRR, 2015). Monteil et al. (2022) emphasize that preparedness strategies are more effective when responsibility is clearly shared and social conditions that hinder engagement are addressed. This shift toward inclusive, community-centred approaches recognizes that disaster preparedness must go beyond technical solutions to adopt forward-looking strategies, such as prospective, corrective, compensatory, and community-based measures that actively engage local populations (Eze and Siegmund, 2024). Embedding local knowledge, fostering collaboration among diverse stakeholders, and addressing the root causes of vulnerability are essential for creating adaptive, equitable strategies capable of tackling systemic risks. A critical component of this transformation is the effective communication of the complexities of CF risks, ensuring that both individual and systemic perspectives are considered (Kruczkiewicz et al., 2021; Ward et al., 2022). By bridging gaps in knowledge and fostering trust among citizens, scientists, and policymakers, preparedness efforts can enhance FRM practices and enable more precise, timely responses. These efforts not only empower communities and strengthen resilience but also build collaborative networks that align societal and scientific goals, bringing the principles of DRR into practice.

Despite extensive research on FRM, critical gaps remain in understanding how to effectively prepare for CCF events. Studies have largely focused on characterizing the physical processes that drive these hazards, while comparatively less attention has been given to strategies for preparedness and management. Yet, the cascading effects and interdependencies that define compound events expose fundamental limitations in prevailing climate risk governance frameworks (Modrakowski et al., 2022). The scarcity of documented case studies further constrains the development of comprehensive frameworks, as current

methodologies tend to overlook the nuanced interplay between environmental, technical, and social dimensions. Bridging this gap calls for innovative frameworks that move beyond linear assumptions to reflect the systemic nature of compound risks. These efforts are essential not only to mitigate immediate impacts but to foster long-term resilience, ensuring that institutions and communities are better prepared to navigate the growing complexities of climate-related hazards (Sacchi et al., 2023). This study presents a systematic literature review that critically examines how FRM practices are evolving to address the intricate challenges of CF in coastal areas—regions where the interplay of vulnerabilities and flood drivers increases risks. The analysis addresses two guiding research questions:

- i. (RQ1) How are preparedness strategies evolving to integrate technical, environmental, and social dimensions in managing CCF risks?
  - ii. (RQ2) What is the role of governance and multi-stakeholder collaboration in enhancing flood preparedness?

By addressing these questions, the study advances the development of more effective preparedness frameworks by analysing how strategies are being reshaped in response to CCF risks across diverse coastal contexts (RQ1), and by improving understanding of the role of governance and collaboration in these processes (RQ2). This approach offers a grounded understanding of the conditions that enable or hinder anticipatory action, not as abstract goals, but as practices embedded in specific institutional and socio-environmental settings. Rather than proposing prescriptive solutions, the paper identifies key levers and recurring patterns that can inform more flexible, integrative, and context-sensitive responses. In doing so, it helps bridge the gap between conceptual debates and the operational realities of managing climate-related threats in increasingly complex risk landscapes.

We adopt a broad understanding of preparedness that goes beyond its conventional role in the DRR cycle—typically associated with EWS, contingency planning, and emergency readiness. Instead, it is framed as a multidimensional process encompassing anticipatory governance, infrastructural and ecosystem-based measures, and behavioural strategies aimed at reducing vulnerability prior to the manifestation of hazardous conditions. In this review, "preparedness strategies" are used in a broad sense to include both conventional preparedness activities (e.g., early warning systems, response planning) and longer-term adaptation measures (e.g., infrastructure upgrades, community capacity building). This expanded usage reflects the growing need for integrated and scalable responses to CCF risks, where the distinction between short-term and long-term interventions is often blurred in practice. This perspective aligns not only with emerging literature on integrated FM (Bark et al., 2021; Konami et al., 2021; De Silva et al., 2022; Sánchez-García et al., 2024), but also firmly grounded in Priority 4 of the SFDRR, which advocates for preparedness actions that include inclusive governance, resilient infrastructure, public education, psychosocial support, and the incorporation of risk reduction into development planning and post-disaster reconstruction (UNDRR, 2015).

#### 2. Background

As extreme weather events increase in frequency and intensity, the limitations of conventional FRM frameworks have become increasingly apparent. Approaches designed around isolated hazards or sector-specific responsibilities often fail to capture the interdependencies between social systems, infrastructure, and the cascading dynamics of compound events— resulting in unanticipated disruptions. These gaps become especially visible in CF scenarios in coastal and estuarine areas, where concurrent drivers can exceed design thresholds, disrupt coordination mechanisms, and expose systemic weaknesses in preparedness and response. (Curtis et al., 2022; Eilander et al., 2023). In Europe, for instance, CF events result in average annual damages of €1.4 billion, with Mediterranean regions especially affected by the joint effects of sea level rise and intense precipitation (Bevacqua et al., 2019; Lopes et al., 2022). While context-specific, the underlying challenges mirror those faced in other coastal settings exposed to multiple CF drivers.

In this evolving risk landscape, preparedness must move beyond traditional boundaries and embrace a more systemic lens—integrating technical, environmental, and social dimensions. Achieving this shift calls for the adoption of nonlinear and compound thinking to design cohesive strategies capable of responding to complex, interacting threats (Cegan et al., 2022; Van Den Hurk et al., 2023). This evolution reflects broader changes in FRM, which increasingly prioritize integrated, multisectoral approaches over isolated hazard-specific models (Sarmah et al., 2024). While international frameworks have laid important groundwork—particularly by highlighting the value of community engagement and resilience-building (Monteil et al., 2022) —it remains unclear whether, and to what extent, existing guidance and institutional practices have explicitly addressed the challenges of CCF or proved effective when such events have occurred.

Recent studies have begun to explore these uncertainties, offering initial guidance while also exposing areas that require further investigation. For example, Van Den Hurk et al. (2023) emphasize the necessity of integrating compound event considerations into DRR, highlighting tools such as advanced hydrometeorological forecasting, decision-support systems, and responsive infrastructure as promising pathways to strengthen preparedness. However, the study remains largely general in scope: key aspects of CCF—such as the interaction between storm surge and extreme rainfall—are only briefly addressed. Furthermore, while the authors advocate for scalable systems and interdisciplinary coordination, there is still limited clarity on how such approaches can be operationalized for CCF across diverse institutional and geographic contexts. Their call for integrated strategies that combine physical, social, and statistical dimensions is compelling, yet still conceptual. Bridging this gap requires targeted research and practice-oriented methodologies capable of translating these frameworks into actionable solutions for CCF preparedness under real-world constraints and rising climate pressures.

Chan et al. (2024) explore CCF risks in Chinese coastal cities, with particular attention to the interplay between storm surges and extreme rainfall as key drivers. Their study documents a set of institutional responses co-developed by central and municipal governments, including the deployment of real-time information technologies (e.g., mobile apps), coordinated emergency protocols, and the implementation of blue-green infrastructure under the "Sponge City" initiative. These measures signal a noteworthy shift toward hybrid approaches that combine engineering design with ecosystem-based adaptation.

Nevertheless, the analysis offers limited insight into how social processes—such as risk perception, local knowledge, or community involvement—are integrated into preparedness planning. This omission is critical, given the role of social dynamics in shaping preparedness. Furthermore, by focusing on only two interacting drivers, it offers targeted insights. This scope may constrain its applicability to broader CCF scenarios. While the findings demonstrate meaningful progress, their emphasis on the Chinese context—marked by strong central governance and rapid urbanization—constrains their transferability to regions with different socio-political and environmental settings. Although climate change is recognized, the focus on present measures leaves open questions about how preparedness can evolve under future compound conditions.

Building on recent advances, (Green et al., 2025) offer a comprehensive synthesis of research on CF, outlining key methodological challenges—particularly the absence of standardized approaches and the complexity of modelling interactions among multiple drivers. Their call for inter-comparison projects and hybrid modelling strategies represents a timely effort to consolidate fragmented knowledge and improve our capacity to characterize compound hazards under increasing climatic uncertainty. Importantly, the study also highlights the relevance of embedding CCF scenarios into infrastructure planning, advocating for anticipatory measures such as Nature-based Solutions (NbS), updated hazard maps, and EWS. While these recommendations align with broader preparedness objectives, the discussion remains largely centred on technical and

As CCF gains relevance, questions persist about how preparedness operates when multiple drivers interact across time and space. Traditional approaches—shaped by single-hazard assumptions—often struggle to reflect the overlapping processes, competing priorities, and the complex conditions that influence institutional frameworks, social dynamics, and individual decisions. This work contributes to ongoing efforts to understand how preparedness—understood a multidimensional process that integrates governance, infrastructure, NbS, and behavioural measures to reduce vulnerability before hazards occur—has been addressed so far, and how compound thinking is beginning to take form within the domain of FRM—while also reflecting on the directions such thinking may take as compound risks become increasingly prominent.

modelling domains, offering limited insight into the governance or societal mechanisms required to translate such measures into practice. As a result, the operational implications of these strategies—particularly in diverse or resource-constrained contexts—remain underexplored, underscoring the need for integrative approaches that connect methodological progress with

#### 3. Methods

inclusive, actionable frameworks.







This systematic literature review explores how preparedness strategies for CCF have evolved in coastal and estuarine environments, where multiple drivers—such as storm surges, river discharge, and extreme rainfall—interact to generate heightened impacts. To capture the complexity of these interactions and the preparedness efforts that respond to them, the study is guided by two broad research questions that frame the examination of this multifaceted topic:

### I. How are preparedness strategies evolving to integrate technical, environmental, and social dimensions in managing CCF risks?

This question seeks to explore how diverse studies conceptualise the integration of technical elements—such as resilient infrastructure predictive models, and EWS—with environmental and social components, including community engagement, and risk perception. It examines how this integration is framed and how it responds to the complexity introduced by multiple interacting drivers. Instead of evaluating these strategies against a predefined framework, the analysis identifies recurring patterns and tensions within the broader context of FRM.

#### II. What is the role of governance and multi-stakeholder collaboration in enhancing CCF preparedness?

The aim is to understand how governance frameworks and collaborative arrangements among governments, local communities, and private actors shape preparedness efforts. The analysis includes examining participatory governance, the integration of indigenous and local knowledge, and the ways in which such interactions support more adaptive and inclusive FRM strategies.

By aligning with the SFDRR and concentrating on recent research trends, this study highlights the critical interplay between physical and social processes as essential to advancing preparedness strategies.

#### 3.1. Research approach and database overview






The methodology follows the PRISMA (Preferred Reporting Items for Systematic Reviews and Meta-Analyses) framework (Page et al., 2021), ensuring a structured and transparent approach to analysing relevant literature. To identify relevant studies, we carried out a systematic search in the Web of Science (WoS) database, applying a multi-layered strategy aimed at capturing research related to preparedness for CF in coastal areas, with a particular focus on community resilience and FRM. This approach was informed by previous reviews on similar topics (Kuhlicke et al., 2023; Sun et al., 2024). No start date limit was applied; all records available in the WoS database up to September 2024 were included in the review. The search was organized into two main steps, combined using an OR operator, allowing articles that matched either word string to be included:

- **First Step:** A search based on topics (TS) that incorporated terms related to CCF, preparedness, and specific geographical features, enhanced by an Author Keywords (AK) query to ensure the inclusion of relevant terms connected to preparedness and flooding.
- Second Step: A more targeted search in the Title (TI) and Abstract (AB) fields, using terms directly related to CCF and preparedness, further complemented by an Author Keywords (AK) query for technical terms.

The specific search syntax used in WoS is presented in Table 1. This comprehensive approach allowed us to capture a broad range of studies focused on preparedness for flooding in coastal areas, including compound events, while ensuring relevance through multiple layers of keyword filtering. The selection was limited to peer-reviewed articles in English, with no restrictions on publication date for the available information.




Table 1. Search strategy and terms used in the PRISMA-based systematic review.

| Search Structure | Search Terms                                                                                                                                                                                                                                                                                                                                                                                                                                                                                                                                                                                                                                                                                                                                                                                   |  |
|------------------|------------------------------------------------------------------------------------------------------------------------------------------------------------------------------------------------------------------------------------------------------------------------------------------------------------------------------------------------------------------------------------------------------------------------------------------------------------------------------------------------------------------------------------------------------------------------------------------------------------------------------------------------------------------------------------------------------------------------------------------------------------------------------------------------|--|
| First Step       | (TS= (("compound flood*" OR "coastal flood*" OR "compound coastal" OR "compound extreme*" OR "compound effect" OR "flood*" OR "inundation") AND ("preparedness" OR "disaster preparedness" OR "community resilience" OR "resilience" OR "coping capacity" OR "adaptive capacity" OR "early warning" OR "contingency planning" OR "community engagement" OR "decision making" OR "local knowledge" OR "indigenous knowledge" OR "traditional knowledge") AND ("estuar*" OR "delta*" OR "lowland*" OR "river mouth*" OR "wetland*" OR "tidal area*" OR "marshland*" OR "bay*" OR "transition zones")) AND AK=("preparedness" OR "disaster preparedness" OR "compound flood*" OR "coastal flood*" OR "compound coastal" OR "compound extreme*" OR "compound effect" OR "flood*" OR "inundation")) |  |
| Second Step      | (TI=("compound flood*" OR "coastal flood*" OR "combined risk" OR "compound effect" ("compound climate") AND AB=("preparedness" OR "disaster preparedness" OR "resilience" ("risk perception" OR "community resilience" OR "coping capacity" OR "early warning" ("adaptive behaviour" OR "contingency planning" OR "estuar*") AND AK=("preparedness" ("disaster preparedness"))                                                                                                                                                                                                                                                                                                                                                                                                                 |  |

Note: The asterisk symbol (\*) is used as a truncation operator to include all possible word endings (e.g., flood\* retrieves flood, floods, flooding). Search field abbreviations include Topics (TS), Author Keywords (AK), Title (TI), and Abstract (AB).

The initial analysis of search results from the WoS database provided a broad perspective on flooding preparedness research, capturing diverse topics and approaches. A total of 874 articles met the defined criteria, addressing key themes such as disaster preparedness, resilience, and flood management across various environments, including coastal and estuarine regions. The use of the broader term "coastal flooding" was intended to capture studies published prior to the widespread adoption of the compound event framework. Consequently, the retrieved literature spans a wide range of disciplinary approaches and timeframes. Many of these contributions focus on the hazard dimension of flood risk, particularly through measures implemented during the preparedness phase of FRM. This broad scope reinforces the need to refine the analysis toward compound hazard configurations, ensuring coherence with the specific objectives of this review. In line with our broadened conceptualization of "preparedness strategies" as encompassing both short-term preparedness and long-term adaptation, we

included studies that addressed either domain—provided they explicitly contributed to risk reduction in the context of CCF.

This inclusive approach reflects the practical and temporal convergence between preparedness and adaptation, and guided the application of our inclusion and exclusion criteria.




To refine the initial dataset and enhance its focus and relevance, we used the Python package *LitStudy*. This tool facilitated the selection and in-depth analysis of the identified publications through visualizations, bibliographic network analysis, and natural language processing techniques (Heldens et al., 2022). Figure 1 illustrates the *word cloud* generated by *LitStudy*, highlighting key themes centred on adaptation, risk management, and community resilience. Prominent terms such as "risk," "adaptation," "communities," and "vulnerability" emerged, reflecting the focus on preparedness strategies. Technical aspects of flood management, including forecasting and urban water governance, were also evident, with clusters emphasizing predictive models, EWS, and urban delta management. Additionally, ecological themes underscored the role of natural systems, particularly wetlands and floodplains, in flood mitigation.

Figure 1. Word cloud visualization of the topics identified in the reviewed articles. Topics were derived using the Python package LitStudy, which applies natural language processing and bibliographic network analysis to extract thematic structures from scientific texts. The resulting word clouds highlight dominant themes related to CCF, adaptation, and risk management. Terms associated with thematically unrelated domains—such as oil recovery, seed banks, and tectonic hazards—were also detected and removed to ensure conceptual consistency across the analysis.

Beyond the dominant themes aligned with flood preparedness, the word cloud also revealed peripheral clusters related to ecological studies—particularly those focused on seed banks, germination processes, and plant propagation—as well as hazards of tectonic origin, such as earthquakes and tsunamis. While thematically adjacent, these topics fall outside the scope of climate-related flood dynamics (Hendry, 2021). Our focus is on CCF events arising from the interaction of meteorological, hydrological, and oceanographic drivers under climate variability and change, in coastal settings. To ensure conceptual coherence and maintain a consistent basis for comparison, studies addressing tectonic hazards or unrelated ecological processes were systematically excluded. The following keywords were removed from the search in the Topic (Ts) field: earthquake, species, tsunami, seed bank, habitat, germination, mangrove, irrigation, lake, soil, bank, food insecurity, organic matter, trees, sediment, dam, ice jam, drought, groundwater, energy. This refinement led to the removal of 152 publications, resulting in a final dataset of 722 articles. The choices underpinning this step are acknowledged and further examined in the limitations section.

#### 3.2. Article screening and data analysis using Active Learning Process

Subsequently, the Python library ASReview Lab, an open-source machine learning tool, was used to streamline the systematic screening and labelling of large-scale textual datasets relevant for this study. ASReview focuses on the title and abstract screening phase—a critical bottleneck in systematic reviews—by combining human expertise with machine learning to efficiently prioritize relevant records.

The process begins with the user uploading the dataset containing metadata (titles, abstracts, and other relevant information) into the software. Author names and citation networks are excluded to prevent bias. Initial *prior knowledge* is provided by labelling at least one relevant and one irrelevant record, as the foundation for training the first machine learning model. The model predicts the relevance of remaining records based on their textual features (titles and abstracts) while purposefully excluding author names and citation networks to prevent bias. This cycle, known as *Researcher-In-The-Loop (RITL)*, involves iterative collaboration between the reviewer and the machine. The system ranks records by predicted relevance and presents them to the reviewer for labelling. The reviewer assigns binary labels (1 for relevant, 0 for irrelevant), and the model is retrained after each session to refine its predictions. This process continues until a user-defined stopping criterion is met, such as the reviewer's confidence that all relevant records have been identified. By prioritizing the most probable records, ASReview significantly reduces the effort required for screening while maintaining transparency and control in the decision-making process. Previous studies have shown that this methodology can reduce screening time by up to 95% without compromising review quality (Van De Schoot et al., 2021). In our study, we manually labelled a set of 34 abstracts selected through random sampling from the retrieved corpus. Titles were deliberately excluded to ensure that relevance assessments relied solely on the substantive content of the abstracts, avoiding potential bias from overly general or misleading phrasing. Each abstract was

evaluated for alignment with the study's research questions and thematic scope, and assigned a binary label (relevant/irrelevant). This categorised subset served as the seed data to initiate the Active Learning Process.







To further enhance the efficiency of the review process, we incorporated a fine-tuned BERT (Bidirectional Encoder Representations from Transformers) model, a state-of-the-art natural language processing tool renowned for its ability to capture nuanced contextual relationships within text. BERT's bidirectional architecture enables it to process entire sentences in context, making it particularly effective for tasks such as document classification. By fine-tuning the model on a subset of labelled data specific to our study, we automated the initial classification of articles retrieved from the WoS database. While BERT provided automated pre-screening, this step did not replace the critical role of the human reviewer. Instead, the pre-labelled data served as input for ASReview, which facilitated an iterative RITL process. In this process, the reviewer actively validated and refined the classification results, ensuring that relevant studies were accurately identified. The synergy between BERT's robust text analysis capabilities and the reviewer's expertise not only accelerated the screening of large datasets but also preserved the rigor and reliability of manual review. This combined approach enhanced the reproducibility of the methodology and reduced the inherent subjectivity of manual review.

After applying the selection methodology to the initial dataset, 49 articles were identified as highly relevant and prioritized for in-depth analysis. These works were selected based on their alignment with the research questions, ensuring that only studies with the greatest potential to meaningfully inform the review were retained. Given the complexity of addressing interacting flood drivers, preparedness strategies that explicitly target compound hazard processes have only recently begun to gain traction. As noted by Serinaldi et al. (2022), persistent ambiguity in the terminology means that such phenomena are repeatedly examined under broader categories—such as coastal flooding—without being explicitly labelled as compound. To address this conceptual overlap and ensure a comprehensive perspective, the scope of the review was deliberately expanded to include a wider range of coastal flood preparedness literature. Relevance to compound processes was assessed during the full-text analysis.

Figure 2 summarizes the systematic review process following the PRISMA framework, from the initial identification of 874 records in the WoS database, through screening via tools such as LitStudy and ASReview, to the final inclusion of 49 full-text articles. Each study was reviewed to extract core characteristics—geographic context, flood drivers, and preparedness aspects highlighted. Emphasis was placed on the treatment of conceptual uncertainties, methodological difficulties, and attempts at operationalization. Limitations acknowledged by the original authors were also documented.

These steps aimed to reduce subjective judgement during the screening phase and to enhance the transparency and reproducibility of the review process. While ASReview and BERT improve efficiency and consistency by reducing manual effort and limiting subjective choices, the final output still depends on earlier decisions—such as how search queries are formulated, and which records are initially labelled as relevant. These aspects are further discussed in the limitations section.

Figure 2. Review workflow following the PRISMA framework. A total of 874 records were retrieved from Web of Science. No duplicates were identified. Topic clustering using *LitStudy* supported the refinement of the search strategy by identifying thematically unrelated content, leading to the exclusion of 152 records through targeted keyword removal. The remaining articles were screened using *ASReview* for title and abstract relevance. A final set of 49 articles was selected for full-text review. The integration of automated tools contributed to a structured and coherent selection process.

#### 4. Results



#### 4.1. Literature Trends and Research Growth

The initial corpus of 874 articles provides a broad overview of how flooding and preparedness have been approached across disciplines. Although heterogeneous in content, the dataset reveals consistent patterns in the framing of these topics. A preliminary analysis of disciplinary categories indicates a marked concentration in Environmental Sciences, Ecology, and Atmospheric Sciences (see Figure 3). This distribution reflects a prevailing emphasis on physical processes and environmental modelling FRM. In contrast, contributions associated with the Social Sciences appear underrepresented, suggesting a limited engagement with institutional, behavioural, and socio-economic dimensions.

Figure 3. Distribution of research fields in the corpus. Research areas follow the classification scheme provided by Web of Science, which may assign multiple categories to a single publication. This overlap leads to a total count that exceeds the number of unique articles.

The number of displayed categories may vary depending on user-defined parameters in the visualization tool. Environmental Sciences, Ecology, and Meteorology appear most frequently, suggesting a predominant focus on biophysical dimensions, while Social Sciences are notably less represented.

The observed asymmetry may reflect how research trajectories have developed over time, shaped by differing priorities as well as methodological, theoretical and disciplinary challenges. Historically, flood risk has been addressed through technical and hazard-centred frameworks, with a strong emphasis on hydrometeorological drivers, modelling, and structural measures, leaving less space for analysing how societies perceive, experience, and respond to flood events (Lechowska, 2022). Sociopolitical dimensions are often treated as secondary, rather than central to how risks are understood and managed. Furthermore, inconsistent terminology and conceptual ambiguity, especially in definitions of multi-hazard and compound events, have contributed to the "fragmentation of the literature," generating redundancy and confusion that hinder interdisciplinary collaboration (Serinaldi et al., 2022; Green et al., 2025). Methodological constraints such as limited data availability, lack of standardization, and the context-dependence of social indicators also restrict their integration (Girons Lopez et al., 2017; Vanelli et al., 2022). Importantly, social and behavioural science research on these topics has been underfunded until the last decade. This undermined not only the theoretical but also the disciplinary development of risk perception, preparedness and communication studies. A more integrated approach is needed to inform preparedness strategies that reflect both the physical dynamics of CCF and the ways in which societies experience and respond to them.

Beyond disciplinary orientation, observing the temporal distribution of publications offers a sense of how academic attention to the topic has developed over time (see Figure 4). Around 6% of studies were published between 1994 and 2011, followed by approximately 9% during 2012–2015. The remaining 85% concentrate in the period from 2016 to 2024. This steep increase does not imply a transformation in research focus, but it provides a structured basis to examine whether the expansion in volume has been accompanied by a broadening in scope, methods, or thematic emphasis. In this regard, early contributions—especially those prior to 2010—were often fragmented and typically addressed single hazards such as riverine flooding, storm surge, or sea-level rise (Burch et al., 2010; Slinger et al., 2007; Zaalberg et al., 2009). These studies tended to overlook the interdependencies among drivers, resulting in a compartmentalized understanding of flooding processes and a limited engagement with systemic risk perspectives. The period after 2010 marked a notable shift, as the shortcomings of hazard-specific approaches became more evident. Concepts such as "compound," "multi-hazard," and "risk management" began to gain traction, reflecting growing awareness of the interconnected nature of natural hazards. This conceptual shift was further supported by global initiatives promoting multi-hazard and cross-sectoral approaches to disaster preparedness, with particular attention to cascading effects and systemic vulnerabilities.

From 2012 onwards, references to preparedness and compound events become increasingly visible, marking a subtle but important evolution in research framing. Yet, this trend should be interpreted in light of broader shifts affecting academic production. As noted by Ioannidis et al. (2018), recent decades have seen a sharp rise in publication rates, greater international collaboration, and the expansion of the global research community. Priem et al. (2022) estimate that over 60% of all scientific articles have been published since 2000, underscoring how structural transformations in the research field may amplify certain patterns. In this context, the surge in publications related to compound risks may reflect not only an emerging awareness of systemic dynamics but also the momentum of a more prolific and interconnected academic environment.

Consistent with these trends, the post-2012 period is characterised not only by a quantitative expansion in CCF and preparedness research, but also by a gradual diversification of its conceptual and methodological landscape. This growth aligns with a broader reconfiguration of natural hazard studies, catalysed by the formal introduction of *compound events* in the IPCC's SREX report (IPCC, 2012). A notable consolidation of this trend is evident after 2015, coinciding with the adoption of the SFDRR, which marked a strategic shift from *disaster management* to *disaster risk management*. By prioritising anticipatory action, early warning, and systemic resilience, Sendai advanced a multi-hazard and risk-informed approach that aligns closely with the emerging discourse on CCF. This convergence between policy and scientific agendas likely contributed to the increased academic focus on CF and preparedness as interdependent concerns. During this transition, various disciplinary perspectives began to confront the limitations of univariate risk characterisation: Yasuhara et al. (2011), for instance, explored the combined impacts of climate and geophysical extremes on coastal infrastructure, introducing the notion of "compounded natural hazards"; Watkins, (2013) called attention to temporally clustered extremes and "wild" fluctuations, challenging the assumptions of traditional hazard modelling; and Zheng et al. (2013) demonstrated statistical dependence between storm surge

and rainfall, undermining the reliability of univariate models in FRM. While emerging from distinct domains, these studies collectively signal a transition toward more integrated representations of compound events.

385

390

This initial framing was further elaborated by Leonard et al. (2014), who emphasized the multivariate nature of CCF and the need for analytical tools capable of capturing such complexity. Freire et al. (2016) subsequently underscored the importance of preparedness in transitional ecosystems, particularly estuarine regions where tides, river flows, wind, and waves converge. Their work highlighted the socio-economic complexities of these systems and emphasized the need for integrated, multi-hazard preparedness strategies capable of addressing the cascading impacts of CCF.

**Figure 4. Annual distribution of published articles.** A marked increase is observed after 2012, with sustained growth consolidating from 2015, a sharp rise from 2018, and a peak in 2022–2023. The value for 2024 refers to records indexed up to September, as the search preceded the end of the year.

Figure 5 offers additional insights into the temporal evolution of thematic emphasis, capturing how certain research domains have gradually gained prominence while others have remained secondary. Although the presence of specific keywords does not guarantee conceptual depth, their distribution provides a useful proxy for identifying shifting priorities within the field. Terms linked to *compound events*, *preparedness*, and *uncertainty* appear with increasing frequency, suggesting a gradual incorporation of systemic and anticipatory dimensions. In contrast, references to *local knowledge* and *community engagement* remain sparse, showing limited integration of community-based perspectives. The distribution is not uniform: while certain themes gain presence, others persist at the margins. This pattern outlines a field in expansion, but not necessarily in balance—where some domains continue to be explored more systematically than others.

Figure 5. Temporal Evolution of Data-Driven Research Themes. The heatmap illustrates the changing prominence of key terms identified through frequency analysis of the abstract corpus. Color intensity represents a 5-year trailing moving average of each term's frequency, calculated to smooth annual fluctuations and capture underlying trends. A non-linear scale is employed to enhance the visibility of variations at lower frequencies, while all values above 100 are saturated to the maximum color intensity. This visualization allows for the identification of emerging, persistent, or declining research topics. Colormap: "lipari\_r" from Scientific Colour Maps (Crameri et al., 2020).

Compound events and preparedness now appear more consistently, reflecting a growing concern with the interconnected nature of hazards and the need to plan. Their rise suggests a move away from hazard-specific views toward more integrated framings. Uncertainty remains a common reference, but often in narrow terms—linked to models or data—without fully addressing its social or institutional implications. In contrast, local knowledge and community engagement appear less frequently. These topics are mentioned but rarely placed at the core of FRM frameworks. The observed pattern reflects not only an expansion in thematic scope, but also a progressive convergence toward a shared vocabulary that mirrors shifts in international agendas and interdisciplinary discourse.

The upward trajectory in the frequency and diversity of key terms signals a maturing research landscape, transitioning from fragmented hazard-specific studies to interdisciplinary, systems-based frameworks. However, this evolution remains incomplete. The limited attention to social vulnerability, participatory governance, and localized knowledge indicates that technical and infrastructural solutions continue to dominate preparedness efforts. Moving forward, the research community must embrace the inherent complexity of CCF by developing adaptive, community-driven strategies that integrate governance, equity, and cascading impacts into preparedness frameworks. Such an approach will not only strengthen resilience but also ensure that preparedness strategies are robust, inclusive, and sustainable, effectively addressing the increasing challenges posed by climate change.

#### 4.2. Overview of Selected Articles

420

435

- From the detailed review of the 49 articles identified through systematic screening, 45 were identified as directly relevant to the study's focus on preparedness for CF in coastal regions. These studies offer critical insights into the integration of technical, environmental, and social dimensions in managing CCF risks, as well as the role of governance and multi-stakeholder collaboration. Although informative, the remaining four articles addressed either non-coastal contexts or broader aspects of preparedness, and were therefore considered less central to the study's scope.
- To facilitate comparative analysis, Table 2 organizes the studies by country and groups them into four broad thematic clusters, based on their primary analytical emphasis. This structure enables a cross-cutting view of how different dimensions of preparedness—social, institutional, and technical—have been explored in the literature, and how these vary across geographic and temporal contexts. The table is intended as a mapping tool to support further synthesis and discussion, not as a definitive typology. Perceptions and behavioural responses are addressed in studies from a broad range of geographic contexts.
  - Forecasting and modelling are covered primarily in recent contributions from China. Governance and participatory approaches appear in fewer cases but span multiple regions. Finally, case studies are concentrated in a small set of countries, with many others absent from the sample.

Table 2. Classification of studies by thematic focus, geographic area, and publication year.

| Key Topic                                   | Geographic focus                  | Year and References                                                                                            |
|---------------------------------------------|-----------------------------------|----------------------------------------------------------------------------------------------------------------|
| Perceptions and<br>behavioural<br>responses | Spain                             | Raaijmakers et al. (2008)                                                                                      |
|                                             | Botswana                          | King et al. (2018); Motsholapheko et al. (2011);                                                               |
|                                             | Vietnam                           | Casse et al. (2015); McElwee et al. (2017); Ngo et al. (2020)                                                  |
|                                             | The Netherlands                   | De Boer et al. (2016); Mol et al. (2020);                                                                      |
|                                             | Fiji                              | Nolet, (2016)                                                                                                  |
|                                             | France                            | Lemée et al. (2019, 2022); Rambonilaza et al. (2016)                                                           |
|                                             | Indonesia                         | Maryati et al. (2019)                                                                                          |
|                                             | USA                               | De Koning et al. (2019); Johns et al. (2020); Richmond and Kunkel, (2024)                                      |
|                                             | Myanmar                           | Lwin et al. (2020)                                                                                             |
|                                             | Brazil                            | Pereira Santos et al. (2022)                                                                                   |
|                                             | Italy                             | Sacchi et al. (2023)                                                                                           |
|                                             | Bangladesh                        | Faruk and Maharjan (2023)                                                                                      |
|                                             | Nigeria                           | Michael (2024)                                                                                                 |
| Compound events                             | China                             | Chan et al. (2024); Du et al. (2020); Guo et al. (2023); Sun et al. (2024); Xu et al. (2024); Yu et al. (2023) |
| forecasting                                 | Mozambique                        | Matos et al., (2023)                                                                                           |
|                                             | Netherland                        | Gerritsen (2005); Oukes et al. (2022)                                                                          |
|                                             | Botswana                          | Shinn (2018)                                                                                                   |
| Governance and policy                       | China                             | Liang et al. (2017); Xie et al. (2023)                                                                         |
| poncy                                       | Canada                            | Chang et al. (2020)                                                                                            |
|                                             | UK                                | Coletta et al. (2024)                                                                                          |
|                                             | Netherland                        | Slinger et al. (2007)                                                                                          |
|                                             | Botswana                          | Motsholapheko et al. (2015)                                                                                    |
|                                             | UK, Netherland,<br>USA, Indonesia | Jeuken et al. (2015)                                                                                           |
| Participatory and                           | USA                               | Cheung et al. (2016)                                                                                           |
| innovative methods<br>for FRM               | Portugal                          | Freire et al. (2016)                                                                                           |
|                                             | Ghana                             | Yankson et al. (2017)                                                                                          |
|                                             | Italy, Portugal                   | Martinez et al. (2018)                                                                                         |
|                                             | China                             | Chan et al. (2023)                                                                                             |
|                                             | Vietnam                           | Binh et al. (2020)                                                                                             |
|                                             | Bangladesh                        | Azad et al. (2022)                                                                                             |

Thematic topics were identified through qualitative content analysis of each study's aims, methodological approach, and main findings.

This grouping intends to highlight recurring analytical concerns across contexts and periods. The resulting classification is meant as a preliminary and illustrative framework, rather than a definitive categorization.

Figure 6 summarizes key patterns across the reviewed studies. Panel a) maps the spatial distribution of case studies, distinguishing those explicitly addressing CCF from those examining coastal flooding more generally. The distribution is not spatially uniform and reflects how research attention has been allocated geographically. Panel b) captures how the contributing elements of compound events are reported. While several studies specify individual drivers—such as storm surge, river discharge, or rainfall—others refer instead to categories like multi-drivers, CCF, or compound risk, without detailing specific components. Panel c) shows the number of studies by country. The distribution is heterogeneous, with research activity concentrated in a limited number of contexts. By analysing key studies, this review sheds light on the challenges and limitations of existing approaches, offering insights that can inform more adaptive, inclusive, and actionable strategies to enhance resilience and preparedness in coastal regions increasingly affected by CF risks.

Figure 6. Global Perspectives on Flood Preparedness Studies: (a) Geographic distribution of preparedness studies, distinguishing between those focused on CCF and those addressing coastal flooding more broadly. (b) Representation of contributing elements in CCF studies. Categories include individual drivers (e.g., storm surge, river discharge, rainfall) as well as more general terms (e.g., multi-drivers, CCF, compound risk). (c) Total number of studies by country, visualized in a bar chart to showcase regional trends in research efforts.

In addition to its descriptive layout, Figure 6 reflects structural patterns in how CCF preparedness has been approached. The simultaneous presence of defined drivers (e.g., storm surge, river discharge) and broader categories (e.g., multi-drivers, CCF, compound risk) indicates that compound processes are represented at varying levels of abstraction, often without explicit articulation of their components. In several cases, the compound nature of the hazard is acknowledged but not formally disaggregated, resulting in formulations that remain general in scope. The dominant focus lies on hydrometeorological variables directly linked to flood generation, such as coastal water levels and rainfall. However, a few studies mention, tangentially, other less frequent related drivers—such as *groundwater flooding* (Green et al., 2025)—that, while relevant in broader compound event typologies, remain marginal within the selected corpus. This fact suggests a prevailing emphasis on short-term, high-intensity interactions, with less attention to slower or antecedent climatic processes. Spatially, the concentration of case studies in a small number of countries defines a selective empirical base that influences not only what is analysed, but also how CCF is framed. Rather than pointing to a unified field, the figure shows a multiplicity of entry points and analytical choices shaped by context, data availability, and disciplinary orientation.

#### 4.2.1. Evolution of preparedness strategies and integration of different dimensions

A marked transition from isolated, hazard-focused measures to integrated approaches that simultaneously address technical, environmental, and social dimensions has been identified. This shift reflects an evolving recognition that CCF risks—emerging from the interplay of multiple drivers such as storm surges, rainfall, and sea-level rise—cannot be effectively mitigated through traditional, siloed interventions. The following analysis delineates this temporal evolution and provides evidence from the literature to explicitly address the research question.

#### • Pre-2010: Technical Dominance



Publications describing preparedness efforts before 2010 were dominated by hazard-specific, infrastructure-based solutions aimed at mitigating singular risks. These measures, while technically robust, often excluded environmental and social dimensions, limiting their capacity to address the systemic nature of CCF. For instance, the Netherlands' Delta Plan (Gerritsen, 2005) epitomized this approach with its focus on advanced dyke systems, storm surge barriers, and hydraulic modelling. Though effective in managing storm surges and sea-level rise, these interventions lacked adaptability to cascading effects or simultaneous hazards. Environmental considerations were peripheral, limited to augmenting engineered defences with natural dunes, while social engagement has been conducted with different types of awareness and preparedness campaigns mainly aimed at addressing conflicts (e.g. with NGOs or other organisations questioning ecological and environmental impacts of the programme).

#### • 2010–2020: Transitioning Toward Integration







The period between 2010 and 2020 marked a pivotal transition, driven by the recognition of limitations in traditional methods. Emerging hybrid approaches sought to integrate technical, environmental, and social strategies, although still in its early stages. For example, Portugal (Freire et al., 2016) adopted GIS-based hazard mapping to enhance flood preparedness, while Fiji (Nolet, 2016) emphasized the preservation of wetlands and mangroves as natural buffers against flooding. Social dimensions gained prominence, with efforts in China (Liang et al., 2017) leveraging informal networks and community-based initiatives to enhance urban preparedness. However, these advancements were often fragmented, and frameworks for addressing the interaction of multiple flood drivers—such as urban runoff, tidal forces, and extreme rainfall—remained underdeveloped. Despite these challenges, this period laid the groundwork for a broader understanding of CCF as a complex, multi-dimensional risk requiring collaborative solutions.

#### • Post-2020: Toward Holistic and Adaptive Approaches

Post-2020, preparedness strategies have embraced the complexity of CCF, integrating advanced technical tools with adaptive, community-focused approaches. Coupled hazard models and bivariate statistical analyses now enable planners to simulate interactions between multiple drivers. For instance, China (Sun et al., 2024) employs hydrodynamic models to predict cascading impacts, while case studies in the UK (Coletta et al., 2024) combine socio-hydrological frameworks with blue-green infrastructure to mitigate long-term flood risks.

NbS have emerged as central to these strategies. Programmes like China's Sponge City initiative (Chan et al., 2024) integrate wetlands and mangroves into urban hydrology restoration, while Nigeria (Michael, 2024) incorporates indigenous practices and gender-focused adaptations to address systemic vulnerabilities. These examples highlight the increasing importance of aligning environmental restoration with technical and social measures. Social inclusion now defines modern preparedness, with participatory governance and equitable decision-making shaping interventions. Case studies in Mozambique (Matos et al., 2023) integrates community surveys into planning, amplifying local knowledge, while other cases in Italy (Sacchi et al., 2023) apply behavioural psychology to address biases in risk perception. Such initiatives reflect a shift from reactive measures to anticipatory frameworks that prioritize resilience.

Figure 7 further reinforces the narrative of this temporal evolution, emphasizing the increasing complexity and interconnectedness of technical, environmental, and social dimensions. Historically, flood preparedness has focused on technical solutions such as risk assessments, forecasting models, and EWS that consider multiple flood drivers. Techniques like hydrodynamic modelling and statistical frameworks have greatly enhanced the prediction of flood zones and inundation scenarios, which are pivotal for mitigation planning (Xu et al., 2024).

Figure 7. Temporal Evolution of Technical, Environmental, and Social Dimensions in Preparedness Strategies for CCF. This visualization presents the evolution of preparedness strategies for CCF, comprising technical, environmental, and social dimensions. It illustrates connections between countries, methodologies, and thematic areas, showing trends, shifts in focus, and the increasing integration of interdisciplinary approaches. An interactive version of this figure is available at: https://doi.org/10.5281/zenodo.15848355 (Gomez et al., 2025).

While technical advancements have flourished, their integration into local risk reduction efforts remains insufficient. Coastal and estuarine communities often lack awareness of the compounded risks they face, and technical insights frequently fail to translate into actionable community plans. Moreover, as Sacchi et al. (2023) notes, individuals tend to oversimplify their risk assessments in the face of compound climate-related hazards, focusing on a single dominant factor instead of considering the complexity of multiple interacting drivers. This cognitive simplification can lead to incomplete evaluations, weakening mitigation and preparedness efforts.

A regional analysis reveals diverse trajectories shaped not only by economic resources, but also by institutional maturity, environmental priorities, and sociocultural dynamics:






• Europe: Across European contexts, preparedness strategies for CCF reflect a longstanding institutional investment in technical and infrastructural solutions, coupled with a gradual evolution toward more integrated, socio-environmentally attuned approaches. This progression has been supported by a common technical and institutional baseline across Member States, underpinned by key policy frameworks such as the EU Floods Directive and the Water Framework Directive. Together, these instruments have standardized hazard mapping, data integration, and basin-scale planning across Europe. While not originally designed for compound events, they provide an operational foundation upon which more integrated, multi-hazard approaches can gradually evolve. Countries such as the Netherlands, the United Kingdom, Portugal, France, Spain, and Italy exhibit high levels of technological maturity, as evidenced by the widespread implementation of hydrodynamic modelling, flood scenario simulations, and GIS-based hazard mapping. In the Dutch case, the Delta Plan stands as an example where engineered infrastructures—including dykes, storm surge barriers, and inland retention basins—are embedded within a broader framework of land-use regulation and polder-based environmental management (Gerritsen, 2005).

However, the robustness of these systems does not lie solely in their technological sophistication but in their increasing capacity to accommodate cross-sectoral integration. The UK, for instance, has advanced toward hybrid strategies that combine blue-green infrastructure with socio-hydrological models, aiming to bridge long-term climate adaptation with real-time operational planning (Coletta et al., 2024). Urban regeneration and climate-responsive drainage schemes reflect this shift, supported by institutionalized participatory mechanisms that incorporate stakeholder perspectives into scenario development and decision-making processes.

Yet, despite these advances, persistent limitations emerge when interrogating the extent to which preparedness strategies address structural inequalities and heterogeneous vulnerabilities. While public awareness campaigns and targeted communication have improved risk perception at the population level, equity-oriented planning remains marginal. The institutional focus on technical optimization often overlooks the differentiated capacities of communities to engage with, respond to, or benefit from these interventions. As such, even in high-capacity settings, preparedness may fall short in ensuring inclusive resilience, particularly when solutions are generalized across diverse social landscapes without adequate consideration of marginalized groups or localized knowledge systems.

Asia: Particularly in rapidly urbanizing regions such as China, strategies suggest an emergent synthesis of technical innovation and environmentally grounded interventions. The evolution of FRM in these settings reflects both the imperative to address multi-hazard contexts and the institutional ambition to operationalize them. China's Sponge City Programme exemplifies this trajectory, combining hydrodynamic engineering with NbS including wetlands, mangroves, and permeable surfaces to restore urban hydrological cycles and reduce flood vulnerability (Chan et al., 2024). This paradigm shift is further supported by the integration of advanced statistical modelling, dynamic simulation, and multidriver scenario analysis (Sun et al., 2024), enabling more granular assessments of cascading impacts and compound interactions. Nevertheless, the consolidation of these technical and environmental dimensions has not been mirrored by a corresponding strengthening of the social axis of preparedness. While informal networks and local capacities—such as those observed in Chinese urban neighbourhoods—often contribute to adaptive behaviours and bottom-up responses (Liang et al., 2017), their institutional anchoring remains weak. Top-down governance structures tend to dominate, resulting in fragmented or ad hoc social strategies that lack consistent incorporation into formal planning frameworks. As a result, preparedness in the region is characterized by a high degree of technical and environmental ambition but constrained by the challenge of embedding equity and participation within multilevel governance regimes. The task of reconciling rapid urban transformation with inclusive and sustainable adaptation remains unresolved, particularly under conditions of spatial heterogeneity and institutional centralization. hazard mitigation.







In contrast, other Southeast Asian contexts reflect distinct trajectories shaped by historical underinvestment in technical infrastructure and greater reliance on social and environmental dimensions of preparedness. In Vietnam, the persistence of structural defences such as high dikes has generated a false sense of security, often suppressing individual adaptation efforts; however, preparedness is now shifting toward more integrated approaches that emphasize risk communication, informal knowledge exchange, and psychosocial drivers of behaviour (Binh et al., 2020; Casse et al., 2015; McElwee et al., 2017; Ngo et al., 2020). Myanmar illustrates a case where environmental awareness and strong community cohesion—rather than formal systems—form the foundation of adaptive strategies, with collective memory and social capital functioning as key enablers in the absence of technical or institutional capacity (Lwin et al., 2020). In Bangladesh, while formal mechanisms such as early warning systems are gradually improving, household-level preparedness remains anchored in lived experience and social learning, particularly within rural farming communities (Azad et al., 2022; Faruk and Maharjan, 2023). Indonesia, despite a growing institutional framework, continues to struggle with fragmentation: strategies remain reactive, and the integration of local knowledge and inclusive governance into formal planning processes is still limited (Jeuken et al., 2015; Maryati et al., 2019). Taken together, these cases suggest a shared regional constraint: although technical and environmental ambition has expanded, the institutional embedding of social preparedness—particularly in terms of equity, participation, and multilevel coordination—remains partial and uneven.

• Small Islands: In countries like Fiji (Nolet, 2016), preparedness efforts unfold within highly localized social and ecological systems, where institutional capacities are often limited but experiential knowledge and community cohesion form the

backbone of adaptive responses. Rather than relying on large-scale infrastructure or data-intensive modelling, these contexts prioritize community-based adaptations grounded in long-standing interactions with the environment. Mangrove preservation, sustainable agriculture, and traditional land management practices constitute core components of environmental strategies, not merely as substitutes for technical solutions, but as culturally embedded mechanisms of hazard mitigation.

Social strategies are similarly shaped by proximity, trust, and informal governance. Community engagement is not treated as a procedural add-on but as a constitutive element of planning and response. The involvement of traditional leaders, local NGOs, and intergenerational knowledge-sharing reinforces preparedness at a scale that is responsive to both lived experience and rapidly changing climatic stressors. These processes are further supported by flexible governance arrangements that, while lacking in formal institutionalization, are often more attuned to community priorities and perceptions of risk. However, the very characteristics that enable these adaptive practices—local embeddedness, flexible authority structures, and reliance on social capital—also expose their fragility in the face of compound hazards and external dependencies. Technical measures, when present, are typically rudimentary, and financial or logistical constraints limit the capacity for broader systematization or upscaling. The challenge, therefore, is not the absence of preparedness, but the structural disconnect between localized adaptive strengths and the mechanisms required for integration into FRM frameworks.

• Africa: Strategies are largely shaped by resource scarcity, institutional fragility, and a persistent reliance on socially embedded forms of adaptation. Rather than emerging from centralized planning or technologically intensive systems, responses in countries such as Mozambique and Nigeria are grounded in the agency of communities and the mobilization of traditional knowledge. Participatory planning mechanisms—such as community surveys and localized vulnerability assessments—serve both as data collection tools and as platforms for amplifying local voices, particularly in contexts where formal governance structures are weak or unevenly distributed (Matos et al., 2023).

Social dimensions acquire prominence in these environments. In Nigeria, for example, gender-focused initiatives have positioned women as central actors in the design and operation of informal adaptation infrastructures, such as flood-resilient marketplaces and makeshift transport systems (Michael, 2024). These practices exemplify the operational role of informal networks, collective memory, and culturally grounded knowledge in sustaining adaptive capacity amid chronic underinvestment. Environmental strategies similarly reflect a bottom-up logic, with NbS adapted to context-specific needs. The integration of ecosystem-based practices—such as mangrove use, agroecological land management, and elevated market structures—is not secondary but central to flood mitigation efforts. However, such strategies are rarely supported by robust technical systems. Where technical measures do exist, they often take the form of ad hoc or temporary solution interventions (e.g., sandbags, drainage trenches), lacking the integration and predictive power of more sophisticated modelling or EWS.

This reliance on community-based and nature-oriented strategies, while effective in many localized instances, underscores a deeper systemic tension: the mismatch between the scale of emerging compound risks and the institutional and financial architectures available to address them. The result is a paradoxical condition in which preparedness is both widespread and precarious—rich in social capital yet constrained in scalability and formalization.

• North America: In the USA, hydrodynamic simulations, flood hazard mapping, and scenario-based planning have been widely institutionalized, forming the technical backbone of FRM frameworks. These tools have enabled the identification of multi-driver hazard zones and the design of resilient infrastructure systems capable of responding to a range of compound threats (Curtis et al., 2022; De Koning et al., 2019). Yet, while technical sophistication remains a defining feature, recent developments point to a gradual reconfiguration of priorities. Increasingly, flood preparedness is expanding to encompass participatory governance, equity-driven policies, and knowledge co-production with communities disproportionately affected by climate-related hazards. Stakeholder-based policy frameworks—often implemented at state and municipal levels—now seek to bridge expert-driven planning with local experiential knowledge.

Canada shows similar patterns. In coastal British Columbia, local governments adopt varying combinations of land-use regulation, construction standards, and structural measures, with decisions more strongly tied to local vulnerability profiles than to institutional capacity(Chang et al., 2020).

This shift, however, is irregular and still emergent. While initiatives exist that foreground community engagement and interdisciplinary collaboration, these are constrained by institutional inertia, political fragmentation, or inconsistencies in funding and policy continuity. As such, the integration of social and environmental dimensions into technically mature systems remains partial. This configuration reveals not a deficiency of capacity, but a strategic inflection point—one in which the challenge is less about technological innovation than about embedding that innovation within frameworks capable of recognizing and responding to the layered vulnerabilities that CCF discloses.

#### 4.2.2. Governance and multi-stakeholder collaboration in enhancing preparedness

Governance and multi-stakeholder collaboration emerge as central themes in the flood preparedness literature, reflecting the interplay between policy frameworks, community engagement, and technical advancements. These elements collectively define the capacity of communities to respond to CCF events by aligning resilience strategies with localized realities.

• Governance: Centralization and inclusivity






Governance frameworks significantly influence the success of preparedness strategies, but their effectiveness often depends on reconciling centralized efficiency with inclusive decision-making. In China, for example, centralized flood management policies, such as large-scale relocation initiatives, have shown technical efficiency but frequently lack the community engagement needed for widespread acceptance (Yu et al., 2023). This gap underscores the importance of participatory

governance models that bridge top-down planning with local needs. Moreover, fostering collaboration and information sharing across sectors is essential to enhance disaster prevention and relief efforts (Guo et al., 2023).

By contrast, projects like the Thamesmead urban regeneration initiative in the UK demonstrate the benefits of stakeholder-driven governance. By actively integrating technical expertise with local knowledge, these models foster trust, enhance public acceptance, and ensure that resilience measures align with community priorities (Coletta et al., 2024). Such approaches highlight how participatory governance can address the challenges of implementing adaptive strategies while maintaining social legitimacy.

• Multi-Stakeholder Collaboration: Strengthening collective capacity




- Collaboration among diverse actors—government agencies, NGOs, private sectors, and local communities—is critical for managing the complex risks of CCF. In China, the Sponge City Programme exemplifies the integration of NbS, such as wetlands and green infrastructure, with urban planning to mitigate flood risks while restoring hydrological cycles (Chan et al., 2024). Similarly, in Fiji, traditional leadership structures, including chiefs and religious leaders, play a vital role in disseminating preparedness messages, strengthening local resilience through cultural trust (Nolet, 2016).
- However, challenges persist in ensuring equitable collaboration. While participatory mapping in Portugal successfully integrates technical and local knowledge for FRM (Freire et al., 2016), many regions still rely heavily on top-down approaches that limit community involvement. This fact is particularly evident in urban projects, where technical solutions often overshadow the inclusion of marginalized voices, reducing the overall effectiveness of resilience strategies. For instance, while China's application of hydrodynamic models emphasizes technical precision, it often overlooks meaningful opportunities for community participation, which limits the integration of local perspectives into flood resilience strategies (Xu et al., 2024).
  - Governance and Technology: Effective preparedness

Addressing CCF risks requires a seamless integration of governance and technological advancements. Advances in hydrodynamic modelling and predictive tools, such as those used in China (Du et al., 2020; Xu et al., 2024), have significantly enhanced predictive accuracy, enabling more efficient resource allocation during flood events. However, as demonstrated by the Sponge City Programme, the full potential of these technologies is realized only when combined with governance frameworks that prioritize inclusivity and community engagement (Chan et al., 2024). Furthermore, the success of EWS depends not only on technical accuracy but also on the accessibility of information conveyed to at-risk populations. Studies from the USA highlight that clear, actionable communication is crucial for ensuring timely community responses to compound hazards (Richmond and Kunkel, 2024). Without such transparency, even the most advanced predictive models' risk being underutilized, leaving vulnerable communities exposed to preventable losses. Similarly, as observed in Italy, these tools

regularly fail to translate into actionable governance frameworks, thereby limiting their effectiveness at the community level (Sacchi et al., 2023).

The integration of participatory governance with cutting-edge technology not only enhances predictive capabilities but also fosters trust among stakeholders, ensuring resilience measures are both scientifically robust and socially relevant. This highlights the importance of hybrid approaches that balance technological precision with the lived realities of vulnerable populations, bridging the gap between technical expertise and local needs.

• Governance Challenges: Addressing fragmentation and enhancing coordination





As CCF risks grow increasingly complex, fragmented governance frameworks exacerbate vulnerabilities and undermine resilience. Figure 8 illustrates the interconnected roles of key actors identified in the literature—local governments, NGOs, research institutions, and traditional leaders—in shaping governance strategies for preparedness. However, the lack of cohesive coordination among these entities highlights a critical barrier: sectors often operate in isolation, focusing on single hazards rather than addressing the interconnected nature of compound risks (Šakić Trogrlić and Hochrainer-Stigler, 2024).

While scientific advancements, such as hydrodynamic modelling and flood forecasting, have significantly improved the understanding of compound hazards, their application in actionable governance remains limited. For example, in China, despite progress in predictive tools, these advancements are rarely integrated into community-specific strategies (Xu et al., 2024). Similarly, Mozambique's urban resilience initiatives, though infrastructure-focused, fail to achieve their full potential due to the exclusion of community participation (Matos et al., 2023). These examples underscore how fragmented governance not only limits inter-agency collaboration but also hinders the equitable allocation of resources, leaving vulnerable populations inadequately supported.

A recurring challenge lies in the failure to institutionalize cross-sectoral coordination. As represented in Figure 8, research institutions play a pivotal role in generating valuable data on compound hazards. However, without clear mechanisms to translate these insights into policy, their potential impact is diminished. This disconnect is especially evident in EWS, where technical precision often does not align with accessible, community-focused communication (Richmond and Kunkel, 2024). The resulting mismatch between technical capabilities and the needs of at-risk communities perpetuates preventable vulnerabilities.

To address these gaps, governance must evolve beyond siloed approaches and embrace systemic frameworks that incorporate multi-hazard or compound thinking into policy and practice. Collaborative models, such as China's Sponge City Programme, exemplify the benefits of aligning technical solutions with participatory governance to address interconnected and cascading risks (Chan et al., 2024). However, these remain exceptions rather than norms. Bridging the gap between science and policy requires harmonized frameworks that integrate cross-sectoral coordination and prioritize inclusive, locally grounded solutions. Such approaches must emphasize the co-production of knowledge, equitable resource distribution, and communication strategies tailored to community needs.

Several mechanisms identified in the literature could support this transition, including policy incentives that promote joint planning, shared funding schemes for inter-agency projects, and formal cooperation platforms that institutionalize collaboration among governments, civil society, and research institutions (Matczak and Hegger, 2021; Nordbeck et al., 2023). Additionally, coordinated data-sharing mechanisms—such as the exchange of historical and real-time information across institutional and spatial boundaries—can support timely communication and collective decision-making across administrative levels (Šakić Trogrlić et al., 2022). Embedding these mechanisms into preparedness strategies is essential not only to improve coordination, but to ensure that responses are inclusive, locally grounded, and operationally viable. Such approaches must prioritize the co-production of knowledge, the redistribution of decision-making power, and communication strategies tailored to community needs, moving from fragmented planning toward adaptive governance frameworks that reflect the complexity of CCF risks.

Figure 8. Governance Dimensions and Actor Interactions in Preparedness Strategies for CCF. This diagram illustrates the fragmented roles of key actors—local governments, NGOs, traditional leaders, research institutions, and communities—in shaping governance strategies for preparedness. Approaches are often siloed, focusing on individual hazards and sectors, with limited interaction across different areas and levels of governance, resulting in unclear responsibilities for compound events.

#### 5. Discussion

750

This review began with the premise that CCF presents a qualitatively distinct challenge for FRM and preparedness strategies.

By examining how preparedness is addressed in 49 studies across diverse geographic and institutional settings, we identified recurrent patterns, conceptual tensions, and operational gaps. This final section reflects on the implications of those findings, returning to the two guiding research questions.

### 5.1. RQ1: How are preparedness strategies evolving to integrate technical, environmental, and social dimensions in managing CCF risks?

The analysis shows an emerging shift from hazard-specific and sectoral approaches toward more integrative preparedness strategies. On the technical side, advances in hydrodynamic modelling, compound event simulations, and EWS are improving anticipatory capacity. However, these tools often remain siloed and dependent on limited driver combinations, typically in bivariate frameworks (e.g., rainfall + storm surge), which limit their ability to capture the full complexity of CCF. From an environmental perspective, there is growing incorporation of ecosystem-based approaches—particularly NbS—that offer 735 multifunctional benefits for flood mitigation and ecological resilience. These interventions are being increasingly recognized not only as protective measures but as integral components of adaptive preparedness planning. In terms of the social dimension, a broader acknowledgment is emerging regarding the role of community awareness, trust in authorities, and the value of local knowledge in shaping effective responses. Some studies engage with participatory approaches or co-production of knowledge, although these remain relatively limited and regularly subordinated to technical objectives. Crucially, as recent studies point 740 out, e.g., Sacchi et al. (2023), the effectiveness of EWS in CCF contexts is often compromised by the way information is interpreted and acted upon. Even when forecasts are technically robust, the multiplicity of drivers/hazards can generate confusion, leading individuals and institutions to focus on a single dominant driver while overlooking other contributing factors. This cognitive simplification, coupled with the lack of integrated communication channels across agencies, weakens the operational relevance of alerts and hampers timely decision-making.

Despite these trends, integration across dimensions remains partial. In many cases, technical solutions are prioritized, and social or environmental aspects are appended rather than embedded. Moreover, compound logic is frequently cited but rarely translated into operational frameworks capable of addressing slow-onset or cascading impacts. This suggests that while preparedness strategies are evolving, they have not yet achieved full integration across the technical, environmental, and social domains.

#### 5.2. RQ2: What is the role of governance and multi-stakeholder collaboration in enhancing CCF preparedness?

The review suggests that governance structures and multi-stakeholder collaboration play an influential—but highly uneven—role. In some countries, governance frameworks have evolved to support cross-sector coordination and participatory planning.

Initiatives such as China's Sponge City Programme and the UK's Thamesmead regeneration project illustrate how co-produced strategies and hybrid infrastructures can foster locally grounded and adaptive preparedness. These examples show the potential of inclusive governance to bridge technical and social dimensions of FRM. However, such integrative efforts remain the exception. In many cases, preparedness continues to be hampered by fragmented institutional arrangements, overlapping mandates, and limited coordination across agencies and levels of government. This misalignment weakens the capacity to operationalize compound thinking. Four cross-cutting themes emerge.






First, while centralized governance structures can facilitate technical efficiency—particularly in countries like China—they often struggle to incorporate local needs and knowledge. The absence of participatory mechanisms weakens their legitimacy and adaptability. Conversely, stakeholder-driven models, such as the Thamesmead initiative in the UK, demonstrate how inclusive governance can enhance public trust, align interventions with community priorities, and support more flexible, adaptive planning.

Second, collaboration among diverse actors—government agencies, NGOs, private sectors, and local communities—proves essential for addressing the multidimensional nature of CCF. Successful examples, such as Portugal's participatory mapping, highlight the value of integrating formal and informal systems. In other contexts, like Fiji, community-based governance and traditional authority structures play a central role in sustaining localized preparedness, even in the absence of formal institutional frameworks. However, many regions still rely heavily on top-down approaches that marginalize local perspectives, limiting the effectiveness and legitimacy of resilience strategies.

Third, technological advancements—such as hydrodynamic modelling and EWS—are enhancing predictive capacity. Yet, their effectiveness depends on the governance frameworks in which they are embedded. Where these tools are deployed without adequate community engagement or accessible communication strategies, their potential remains underutilized. This is evident in both high-capacity settings like Italy and emerging initiatives in countries like China and Mozambique.

Finally, the review underscores a persistent governance barrier: fragmented governance undermines coordination, slows down policy translation, and weakens preparedness. Despite the proliferation of actors and tools, many strategies remain siloed, focusing on individual hazards rather than interconnected drivers and hazards. Figure 8 illustrates how misalignment among key actors leads to unclear responsibilities, duplication of efforts, and missed opportunities for co-produced solutions.

In sum, while governance and multi-stakeholder collaboration are widely recognized as key elements of flood preparedness, their actual impact depends on their capacity to promote integration across sectors, support meaningful participation, and reflect the complexity of CCF hazard. Moving from isolated initiatives to broader institutional change requires embedding these principles into planning frameworks and aligning them with the realities of diverse and unequal territories.

#### 5.3. Limitations







While this review offers a comprehensive synthesis of how preparedness strategies are evolving in response to CCF risks, several limitations must be acknowledged. These limitations stem not only from the characteristics of the available literature but also from the methodological and interpretive choices made throughout the process.

First, the scope of the analysis is shaped by the selection criteria used. Although the systematic search aimed to capture a broad range of studies on CCF preparedness, the terminology surrounding compound events remains ambiguous. As a result, relevant contributions framed under alternative terms may have been overlooked. This semantic ambiguity continues to pose a challenge for delineating the contours of an evolving research area.

Second, while the screening process combined machine learning tools (ASReview, BERT) with human judgement to minimize bias and improve transparency, it remains susceptible to subjective decisions—particularly in the labelling of borderline cases and the interpretation of "compound". Furthermore, the reliance on abstracts and titles during the early stages of screening may have led to the omission of studies that substantively engage with CCF preparedness but do not make this explicit in their metadata. Although this approach was designed to pursue methodological transparency and computational scalability, it inevitably limits the depth of the review. Recent advances in artificial intelligence—particularly in Natural Language Processing (NLP) and the development of transformer-based Large Language Models (LLMs)—have shown promise in enabling full-text mining and semantic extraction from scientific publications. These tools can enhance the identification of nuanced content and latent connections that may be overlooked when relying solely on metadata. For instance, (Hill et al., 2024) showed the potential of AI-powered tools to extract targeted methodological details from full texts, while (Lieberum et al., 2025) emphasized both the opportunities and the limitations of using LLMs in evidence synthesis, noting concerns related to reproducibility, hallucinations, and prompt sensitivity. Given these challenges, the decision to rely on abstracts and titles remains methodologically justified, though future applications of AI-supported full-text analysis may offer greater depth and coverage, provided robust validation frameworks are in place.

Third, the analysis of preparedness strategies relied heavily on the content of peer-reviewed articles, many of which focus on theoretical frameworks or modelling approaches rather than grounded, empirical documentation of preparedness practices. As such, the review may underrepresent informal or practice-based knowledge, especially in low-resource settings where scientific publication may not reflect the full range of community efforts and governance dynamics.

Fourth, the review emphasizes coastal and estuarine contexts, in line with its research objective. While this focus allows for greater depth, it limits the generalizability of findings to other environments where CCF also occurs, such as inland regions or urban basins exposed to simultaneous pluvial and fluvial drivers.

Fifth, although this review aimed to reflect a balance among technical, environmental, and social dimensions, the underlying literature remains structurally skewed toward technical approaches. Social and behavioural perspectives—despite their recognized importance in shaping preparedness—are less frequently addressed in ways that allow for meaningful comparison. This imbalance may stem from systemic barriers, including funding schemes that prioritize technological innovation,

disciplinary silos, and limited availability of empirical social data. As a result, aspects such as trust, participation, and local knowledge—critical to the design and effectiveness of preparedness strategies—are often underrepresented. This gap constrains not only the integrative capacity of the review but also the potential to assess how preparedness operates in real-world, socially embedded contexts.

Finally, this study does not provide a formal meta-analysis or quantitative synthesis, as the heterogeneity of methods, definitions, and scales across studies makes such aggregation analytically problematic. Instead, the emphasis was placed on qualitative synthesis and thematic integration. While this approach enables interpretive depth, it may limit reproducibility and comparability across reviews.

#### 6. Future research and reflections

The literature reveals a field in conceptual and methodological evolution. The proliferation of compound event frameworks has widened the lens through which flooding is viewed, yet many studies stop short of embracing this complexity in actionable terms. A vast majority of the analysed studies does not incorporate behavioural insights into preparedness frameworks. This is a critical omission because individuals—and institutions—tend to simplify complex risks, often failing to account for compound dynamics. Consequently, communication, EWS, and planning efforts must be adapted to counteract these tendencies and promote a more comprehensive understanding of risk.

Operationalizing more integrative preparedness also requires facing persistent limitations in data availability, model interoperability, and transferability. Comparative analysis is hindered by heterogeneous methodologies and inconsistent definitions, particularly regarding what qualifies as "compound". While standardization may help address some of these issues, the diversity of CCF contexts demands a parallel investment in methodological pluralism and context-sensitive planning. Future research should also explore how to balance and integrate green, blue, and gray (engineered) infrastructures in ways that reflect local needs, environmental conditions, and available resources. Such integrative approaches can enhance both technical robustness and social legitimacy in preparedness strategies.

Rather than being treated merely as a phase within the DRR cycle, preparedness should be understood as a systemic and socially embedded process, as emphasized in the SFDRR. Enhancing it involves more than developing tools or protocols—it calls for inclusive mechanisms that enable those at risk to act as co-producers of their own safety. This process is shaped by power relations, timing mismatches, and epistemic hierarchies that influence whose knowledge is recognized and who holds decision-making authority. The failure to integrate community insights or redistribute decision-making power limits the transformative potential of preparedness. When local perspectives are sidelined or authority remains concentrated, meaningful change becomes unlikely. In this sense, governance fragmentation reflects not only institutional limitations but also deeper asymmetries in how risk is conceptualized and addressed.

- To move forward, several directions emerge. First, CCF preparedness must explicitly incorporate behavioural research—not only to understand individual perceptions, but to inform the design of EWS, participatory tools, and adaptive learning mechanisms. Second, operational strategies must be stress-tested against real-world constraints—such as limited data, scarce resources, and unclear mandates—particularly in under-resourced contexts. Third, governance must evolve to facilitate co-production through shared platforms, iterative learning, and both vertical and horizontal coordination.
- Finally, preparedness should be conceived as both anticipatory—by integrating uncertainty into planning—and reflexive—by allowing for continuous adjustment based on evolving conditions and knowledge. Rather than prescribing fixed solutions, it should enable adaptive coordination across sectors, institutions, and scales, while empowering communities as active agents in managing CCF risk.
- Data availability. An interactive version of Figure 7, illustrating the evolution of preparedness strategies, is available at: <a href="https://doi.org/10.5281/zenodo.15848424">https://doi.org/10.5281/zenodo.15848424</a> (Gomez et al., 2025).

Author contributions. DG: conceptualization, formal analysis, investigation, visualization, writing (original draft preparation). AS: conceptualization, methodology, writing (review and editing). MdJ: validation, writing (review and editing). All authors contributed to the interpretation of results and provided feedback on the manuscript.

Competing interests. The authors declare no competing interests.

Acknowledgements. This research was carried out within the framework of the Specialisation Certificate for the Assessment and Management of Geological and Climate-related Risk of the University of Geneva (CERG-C) (https://www.unige.ch/sciences/terre/CERG-C/). The project received financial support from the Government of Cantabria through the Fénix Programme and from GrantRTI2018-096449-B-I00, funded by MCIN/AEI/10.13039/501100011033 and by "ERDF A way of making Europe." Language editing and the generation of a limited number of illustrative images (e.g., for the graphical abstract) involved the use of generative AI tools. In addition, AI-supported tools were employed to assist the literature review process, as detailed in the Methods section. These tools were used to facilitate specific tasks and did not affect the scientific reasoning, data treatment, or interpretation of findings.

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
