# Peer review of "Review article: Rethinking Preparedness for Coastal Compound Flooding: Insights from a Systematic Review."

_EGUsphere, 2025_

## Referee Comment (RC1)

**Review report:** Rethinking Preparedness for Coastal Compound Flooding: Insights from a Systematic Review.

The review paper called "Review article: Rethinking Preparedness for Coastal Compound Flooding: Insights from a Systematic Review" aims to explore how preparedness strategies are growing to be more integrative and how governance and stakeholder collaboration enhance adaptive approaches. The paper can contribute to the literature on coastal compound flooding preparedness upon undergoing some terminology corrections and restructuring. My main concern relates to the unclear differentiation between drivers and hazards, which results in considerable and recurring confusion in the manuscript. This issue may have introduced uncertainty into the search and filtering protocol in the literature review.

I commend the authors on their research efforts. Please find below the review comments aimed at enhancing the clarity and impact of the paper.

**Abstract**

- The reader should be provided with the number of reviewed studies, as well as some details about them (time period, spatial scope, for instance).

- The findings noted at lines 13-20 should be reframed to be more coherent, as they currently miss a red thread.

- The aim noted in the Abstract differs from the one stated in the Introduction ("This review explores how preparedness strategies are evolving to integrate technical, environmental, and social dimensions while evaluating the role of governance and collaboration in enhancing adaptive approaches."). It is unclear to me what type of preparedness (against what) do the authors investigate. Adaptive approaches to what? The framing here is rather vague.

- Line 9: the drivers listed here are actually hazards (except for river discharge, which is not even a driver).

Line 10: sectoral silos is an unclear term.

**Graphical abstract**

- What do the authors mean by "a primary hazard that triggers ripple effects"? Isn't this phrasing redundant with cascading impacts?

- I recommend reordering the outcomes based on their importance. The current order seems rather random.

**Introduction**

- Line 26 ("The greatest risks from a changing climate may not come from individual impacts") contains a logical error considering the terminology of impact and disaster risk as proposed by UNDRR. Please revise the definitions of these terms and modify accordingly.

- Line 27: What are intersectional vulnerabilities? The term is rather confusing in this context.

- The authors should clarify from the beginning what do they call a driver. Is this term interchangeably used with hazard, as shown at lines 29 and 40? I do not recommend using them as synonyms, but to always clearly specify if they refer to a hazard or the driver (of what, of CF as a hazard)?

- I recommend rephrasing the research questions in a clearer way (1st questions – dimensions of what?; 2nd question is rather blurred and it is not a question per se). Reaching the Methodology section, I see that the questions are clearly formulated here, but they should also be written like this in the Introduction.

- The authors do not specify how gaining answers to the 2 proposed research questions will contribute to the development of adaptive frameworks: "By addressing these critical issues, this study seeks to contribute to the development of adaptive frameworks that strengthen resilience and enhance preparedness in the face of complex and evolving CF risks".

**Methodology**

- Line 163-164: "By examining these integrations, we assess how well they address the complex and compounding risks associated with multiple flood drivers." – what does this assessment involve? Is there a clear framework for assessing the degree to which the listed elements address the CF risk involving multiple flood drivers?

- Upon reading section 3.1., it is unclear to me the time period targeted by this literature review.

- Please check Table 1 for typos.

- Why were studies on tsunamis ("disasters such as tsunamis and earthquakes, which were beyond the scope of this work") beyond the scope of this work, if they related to preparedness for such hazards?

- Is the exclusion process described at lines 205-210 mainstream for literature reviews in flood preparedness? Is this method sound enough to correctly identify the papers that did not align with the objectives of the review? To me, the procedure sounds rather inconsistent and relevant studies may have been removed from the pool. Perhaps list this as a methodological limitation.

- What were the relevant and irrelevant records that served as the foundation for training the first machine learning model?

**Results**

- Line 261: Social Sciences should also be written with capital letters.

- Figure 3: I recommend replacing this polar chart with another type of representation. Such charts are harder to read, and the same information can be conveyed in more classical and clearer ways.

- I would like to see a more extensive explanation of this point: "This notable growth in scientific attention after 2012 aligns with a broader shift in natural hazard research paradigms, particularly following significant developments in climate risk frameworks."

- "The surge in publications, particularly after 2015, coincides with the growing recognition of the need for integrated approaches that address the complexities of compound flooding and other interconnected hazards" – this can be linked with the Sendai Framework.

- I advise the authors to draw another timeline figure identifying the key trends discussed in section 4.1. The 0x is temporal, and the rest includes the emergence of key trends (start and end points). This figure can help the reader identify the diversification tendencies and the introduction of new terms (e.g., compound events, compound effects, multi-hazard) more readily ,and it would make a valuable addition to the already rich and high-quality material in this paper. The figure can also include a similar design for the details in sections 4.2.

- Table 2: there is no need to separately provide the year. The reference alone looks neater. I also think the caption of the table should provide some details on the methodology of eliciting the key topics.

Figure 6: I recommend replacing the pie chart with another type of chart. It is well known that pie charts are misleading and harder to read for most people. Also, on the bar chart, please replace the Count on 0y with a more appropriate label.

- Figure 7: Please improve the readability of the text in this picture. Providing some contrasting background for the text would be beneficial to the reader.

**Conclusions**

- What is understood here by systemic vulnerability and systemic risk? The authors should clearly define these terms (also used in the Conclusions and throughout the text) in the introductory part.

- Line 551: complex interactions of what?

**Additional comments**

- I recommend adding a dedicated Reflections section to consolidate the paper's key contributions. It can be placed after Results. This section should include clear answers to the two research questions and compare insights on CF preparedness with preparedness for other hazards influenced by climate change (in terms of frequency, intensity). By critically discussing these findings, this section would serve as the intellectual "heart" of the paper.

---

## Referee Comment (RC2)

Journal Paper Reviewed: Gomez Rave, D. V., Scolobig, A., and del Jesus, M.: Review article: Rethinking Preparedness for Coastal Compound Flooding: Insights from a Systematic Review, EGUsphere [preprint], https://doi.org/10.5194/egusphere-2025-262, 2025.

Journal: Natural Hazards and Earth System Sciences (NHESS)

**1. General Comments**

The preprint titled "Rethinking Preparedness for Coastal Compound Flooding (CF): Insights from a Systematic Review" provides an insightful examination of strategies for managing compound flooding (CF) risks based on a structured literature review. The authors address the need to consider the multiple aspects of compound flooding risk including solutions that combine technical, environmental, and social dimensions, as well as the critical role of governance and multi-stakeholder collaboration.

Strengths of the paper include illustrating the evolution of CF research—from hazard-specific technical approaches to more holistic frameworks, while offering a critical lens on the shortcomings of current governance structures and participatory strategies. However, clear definitions and use of flood risk and disaster management terms are lacking. As a result, the paper framing lacks clarity and accurate use of terms which are well defined in the scientific literature. In particular the use of the term "preparedness" seems to be applied to more than just the preparedness phase of the disaster management cycle but rather flood risk and adaptation more broadly. The definition and use of this term, which also appears in the title should be clear.

Additionally, the integration of case studies based on the most relevant papers (e.g., China's Sponge City Program, the Netherlands' Delta Plan) adds depth to the analysis. However, the paper would be improved with a more explicit discussion of the limitations of the reviewed studies, particularly in terms of data availability and transferability. In addition, a more cohesive discussion section which distills and structures the findings for future research and practical applications would improve the impact of the paper.

Overall, this preprint makes a valuable contribution to the literature on disaster risk reduction and climate adaptation. With major revisions, it has the potential to contribute meaningfully to the scientific literature on compound flood risk management.

**2. Specific Comments**

Framing

The flood risk and disaster management terms used are not defined and therefore the framing is unclear. For example, the stated focus is on preparedness, however, Blue & Green Infrastructure for example is more connected to adaptation or mitigation of hazards rather than preparedness.

I would encourage the authors to clearly define the risk equation they are using (hazard, exposure, vulnerability) and the disaster management cycle (preparedness, event/disaster, response, recovery, mitigation/adaptation) and cite relevant literature (for example Koks et. al., 2015).

It seems that the intended focus is more risk reduction strategies across the disaster risk management cycle for compound floods in coastal areas. The conclusion does not mention the coastal context at all which is supposed to be the focus of this study. The findings should connect back to the focus area and provide an outlook related to that context.

Methodology

The use of ASReview and BERT model is innovative and the steps are clearly explained. It is mentioned that the ASReview model is based on their textual features to prevent author name and citation network biases. However, other biases can exist while using machine learning screening (e.g., keyword selection, training data). If these were addressed or at least identified this could be added.

Also, a clearer explanation of how subjective decisions were minimized would enhance reproducibility. The PRISMA flowchart (Figure 2) is clear but it would be helpful to add more detail on how the 49 articles were assessed to align with the research questions.

Thematic Gaps

While the paper acknowledges underrepresented themes like governance and behavioral dimensions, it stops short of proposing specific pathways for addressing them. The conclusion hints at the need for co-production and hybrid strategies but could be more explicit in offering guidance for implementation, especially in varied socio-political contexts.

The discussion on fragmented governance (Figure 8) and the challenges is valuable but could be strengthened by referencing mechanisms known in the literature to improve cross-sectoral coordination such as policy incentives or joint funding programs.

In addition, it may be helpful to look at the broader literature on several points. For example, it is mentioned that nature-based solutions are rather implemented in middle income countries but there are many projects that incorporate NbS in all income levels. For example, green dike and making room for the river projects in the Netherlands and Mangrove restoration in many countries globally. In addition, Indigenous Knowledge is integrated into preparedness and adaptation in high income countries (e.g. New Zealand, Australia, Canada). Perhaps rather than classification based on income, the approaches could be referenced (eg: NbS, Indigenous Knowledge) with some reference to regional strengths and challenges.

Figures and Visualizations

Figures are generally helpful and relevant, however, the design of some visuals (e.g., Figures 6 and 7) are dense and would benefit from simplification or improved legends to enhance readability.

Integration of Social Dimensions

The paper identifies a gap in social science research within the reviewed literature (Figure 3). It would be helpful to discuss why this gap exists and how it might impact the effectiveness of preparedness strategies. For example, are there biases in funding or publication trends that favor technical over social studies? Are there challenges with data collection or availability?

Regional Disparities:

The analysis of high-, middle-, and low-income countries is useful but somewhat generalized. More nuanced comparisons (e.g., within middle-income countries) could reveal additional insights about contextual factors influencing preparedness.

3.  Technical Corrections

| TC # | Line # | |
|---|---|---|
| 1 | Throughout | Consider rephrasing long or complex sentences to improve readability, especially in the methods and discussion sections. |

| 2 | Throughout | With the term "compound flooding" you sometimes abbreviate as "CF" and sometimes don't. This should be standardized throughout the paper. |
|---|---|---|
| 3 | Line 8 to 22 | Abstract should mention the methods used and highlight key results. |
| 4 | Lines 55, 103, and 297 | Sendai Framework is introduced twice Line 55 and 103. Phrasing about Sendai on Line 297 also sounds like it wasn't mentioned before. Connect these references. |
| 5 | Line 155-157 | You mention "storm surges, river flooding, and extreme rainfall" create heightened risk. These are all related to the hazard component of risk. If you only focus on hazard then this should be clearly stated. However, you later specify that you are looking at how strategies integrate technical, environmental, and social dimensions which suggests you look at drivers related to multiple components of risk. Be clear about how you define a use risk and hazard terminology. |
| 6 | Line 194 | What is meant by "reflecting the diverse strategies employed to address flood risk and preparedness". Flood risk is something exists due to a combination of hazard, exposure, and vulnerability. Risk reduction measures can target each of these components. Actions for risk reduction can also be framed as targeting particular phases of the disaster management cycle including preparedness. |
| 7 | Line 224 | Researcher-In-The-Loop (RITL) is mentioned in full twice with the abbreviation. Just include this once and then use the abbreviation. |
| 8 | Lines 273 - 276 | The two sentences starting with "In parallel, it is important to acknowledge…" are a bit awkward. Consider rephrasing. |
| 9 | Table 2 | Clarify that the years listed are publication years, and ensure consistent formatting across entries.
Possible double entry error for year with (Chan et al., 2023)
The years and references with years are also somewhat redundant. Consider reformatting and perhaps only include the reference. |
| 10 | Line 307 | Reference to the literature would fit here at the end of the sentence. |
| 11 | Line 246-247 | "…this nuanced aspect of preparedness…" It is unclear what this refers to. |
| 12 | Line 544 | You mention "cognitive bias" here for the first time in the conclusions. While cognitive simplification is mentioned earlier in the article with regards to CF "cognitive bias" is not clearly addressed in the article. Be clear about what you mean in the conclusions and/or reference how you use the term earlier in the article. |
| 13 | Line 422 | Typo with extra period |
| 14 | Lines 281 to 284 | Provide reference and quantification of increase in publications on natural hazard research. |

---

## Author Comment (AC1)

**Reply to the Reviewer**

*Re: Manuscript ID Preprint egusphere-2025-262*
*"Review article: Rethinking Preparedness for Coastal Compound Flooding: Insights from a Systematic Review"*
*Dina Vanessa Gómez-Rave, Anna Scolobig, Manuel del Jesus*
*Natural Hazards and Earth System Sciences - NHESS*

**Response to Reviewer 1**
We would like to thank Professor Cosmina Albulescu for providing a rigorous and insightful review that helped us identify key areas for improving the focus and conceptual consistency of the manuscript. Each of her comments has been carefully considered, and we describe below how they will be addressed in the revised version.
* * *
**Review Report**

**The review paper called *"Rethinking Preparedness for Coastal Compound Flooding (CF): Insights from a Systematic Review"* aims to explore how preparedness strategies are growing to be more integrative and how governance and stakeholder collaboration enhance adaptive approaches. The paper can contribute to the literature on coastal compound flooding preparedness upon undergoing some terminology corrections and restructuring. My main concern relates to the unclear differentiation between drivers and hazards, which results in considerable and recurring confusion in the manuscript. This issue may have introduced uncertainty into the search and filtering protocol in the literature review.**
**I commend the authors on their research efforts. Please find below the review comments aimed at enhancing the clarity and impact of the paper.**

We acknowledge that some inconsistencies in wording may have led to confusion between "drivers" and "hazards." However, the review is conceptually grounded in the typology proposed by Zscheischler et al. [2020], where compound events arise from combinations of multiple climate drivers and/or hazards. In the case of coastal compound flooding, this distinction is operationalized through the interaction of physical drivers (e.g., rainfall, storm surge) that give rise to a hazard (e.g., flooding).

Our study focuses specifically on climatic drivers, given their increasing relevance in the context of climate change and their central role in compound event analysis. As further detailed in a later response, this focus also justifies the exclusion of events such as tsunamis, whose non-climatic origin, limited predictability, and distinct generation mechanisms fall outside the analytical scope and methodological foundation of this review.

Following this framework, our search and filtering protocol was specifically designed to capture the literature addressing the interacting nature of these drivers and their contribution to flood hazard in coastal environments. To improve clarity and avoid ambiguity, we will revise the manuscript to explicitly define these terms early in the Introduction, citing the underlying literature that informs this distinction. Terminology will also be carefully reviewed throughout the manuscript to ensure consistency with this established typology.

**Abstract**

- **The reader should be provided with the number of reviewed studies, as well as some details about them (time period, spatial scope, for instance).**

- **The findings noted at lines 13–20 should be reframed to be more coherent, as they currently miss a red thread.**

- **The aim noted in the Abstract differs from the one stated in the Introduction ("This review explores how preparedness strategies are evolving to integrate technical, environmental, and social dimensions while evaluating the role of governance and collaboration in enhancing adaptive approaches."). It is unclear to me what type of preparedness (against what) do the authors investigate. Adaptive approaches to what? The framing here is rather vague.**

- **Line 9: the drivers listed here are actually hazards (except for river discharge, which is not even a driver).**

- **Line 10: sectoral silos is an unclear term.**

The abstract has been revised to include the number of reviewed studies, their spatial and thematic focus, and to ensure consistency with the aim stated in the Introduction. It defines the type of preparedness that we are reviewing—strategies targeting compound flooding in estuarine and coastal areas— and clarifies that adaptive approaches refer to measures addressing interacting climate-related drivers. The distinction between drivers and hazards has been clarified to follow current compound event literature. The findings have been reorganized to improve coherence and narrative flow, and the term "sectoral silos" has been replaced with "limited cross-sectoral coordination", as referred to in the literature to describe institutional fragmentation that limits integrated responses (e.g., Oseland [2019]; Sakic-Trogrlic and Hochrainer-Stigler [2024]).

The revised version, incorporating these adjustments, now reads as follows:

*Tackling the growing risks of Compound Flooding (CF) requires transformative preparedness strategies, particularly in estuarine and coastal regions, where interaction of climatic drivers such as storm surges, rainfall, and river discharge exacerbates impacts. Despite progress, fragmented governance, limited cross-sectoral coordination, and insufficient integration of scientific insights hinder effective responses. This systematic review draws on 49 studies covering estuarine and coastal regions globally to explore how preparedness strategies are evolving to integrate technical, environmental, and social dimensions while evaluating the role of governance and collaboration in enhancing adaptive approaches. Hybrid early warning systems combining statistical and hydrodynamic models with real-time data are critical for forecast accuracy and timely decision-making. Similarly, balanced implementation of green, blue, and gray infrastructure provides sustainable responses, with nature-based solutions complementing traditional engineering. Our results also show that strengthening governance and communication is essential to improve preparedness. Involving communities in land-use planning, building regulations, and communication ensures measures are both actionable and context-specific. Incorporating psychological and behavioral insights into preparedness frameworks helps translate awareness into proactive, effective actions. Enhanced coordination across sectors and levels of government is also vital to addressing the systemic nature of CF risks, moving beyond siloed, single-hazard responses.*

**Graphical abstract**

- **What do the authors mean by "a primary hazard that triggers ripple effects"? Isn't this phrasing redundant with cascading impacts?**

- **I recommend reordering the outcomes based on their importance. The current order seems rather random.**

The expression "a primary hazard that triggers ripple effects" has been revised to avoid redundancy with the concept of cascading impacts and to improve terminological precision.

While we acknowledge the suggestion regarding the ordering of outcomes, we consider these elements to represent interdependent pillars of preparedness rather than a strict hierarchy of importance. Nevertheless, minor adjustments have been made to improve narrative flow and thematic coherence in the graphical abstract, as shown below:

[Figure]

**Introduction**

- **Line 26 ("The greatest risks from a changing climate may not come from individual impacts") contains a logical error considering the terminology of impact and disaster risk as proposed by UNDRR. Please revise the definitions of these terms and modify accordingly.**

We have revised the sentence to align with the terminology used in the UNDRR framework and in recent compound event literature (e.g., Zscheischler et al. [2020]), which emphasizes the interaction of multiple drivers and/or hazards.

The updated version reads:

*"The greatest risks from a changing climate may not arise from single hazards, but from the interaction of multiple climatic drivers or hazards that challenge existing response capacities."*

- **Line 27: What are intersectional vulnerabilities? The term is rather confusing in this context.**

The use of the term "intersectional vulnerabilities" follows the framing adopted by Simpson et al. [2023], where risks from a changing climate are described as emerging from the nexus between compound hazards, exposures, and overlapping forms of vulnerability. In this context, "intersectional" does not refer to a specific theory or framework, but rather captures the way in which multiple social, economic, and geopolitical factors combine to shape differentiated levels of risk. We use the term to emphasize that vulnerability to compound flooding is not evenly distributed, but is often amplified for those at the intersection of disadvantage. Given that the phrasing is directly aligned with the cited source and captures a key dimension of risk complexity, we consider its inclusion appropriate and meaningful in this context.

- **The authors should clarify from the beginning what they call a driver. Is this term interchangeably used with hazard, as shown at lines 29 and 40? I do not recommend using them as synonyms, but to always clearly specify if they refer to a hazard or the driver (of what, of CF as a hazard)?**

As Prof. Abulescu rightly points out, it is essential to distinguish between drivers and hazards. Following the terminology proposed by Zscheischler et al. [2020], we will clarify in the revised manuscript that our work focuses on climatic *drivers*, as the underlying meteorological, hydrological, or oceanographic conditions—such as heavy rainfall, storm surge, and river discharge—that may act independently or in

combination to give rise to a hazard. *Hazards*, in contrast, are the resulting events with the potential to cause damage—such as compound flooding in coastal and estuarine areas, where the multiple climatic drivers interact and exceed natural or built drainage capacities.

Coastal areas encompass a wide range of geomorphological settings, including beaches, cliffs, estuaries, and deltas—zones where both marine and terrestrial processes converge. In such environments, the occurrence of flooding is not solely driven by marine conditions (e.g., storm surge or high tide), but also by land-based contributions such as river discharge.

We will include this distinction early in the manuscript and ensure consistent terminology throughout.

- **I recommend rephrasing the research questions in a clearer way (1st question – dimensions of what?; 2nd question is rather blurred and it is not a question per se). Reaching the Methodology section, I see that the questions are clearly formulated here, but they should also be written like this in the Introduction.**

We appreciate Prof. Abulescu's observation regarding the formulation of the research questions. In the revised version, we will reword the questions in the Introduction to reflect the more precise and structured version already presented in the Methodology. The first question will explicitly state which dimensions are being addressed, and the second will be reshaped into a proper interrogative form with a more defined scope. These changes will contribute to a more consistent and well-structured narrative throughout the manuscript.

The updated paragraph in the Introduction will read as follows:

*This paper conducts a systematic literature review to critically examine how climate risk management practices are evolving to address the intricate challenges of compound flooding in coastal areas—regions where the interplay of vulnerabilities and flood drivers increases risks. The analysis centers on two guiding research questions:*

- i) *How are preparedness strategies evolving to integrate technical, environmental, and social dimensions in managing compound flood risks?*
- ii) *What is the role of governance and multi-stakeholder collaboration in enhancing flood preparedness?*

*By addressing these key issues, this study seeks to contribute to the development of adaptive frameworks that strengthen resilience and improve preparedness in the face of complex and evolving CF risks.*

- **The authors do not specify how gaining answers to the 2 proposed research questions will contribute to the development of adaptive frameworks: "By addressing these critical issues, this study seeks to contribute to the development of adaptive frameworks that strengthen resilience and enhance preparedness in the face of complex and evolving CF risks".**

To address this point, the revised manuscript will include a brief paragraph specifying how the two research questions contribute to the development of adaptive frameworks. The analysis sheds light on the operational, institutional, and socio-environmental conditions that influence preparedness efforts across coastal contexts. By examining how these dimensions are addressed (RQ1) and how governance and collaboration mechanisms support them (RQ2), the study identifies gaps, enabling factors, and practices that inform the evolution of preparedness. Instead of proposing a universal model, the contribution lies in highlighting the key elements that shape adaptive strategies and in offering insights to support more flexible, integrative, and context-sensitive approaches to compound flood risk.

**Methodology**

- **Line 163–164: "By examining these integrations, we assess how well they address the complex and compounding risks associated with multiple flood drivers." – what does this assessment involve? Is there a clear framework for assessing the degree to which the listed elements address the CF risk involving multiple flood drivers?**

The revised manuscript will specify that the assessment is interpretive in nature, based on a close examination of how each study approaches the integration of technical, environmental, and social components in relation to compound flood risk. Rather than applying a predefined framework, the analysis builds on recurring patterns and tensions across the literature to explore how integration is framed, where it is advanced, and where it remains limited or implicit.

- **Upon reading section 3.1., it is unclear to me the time period targeted by this literature review.**

  We acknowledge that the time period covered by the review was not clearly stated. The search did not impose a restriction on the starting year; all records available in the Web of Science (WoS) database up to September 2024 were considered. In the revised manuscript, this will be made explicit in Section 3.1. The paragraph will now read:

  *"... To identify relevant studies, we carried out a systematic search in the Web of Science (WoS) database, applying a multi-layered strategy aimed at capturing research related to preparedness for compound flooding in coastal areas, with a particular focus on community resilience and risk management. This approach was informed by previous reviews on similar topics (Kuhlicke et al., 2023; Sun et al., 2024). No lower date limit was applied; all records available in the WoS database up to September 2024 were included in the review. The search was organized ..."*

- **Please check Table 1 for typos.**

  Table 1 will be carefully reviewed, and any typos or inconsistencies will be corrected in the revised version.

- **Why were studies on tsunamis ("disasters such as tsunamis and earthquakes, which were beyond the scope of this work") beyond the scope of this work, if they related to preparedness for such hazards?**

  The updated document will make explicit that studies on tsunamis were excluded because they are not aligned with the analytical scope of this review. As noted by Hendry [2021], tsunamis are of geophysical origin and do not result from the interaction of climate-driven processes, which is the core focus of compound flooding events considered here. Their exclusion is not based on relevance to preparedness in general, but on the need for conceptual consistency: the review targets flood risks arising from the conjunction of meteorological, hydrological, and oceanographic drivers linked to climate variability and change. Including tsunamis would compromise the coherence of the framework and the comparability of the selected studies.

- **Is the exclusion process described at lines 205–210 mainstream for literature reviews in flood preparedness? Is this method sound enough to correctly identify the papers that did not align with the objectives of the review? To me, the procedure sounds rather inconsistent and relevant studies may have been removed from the pool. Perhaps list this as a methodological limitation.**

  We will provide a more explicit and robust justification of the exclusion process. This procedure combined topic modeling with expert judgment to refine the initial pool of articles. Using the Python-based tool Litstudy for trend visualization, we generated word clouds to identify prominent terms across the dataset. This helped us pinpoint thematic clusters that, despite matching the search strings, were conceptually misaligned with the scope of the review. For example, terms such as "oil" and "surfactant" were associated with studies on petroleum extraction, while others like "seed bank" and "germination" pertained to plant physiology research in coastal ecosystems. Upon further inspection, these studies were excluded as they did not address compound flooding or preparedness strategies.

  To ensure transparency, the revised manuscript will include the complete refined search query along with the terms that were excluded. The following terms were removed from the search in the Topics (TS) field due to their lack of relevance to the review's objectives:

*earthquake, species, tsunami, seed bank, habitat, germination, mangrove, irrigation, lake, soil, bank, food insecurity, organic matter, trees, sediment, dam, ice jam, drought, groundwater, energy*

The exclusion process will be acknowledged as a methodological limitation, as it involved some interpretive judgment. However, we believe that tools like Litstudy are extremely useful when dealing with large amounts of bibliographic data. These tools help identify patterns and inconsistencies that might not be easily detected through manual screening alone. This approach ensures that the analysis stays focused on the key topics of compound flooding risks and preparedness strategies.

- **What were the relevant and irrelevant records that served as the foundation for training the first machine learning model?**

The relevant and irrelevant records used to train the first machine learning model in ASReview were initially identified through random selection, as provided by the tool itself. For the training phase, 34 records were manually labeled by the researchers, with classification based solely on the abstracts. Only the abstracts were presented, while the titles were withheld, ensuring that the classification process was based on more comprehensive information rather than potentially misleading or incomplete titles.

Once the initial set of records was labeled, the machine learning model began automatically selecting and presenting additional records for classification. As the researcher labeled more records, the model iteratively improved its ability to identify relevant studies, progressively focusing on the most pertinent literature. In total, approximately 40% of the articles retrieved from Web of Science were screened through this process.

**Results**

- **Line 261: Social Sciences should also be written with capital letters.**

Thank you for pointing that out. *Social Sciences* will be corrected to uppercase in line 261.

- **Figure 3: I recommend replacing this polar chart with another type of representation. Such charts are harder to read, and the same information can be conveyed in more classical and clearer ways.**

The figure has been revised to replace the polar chart with a clearer representation. The updated version, shown below, presents the same information in a more straightforward and readable format to enhance clarity.

- **I would like to see a more extensive explanation of this point: "This notable growth in scientific attention after 2012 aligns with a broader shift in natural hazard research paradigms, particularly following significant developments in climate risk frameworks."**

In response to the reviewer's request, the revised manuscript will expand on how scientific focus shifted after 2012, particularly with the introduction of "compound events" in the IPCC's 2012 SREX report. This shift reflects broader changes in natural hazard research and climate risk frameworks.

- **"The surge in publications, particularly after 2015, coincides with the growing recognition of the need for integrated approaches that address the complexities of compound flooding and other interconnected hazards" – this can be linked with the Sendai Framework.**

Thank you for your comment. The surge in publications since 2015 is indeed linked to the growing recognition of the need for integrated approaches to complex hazards like compound flooding. This fact aligns with the Sendai Framework for Disaster Risk Reduction (2015-2030), which emphasizes multi-hazard strategies. We will explicitly highlight this connection in the revised manuscript.

- **I advise the authors to draw another timeline figure identifying the key trends discussed in section 4.1. The 0x is temporal, and the rest includes the emergence of key trends (start and end points). This figure can help the reader identify the diversification tendencies and the**

introduction of new terms (e.g., compound events, compound effects, multi-hazard) more readily, and it would make a valuable addition to the already rich and high-quality material in this paper. The figure can also include a similar design for the details in sections 4.2.

A timeline figure will be included to show how key concepts discussed in section 4.1 have emerged and evolved over time, including shifts in terminology and focus. A second panel will reflect the main research directions outlined in section 4.2.

- **Table 2: there is no need to separately provide the year. The reference alone looks neater. I also think the caption of the table should provide some details on the methodology of eliciting the key topics.**

The inclusion of the year in the reference format was an oversight and will be corrected to reflect the proper reference style. Additionally, the caption of Table 2 will be revised to include a more detailed description of the methodology used to elicit the key topics.

- **Figure 6: I recommend replacing the pie chart with another type of chart. It is well known that pie charts are misleading and harder to read for most people. Also, on the bar chart, please replace the Count on Y with a more appropriate label.**

We will replace the pie chart with a more suitable visualization. Additionally, the Y-axis label on the bar chart will be updated to more accurately reflect the data.

- **Figure 7: Please improve the readability of the text in this picture. Providing some contrasting background for the text would be beneficial to the reader.**

In response to the comment, we will also increase the font size and adjust the text formatting to further enhance readability. Additionally, we will make sure that the text is clear and well-contrasted against the background.

**Conclusions**

- **What is understood here by systemic vulnerability and systemic risk? The authors should clearly define these terms (also used in the Conclusions and throughout the text) in the introductory part.**

The concepts of systemic vulnerability and systemic risk will be explicitly defined in the revised introduction. In line with Weir et al. [2024], systemic vulnerability refers to the susceptibility of interconnected systems to disruption under external stress, driven by interdependencies among their components, while systemic risk captures the potential for such disruptions to propagate across systems, triggering cascading and large-scale impacts. Armaș et al. [2025] further conceptualize systemic vulnerability as the enduring core of vulnerability that persists across time and space, regardless of mitigation efforts, and has the potential to reinforce future impacts or obstruct adaptation processes. These definitions will be used consistently throughout the manuscript to clarify their relevance in multi-hazard and dynamic risk contexts.

- **Line 551: complex interactions of what?**

The sentence has been revised to specify the meaning of "complex interactions," now framed as the interplay among physical processes, socio-institutional dynamics, and evolving conditions within coupled human–natural systems.

Revised sentence (line 551):

*Cascading impacts, non-linear climate feedback, and systemic vulnerabilities demand adaptive frameworks capable of anticipating complex interactions among physical systems and socio-institutional structures.*

**Additional comments**

**I recommend adding a dedicated Reflections section to consolidate the paper's key contributions. It can be placed after Results. This section should include clear answers to the two research questions and compare insights on CF preparedness with preparedness for other hazards influenced by climate change (in terms of frequency, intensity). By critically discussing these findings, this section would serve as the intellectual "heart" of the paper.**

As suggested, the revised manuscript will include a Reflections section after the Results. This section will revisit the two guiding research questions and draw together the main insights from the review, while also acknowledging the uncertainties, tensions, and limitations that persist in the field of compound flood preparedness. Rather than offering definitive answers, it will propose a set of considerations that may help inform future research and practice—such as the importance of addressing governance fragmentation, incorporating behavioral dimensions, and exploring ways to make integrative strategies more actionable across diverse contexts. The aim is to close the manuscript with a structured synthesis that consolidates the findings and suggests potential directions for further work.

**References**

Iuliana Armaș, Andra-Cosmina Albulescu, and Daniela Dobre. Towards managing vulnerability. A new model of systemic vulnerability. *International Journal of Disaster Risk Reduction*, 121:105378, 2025. ISSN 2212-4209. doi: https://doi.org/10.1016/j.ijdrr.2025.105378. URL `https://www.sciencedirect.com/science/article/pii/S221242092500202X`.

Alistair Hendry. *Compound Flooding in the UK: Past, Present and Future Co-occurring Extreme Flooding Hazard Sources*. PhD thesis, University of Southampton, October 2021. URL `https://eprints.soton.ac.uk/452419/`.

Stina Ellevseth Oseland. Breaking silos: can cities break down institutional barriers in climate planning? *Journal of Environmental Policy & Planning*, 21(4):345–357, July 2019. ISSN 1523-908X, 1522-7200. doi: 10.1080/1523908X.2019.1623657. URL `https://www.tandfonline.com/doi/full/10.1080/1523908X.2019.1623657`.

Robert Sakic-Trogrlic and Stefan Hochrainer-Stigler. Navigating multi-hazard risks: building resilience in a systemic risk landscape, October 2024. URL `https://iiasa.ac.at/blog/oct-2024/navigating-multi-hazard-risks-building-resilience-in-systemic-risk-landscape`.

Nicholas P. Simpson, Portia Adade Williams, Katharine J. Mach, Lea Berrang-Ford, Robbert Biesbroek, Marjolijn Haasnoot, Alcade C. Segnon, Donovan Campbell, Justice Issah Musah-Surugu, Elphin Tom Joe, Abraham Marshall Nunbogu, Salma Sabour, Andreas L.S. Meyer, Talbot M. Andrews, Chandni Singh, A.R. Siders, Judy Lawrence, Maarten Van Aalst, and Christopher H. Trisos. Adaptation to compound climate risks: A systematic global stocktake. *iScience*, 26(2):105926, February 2023. ISSN 25890042. doi: 10.1016/j.isci.2023.105926. URL `https://linkinghub.elsevier.com/retrieve/pii/S2589004223000032`.

Alana M. Weir, Thomas M. Wilson, Mark S. Bebbington, Craig Campbell-Smart, James H. Williams, and Roger Fairclough. Quantifying systemic vulnerability of interdependent critical infrastructure networks: A case study for volcanic hazards. *International Journal of Disaster Risk Reduction*, 114:104997, November 2024. ISSN 22124209. doi: 10.1016/j.ijdrr.2024.104997. URL `https://linkinghub.elsevier.com/retrieve/pii/S2212420924007593`.

Jakob Zscheischler, Olivia Martius, Seth Westra, Emanuele Bevacqua, Colin Raymond, Radley M. Horton, Bart Van Den Hurk, Amir AghaKouchak, Aglaé Jézéquel, Miguel D. Mahecha, Douglas Maraun, Alexandre M. Ramos, Nina N. Ridder, Wim Thiery, and Edoardo Vignotto. A typology of compound weather and climate events. *Nature Reviews Earth & Environment*, 1(7):333–347, June 2020. ISSN 2662-138X. doi: 10.1038/s43017-020-0060-z. URL `https://www.nature.com/articles/s43017-020-0060-z`.

---

## Author Comment (AC2)

**Reply to the Reviewer**

*Re: Manuscript ID Preprint egusphere-2025-262*
*"Review article: Rethinking Preparedness for Coastal Compound Flooding: Insights from a Systematic Review"*
*Dina Vanessa Gómez-Rave, Anna Scolobig, Manuel del Jesus*
*Natural Hazards and Earth System Sciences - NHESS*

**Response to Reviewer 2**

We thank the reviewer for the thoughtful and detailed feedback on our manuscript. Below we provide point-by-point responses addressing each comment.
* * *
**1. General Comments**

**The preprint titled *"Rethinking Preparedness for Coastal Compound Flooding (CF): Insights from a Systematic Review"* provides an insightful examination of strategies for managing compound flooding (CF) risks based on a structured literature review. The authors address the need to consider the multiple aspects of compound flooding risk including solutions that combine technical, environmental, and social dimensions, as well as the critical role of governance and multi-stakeholder collaboration.**

**Strengths of the paper include illustrating the evolution of CF research—from hazard-specific technical approaches to more holistic frameworks, while offering a critical lens on the shortcomings of current governance structures and participatory strategies. However, clear definitions and use of flood risk and disaster management terms are lacking. As a result, the paper framing lacks clarity and accurate use of terms which are well defined in the scientific literature. In particular the use of the term "preparedness" seems to be applied to more than just the preparedness phase of the disaster management cycle but rather flood risk and adaptation more broadly. The definition and use of this term, which also appears in the title should be clear.**

We understand the concern regarding the use of the term preparedness. In the revised manuscript, we will specify that our focus is on strategies aligned with the preparedness phase, while acknowledging their intersections with broader risk reduction and adaptation efforts. Key terms—such as drivers, hazard and risk will be used consistently and defined where relevant to ensure conceptual clarity throughout the paper.

**Additionally, the integration of case studies based on the most relevant papers (e.g., China's Sponge City Program, the Netherlands' Delta Plan) adds depth to the analysis. However, the paper would be improved with a more explicit discussion of the limitations of the reviewed studies, particularly in terms of data availability and transferability. In addition, a more cohesive discussion section which distills and structures the findings for future research and practical applications would improve the impact of the paper.**

**Overall, this preprint makes a valuable contribution to the literature on disaster risk reduction and climate adaptation. With major revisions, it has the potential to contribute meaningfully to the scientific literature on compound flood risk management.**

We will revise the discussion section to more clearly acknowledge limitations in the reviewed studies, particularly those related to data availability and context-specific applicability. The section will also be restructured to better synthesize the main insights and articulate how these inform future research directions and practical implementation.

**2. Specific Comments**

**Framing**

**The flood risk and disaster management terms used are not defined and therefore the framing is unclear. For example, the stated focus is on preparedness, however, Blue & Green Infrastructure for example is more connected to adaptation or mitigation of hazards rather than preparedness.**

**I would encourage the authors to clearly define the risk equation they are using (hazard, exposure, vulnerability) and the disaster management cycle (preparedness, event/disaster, response, recovery, mitigation/adaptation) and cite relevant literature (for example Koks et al., 2015).**

**It seems that the intended focus is more risk reduction strategies across the disaster risk management cycle for compound floods in coastal areas. The conclusion does not mention the coastal context at all which is supposed to be the focus of this study. The findings should connect back to the focus area and provide an outlook related to that context.**

We appreciate the reviewer's insights regarding conceptual framing and terminology. We agree that a clearer articulation of the underlying risk framework would strengthen the manuscript. While our focus is on preparedness, we recognize that many of the strategies reviewed—particularly nature-based solutions—operate at the interface of preparedness, mitigation, and adaptation. Consequently, we will situate our approach within the broader disaster risk management cycle and explicitly introduce the risk components (hazard, exposure, vulnerability) as part of the conceptual background. We will also revise the conclusion to clearly reconnect the findings with the specific context of compound flooding in coastal areas, which frames the scope and relevance of the review.

**Methodology**

**The use of ASReview and BERT model is innovative and the steps are clearly explained. It is mentioned that the ASReview model is based on their textual features to prevent author name and citation network biases. However, other biases can exist while using machine learning screening (e.g., keyword selection, training data). If these were addressed or at least identified this could be added.**

**Also, a clearer explanation of how subjective decisions were minimized would enhance reproducibility. The PRISMA flowchart (Figure 2) is clear but it would be helpful to add more detail on how the 49 articles were assessed to align with the research questions.**

In the revised manuscript, we will briefly address additional sources of potential bias in machine learning-assisted screening—such as keyword selection and training data—while noting that, despite mitigation efforts (e.g., through ASReview's design), some degree of bias may remain. This will be acknowledged as a limitation. We will also clarify how subjective decisions were reduced through predefined inclusion criteria and iterative calibration. Finally, we will expand on how the 49 selected articles were assessed and linked to the research questions to enhance transparency and reproducibility.

**Thematic Gaps**

**While the paper acknowledges underrepresented themes like governance and behavioral dimensions, it stops short of proposing specific pathways for addressing them. The conclusion hints at the need for co-production and hybrid strategies but could be more explicit in offering guidance for implementation, especially in varied socio-political contexts.**

**The discussion on fragmented governance (Figure 8) and the challenges is valuable but could be strengthened by referencing mechanisms known in the literature to improve cross-sectoral coordination such as policy incentives or joint funding programs.**

In addition, it may be helpful to look at the broader literature on several points. For example, it is mentioned that nature-based solutions are rather implemented in middle income countries but there are many projects that incorporate NbS in all income levels. For example, green dike and making room for the river projects in the Netherlands and Mangrove restoration in many countries globally. In addition, Indigenous Knowledge is integrated into preparedness and adaptation in high income countries (e.g. New Zealand, Australia, Canada). Perhaps rather than classification based on income, the approaches could be referenced (eg: NbS, Indigenous Knowledge) with some reference to regional strengths and challenges.

The discussion section will be strengthened to more clearly address the governance dimension by referencing mechanisms that can help overcome coordination barriers. Furthermore, while nature-based approaches are already included in the manuscript, for instance, the Sponge City Program, we will make their contribution more explicit and provide examples from a wider range of contexts, including high-income countries. Although the income-based framing has been useful to structure the comparison, we will complement it by drawing attention to regional strengths, enabling conditions, and context-specific challenges that shape implementation strategies such as co-production and Indigenous Knowledge.

**Figures and Visualizations**

**Figures are generally helpful and relevant, however, the design of some visuals (e.g., Figures 6 and 7) are dense and would benefit from simplification or improved legends to enhance readability.**

Figures 6 and 7 will be revised to enhance readability by simplifying the visual structure, increasing font sizes where necessary, adjusting color contrasts (particularly background and node/link tones), and improving the clarity of legends. These modifications aim to make the figures more accessible without compromising the message of the information conveyed, which would require to keep some of the complexity.

**Integration of Social Dimensions**

**The paper identifies a gap in social science research within the reviewed literature (Figure 3). It would be helpful to discuss why this gap exists and how it might impact the effectiveness of preparedness strategies. For example, are there biases in funding or publication trends that favor technical over social studies? Are there challenges with data collection or availability?**

We will expand the discussion around Figure 3 to briefly reflect on the limited presence of social science perspectives in the reviewed literature. This will include a concise paragraph identifying possible reasons and how these limitations may influence the development and implementation of preparedness strategies. The updated paragraph would read as follows:

*The limited integration of social science perspectives in preparedness research on compound flooding can be rooted in both methodological challenges and disciplinary boundaries. Historically, flood risk has been approached through technical and hazard-centered frameworks, with a strong focus on hydrometeorological drivers, modeling, and structural responses, leaving less room for analysis of how society understands, experiences and responds to flood events [Lechowska, 2022]. Furthermore, inconsistent terminology and conceptual fuzziness exacerbate the disconnect. Varied definitions of multi-hazard and compound events have led to "fragmentation of the literature," with overlapping terms sowing redundancy and confusion, and hindering collaboration across fields[Serinaldi et al., 2022, Green et al., 2024]. In addition, methodological challenges, such as limited data availability, lack of standardization, and the context-specific nature of social indicators, also restrict their inclusion [Girons Lopez et al., 2017, Vanelli et al., 2022]. A more integrated approach is needed to inform preparedness strategies that reflect both the physical dynamics of compound flooding and the ways in which societies experience and respond to them.*

**Regional Disparities**

**The analysis of high-, middle-, and low-income countries is useful but somewhat generalized. More nuanced comparisons (e.g., within middle-income countries) could reveal additional insights about contextual factors influencing preparedness.**

The revised manuscript will strengthen the comparative framework by incorporating concise, literature-based insights that illustrate how preparedness strategies vary not only between but also within income groups. This includes acknowledging relevant contextual differences where clearly documented.

**3. Technical Corrections**

| TC# | Line # | Comment and Response |
|---|---|---|
| 1 | Throughout | **Consider rephrasing long or complex sentences to improve readability, especially in the methods and discussion sections.** We will revise the manuscript to streamline overly long or complex sentences, focusing in particular on the Methods and Discussion sections. These edits aim to improve readability and strengthen the overall flow of the text. |
| 2 | Throughout | **With the term "compound flooding" you sometimes abbreviate as "CF" and sometimes don't. This should be standardized throughout the paper.** The use of the term "compound flooding" and its abbreviation "CF" will be standardized throughout the manuscript to ensure consistency and avoid confusion. |
| 3 | Line 8 to 22 | **Abstract should mention the methods used and highlight key results.** We will update the abstract to include the methods used and summarize key findings in line with the paper's focus. |
| 4 | Lines 55, 103, and 297 | **Sendai Framework is introduced twice (Lines 55 and 103). Phrasing about Sendai on Line 297 also sounds like it wasn't mentioned before. Connect these references.** References to the Sendai Framework will be streamlined and connected to avoid redundancy. |
| 5 | Lines 155–157 | **You mention "storm surges, river flooding, and extreme rainfall" create heightened risk.These are all related to the hazard component of risk. If you only focus on hazard then this should be clearly stated. However, you later specify that you are looking at how strategies integrate technical, environmental, and social dimensions which suggests you look at drivers related to multiple components of risk. Be clear about how you define a use risk and hazard terminology.** The manuscript focuses on the hazard component of risk, particularly in relation to climate-related drivers. Although technical, environmental, and social dimensions are discussed in relation to preparedness strategies, the analysis does not evaluate risk in terms of exposure or vulnerability. To avoid confusion, key terms such as hazard, drivers, and risk will be defined at the beginning of the manuscript, and the scope of the analysis will be clarified early on. |

*(Continued on next page)*

| TC# | Line # | Comment and Response |
|---|---|---|
| 6 | Line 194 | **What is meant by "reflecting the diverse strategies employed to address flood risk and preparedness". Flood risk is something exists due to a combination of hazard, exposure, and vulnerability. Risk reduction measures can target each of these components. Actions for risk reduction can also be framed as targeting particular phases of the disaster management cycle including preparedness.**
The phrasing will be adjusted to emphasize that the review examines strategies targeting the hazard component of flood risk, with a particular focus on measures implemented during the preparedness phase of the disaster management cycle. |
| 7 | Line 224 | **Researcher-In-The-Loop (RITL) is mentioned in full twice with the abbreviation. Just include this once and then use the abbreviation.**
We will ensure that Researcher-In-The-Loop (RITL) is introduced only once, and the abbreviation is used consistently thereafter. |
| 8 | Lines 273–276 | **The two sentences starting with "In parallel, it is important to acknowledge. . . " are a bit awkward. Consider rephrasing.**
The phrasing of both sentences will be revised to enhance readability and avoid repetition. |
| 9 | Table 2 | **Clarify that the years listed are publication years, and ensure consistent formatting across entries.**

**Possible double entry error for year with (Chan et al., 2023).**

**The years and references with years are also somewhat redundant.**

**Consider reformatting and perhaps only include the reference.**

We'll adjust formatting inconsistencies and remove duplicate entries. |
| 10 | Line 307 | **Reference to the literature would fit here at the end of the sentence.**
The reference will be included at the end of the sentence, as suggested. |
| 11 | Lines 246–247 | **". . . this nuanced aspect of preparedness. . . " It is unclear what this refers to.**
We will revise the sentence to explicitly indicate that the reference is to preparedness actions focused on hazard-related drivers within the disaster risk management cycle. |
| 12 | Line 544 | **You mention "cognitive bias" here for the first time in the conclusions. While cognitive simplification is mentioned earlier in the article with regards to CF "cognitive bias" is not clearly addressed in the article. Be clear about what you mean in the conclusions and/or reference how you use the term earlier in the article.**
We agree that the mention of "cognitive bias" in the conclusion should be more closely aligned with the terminology used in the main text. In the revised manuscript, we will ensure consistency and briefly explain that this refers to mental shortcuts—such as focusing on familiar or isolated drivers—that influence how compound flood risks are perceived and addressed. |

| TC# | Line # | Comment and Response |
|---|---|---|
| 13 | Line 422 | **Typo with extra period.**
Extra period will be removed. |
| 14 | Lines 281 to 284 | **Provide reference and quantification of increase in publications on natural hazard research.**
Relevant publication trends will be briefly referenced and supported with data to substantiate the observation. |

**References**

Marc Girons Lopez, Giuliano Di Baldassarre, and Jan Seibert. Impact of social preparedness on flood early warning systems. *Water Resources Research*, 53(1):522–534, 2017.

Joshua Green, Ivan D. Haigh, Niall Quinn, Jeff Neal, Thomas Wahl, Melissa Wood, Dirk Eilander, Marleen de Ruiter, Philip Ward, and Paula Camus. A Comprehensive Review of Coastal Compound Flooding Literature, 2024. URL `https://arxiv.org/abs/2404.01321`. Version Number: 1.

Ewa Lechowska. Approaches in research on flood risk perception and their importance in flood risk management: a review. *Natural Hazards*, 111(3):2343–2378, 2022.

Francesco Serinaldi, Federico Lombardo, and Chris G Kilsby. Testing tests before testing data: an untold tale of compound events and binary dependence. *Stochastic Environmental Research and Risk Assessment*, 36(5):1373–1395, 2022.

Franciele Maria Vanelli, Masato Kobiyama, and Mariana Madruga de Brito. To which extent are socio-hydrology studies truly integrative? the case of natural hazards and disaster research. *Hydrology and Earth System Sciences*, 26(8):2301–2317, 2022.

---

## Author Response (AR1)

**Reply to the Reviewer**

Re: Manuscript ID Preprint egusphere-2025-262

"Review article: Rethinking Preparedness for Coastal Compound Flooding: Insights from a Systematic Review"

Dina Vanessa Gómez-Rave, Anna Scolobig, Manuel del Jesus Natural Hazards and Earth System Sciences - NHESS

**Response to Reviewer 1**

We would like to thank Professor Cosmina Albulescu for providing a rigorous and insightful review that helped us identify key areas for improving the focus and conceptual consistency of the manuscript. Each of her comments has been carefully considered, and we describe below how they will be addressed in the revised version.

**Review Report**

The review paper called "Rethinking Preparedness for Coastal Compound Flooding (CF): Insights from a Systematic Review" aims to explore how preparedness strategies are growing to be more integrative and how governance and stakeholder collaboration enhance adaptive approaches. The paper can contribute to the literature on coastal compound flooding preparedness upon undergoing some terminology corrections and restructuring. My main concern relates to the unclear differentiation between drivers and hazards, which results in considerable and recurring confusion in the manuscript. This issue may have introduced uncertainty into the search and filtering protocol in the literature review.

I commend the authors on their research efforts. Please find below the review comments aimed at enhancing the clarity and impact of the paper.

We acknowledge that some inconsistencies in wording may have previously led to confusion between "drivers" and "hazards." However, the revised manuscript now clearly articulates the conceptual distinction, grounded in the typology proposed by Zscheischler et al. [2020], where compound events arise from combinations of multiple climate drivers and/or hazards. In the case of coastal compound flooding, this distinction is now explicitly operationalized as the interaction of physical drivers (e.g., rainfall, storm surge) that give rise to a hazard (e.g., flooding).

Our revised study maintains its focus on climatic drivers, reflecting their growing relevance in the context of climate change and their central role in compound event analysis. As noted later in the manuscript, this scope also justifies the exclusion of events such as tsunamis, which are non-climatic in origin and fall outside the methodological foundation of the review.

The search and filtering protocol itself has not been modified; rather, the revised manuscript clarifies the underlying rationale behind it. To improve clarity and avoid ambiguity, terminology is now introduced early in the Introduction (from Line 32 onward), with appropriate citations, and has been carefully reviewed for consistency throughout the manuscript.

As now stated in the Introduction, the following passage defines the distinction more clearly:

"...Drivers encompass processes, variables, and phenomena in the climate and weather domain—such as precipitation, temperature, river flow, coastal water levels, atmospheric humidity, soil moisture or wind speed—that may operate across multiple spatial and temporal scales. Hazards, in contrast, denote the immediate physical phenomena—such as floods, heatwaves, or landslides—that may trigger impacts when they coincide with exposure—the presence of people, infrastructure, or ecosystems in harm's way—and vulnerability—their propensity to suffer damage or loss due to limited capacity to anticipate, cope with, or recover from the event (Koks et al., 2015; Zscheischler et al., 2020; IPCC, 2023). The interplay among these components can result in compound risks, arising from single extremes or co-occurring events affecting critical

systems or sectors (IPCC, 2023). This conceptual framing provides a basis for analysing how interacting climatic conditions can evolve into complex events—and how their consequences ripple through interconnected systems."

**Abstract**

- The reader should be provided with the number of reviewed studies, as well as some details about them (time period, spatial scope, for instance).
- The findings noted at lines 13–20 should be reframed to be more coherent, as they currently miss a red thread.
- The aim noted in the Abstract differs from the one stated in the Introduction ("This review explores how preparedness strategies are evolving to integrate technical, environmental, and social dimensions while evaluating the role of governance and collaboration in enhancing adaptive approaches."). It is unclear to me what type of preparedness (against what) do the authors investigate. Adaptive approaches to what? The framing here is rather vague.
- Line 9: the drivers listed here are actually hazards (except for river discharge, which is not even a driver).
- Line 10: sectoral silos is an unclear term.

The abstract has been revised to include the number of reviewed studies, their spatial and thematic focus, and to ensure consistency with the aim stated in the Introduction. It now specifies the type of preparedness addressed—strategies targeting compound flooding in estuarine and coastal areas—and explains that adaptive approaches refer to measures dealing with interacting climate-related drivers. The distinction between drivers and hazards has been refined to reflect current terminology in compound event literature. The findings have been reorganized to enhance coherence and narrative flow, and the term "sectoral silos" has been replaced with "limited cross-sectoral coordination," as commonly used in the literature to describe institutional fragmentation that hampers integrated responses (e.g., Oseland [2019]; Sakic Trogrlic and [Hochrainer-Stigler] [2024]).

The revised version, incorporating these adjustments, now reads as follows:

Tackling the growing risks of Compound Flooding (CF) requires transformative preparedness strategies, particularly in estuarine and coastal regions, where the interaction of drivers such as storm surges, rainfall, and river discharge exacerbates impacts. Despite progress, fragmented governance, weak cross-sectoral coordination, and the limited integration of scientific insights hinder effective responses. This systematic review draws on 49 studies to explore how preparedness strategies are evolving to integrate technical, environmental, and social dimensions while evaluating the role of governance and collaboration in enhancing adaptive approaches. Hybrid Early Warning Systems combining statistical and hydrodynamic models with real-time data are critical for forecast accuracy and timely decision-making. Similarly, balanced implementation of green, blue, and gray infrastructure provides sustainable responses, with Nature-based Solutions complementing traditional engineering. Our results also show that strengthening governance and communication is essential to improve preparedness. Involving communities in land-use planning, building regulations, and communication ensures that measures are both actionable and context-specific. Incorporating psychological and behavioural data into preparedness frameworks and models helps strengthening the link between awareness and behaviours. Enhanced coordination across sectors and levels of government is also vital to addressing the systemic nature of CF risks, moving beyond siloed, single-hazard responses.

**Graphical abstract**

• What do the authors mean by "a primary hazard that triggers ripple effects"? Isn't this phrasing redundant with cascading impacts?

July 10, 2025 2 of 9

• I recommend reordering the outcomes based on their importance. The current order seems rather random.

The expression "a primary hazard that triggers ripple effects" has been revised to avoid redundancy with the concept of cascading impacts and to improve terminological precision.

While we acknowledge the suggestion regarding the ordering of outcomes, we view these elements as interdependent pillars of preparedness rather than components of a strict hierarchy. Nonetheless, minor adjustments have been made to enhance the narrative flow and thematic coherence in the graphical abstract, as shown below:

Figure 1: Graphical Abstract. Modified to reflect reviewer feedback.

**Introduction**

• Line 26 ("The greatest risks from a changing climate may not come from individual impacts") contains a logical error considering the terminology of impact and disaster risk as proposed by UNDRR. Please revise the definitions of these terms and modify accordingly.

We have revised the sentence to align with the terminology used in the UNDRR framework and in recent compound event literature (e.g., Zscheischler et al. [2020]), which emphasizes the interaction of multiple drivers and/or hazards.

The updated version reads (Line 25):

"The greatest risks from a changing climate may not arise from single hazards, but from the interaction of multiple climatic drivers and/or hazards that intersect with diverse forms of exposure, intersectional socio-economic and geopolitical vulnerabilities, and multiple types of human response—often exceeding existing response capacities Simpson et al. [2023]"

July 10, 2025 3 of 9

• Line 27: What are intersectional vulnerabilities? The term is rather confusing in this context.

The use of the term "intersectional vulnerabilities" follows the framing adopted by Simpson et al. [2023], where risks from a changing climate are described as emerging from the nexus between compound hazards, exposures, and overlapping forms of vulnerability. In this context, "intersectional" does not refer to a specific theory or framework, but rather captures the way in which multiple social, economic, and geopolitical factors combine to shape differentiated levels of risk. We use the term to emphasize that vulnerability to compound flooding is not evenly distributed, but it is often amplified for those at the intersection of disadvantage. To improve clarity, the revised manuscript now makes these forms of vulnerability explicit, while retaining the original terminology, which remains in line with the cited source and conveys a key dimension of risk complexity (see revision for Line 26 for context).

• The authors should clarify from the beginning what they call a driver. Is this term interchangeably used with hazard, as shown at lines 29 and 40? I do not recommend using them as synonyms, but to always clearly specify if they refer to a hazard or the driver (of what, of CF as a hazard)?

As Prof. Abulescu rightly points out, it is essential to distinguish between drivers and hazards. Following the terminology proposed by Zscheischler et al. [2020], we clarified in the revised manuscript that our work focuses on climatic drivers, as the underlying meteorological, hydrological, or oceanographic conditions—such as heavy rainfall, storm surge, and river discharge—that may act independently or in combination to give rise to a hazard. Hazards, in contrast, are the resulting events with the potential to cause damage—such as compound flooding in coastal and estuarine areas, where the multiple climatic drivers interact and exceed natural or built drainage capacities.

Coastal areas encompass a wide range of geomorphological settings, including beaches, cliffs, estuaries, and deltas—zones where both marine and terrestrial processes converge. In such environments, the occurrence of flooding is not solely driven by marine conditions (e.g., storm surge or high tide), but also by land-based contributions such as river discharge.

The distinction is now introduced early in the manuscript (from Line 32 onward), and terminology has been revised to ensure consistency throughout. As mentioned in our general response to the *Review Report*, the revised version incorporates a formal definition of drivers, hazards, and their interplay within compound risk settings.

• I recommend rephrasing the research questions in a clearer way (1st question – dimensions of what?; 2nd question is rather blurred and it is not a question per se). Reaching the Methodology section, I see that the questions are clearly formulated here, but they should also be written like this in the Introduction.

We appreciate this observation regarding the formulation of the research questions. In the revised version, we reworded the questions in the Introduction to align with the more precise and structured version already presented in the Methodology. The first question now explicitly states which dimensions are being addressed, while the second has been reshaped into a proper interrogative form with a more defined scope. These changes contribute to a more consistent and well-structured narrative throughout the manuscript.

The updated paragraph in the Introduction now reads as follows (Line 104):

"This study presents a systematic literature review that critically examines how FRM practices are evolving to address the intricate challenges of CF in coastal areas—regions where the interplay of vulnerabilities and flood drivers increases risks. The analysis centers on two guiding research questions:

- i) i. (RQ1) How are preparedness strategies evolving to integrate technical, environmental, and social dimensions in managing CF risks?
- ii) ii. (RQ2) What is the role of governance and multi-stakeholder collaboration in enhancing flood preparedness?"
- The authors do not specify how gaining answers to the 2 proposed research questions will contribute to the development of adaptive frameworks: "By addressing these critical issues,

July 10, 2025 4 of 9

this study seeks to contribute to the development of adaptive frameworks that strengthen resilience and enhance preparedness in the face of complex and evolving CF risks".

This point has been addressed in the revised manuscript through a new paragraph that outlines how the two research questions contribute to the development of adaptive preparedness frameworks. The text emphasizes the conditions shaping preparedness and identifies key levers for more flexible, context-sensitive responses.

The revised paragraph now reads as follows (Line 110):

"By addressing these questions, the study advances the development of more effective preparedness frameworks by analysing how strategies are being reshaped in response to CF risks across diverse coastal contexts (RQ1), and by improving understanding of the role of governance and collaboration in these processes (RQ2). This approach offers a grounded understanding of the conditions that enable or hinder anticipatory action, not as abstract goals, but as practices embedded in specific institutional and socio-environmental settings. Rather than proposing prescriptive solutions, the paper identifies key levers and recurring patterns that can inform more flexible, integrative, and context-sensitive responses. In doing so, it helps bridge the gap between conceptual debates and the operational realities of managing climate-related threats in increasingly complex risk landscapes."

**Methodology**

• Line 163–164: "By examining these integrations, we assess how well they address the complex and compounding risks associated with multiple flood drivers." – what does this assessment involve? Is there a clear framework for assessing the degree to which the listed elements address the CF risk involving multiple flood drivers?

The updated version now specifies that the assessment is interpretive, identifying recurring patterns and tensions in how integration is framed and how it responds to the complexity introduced by multiple interacting drivers, rather than relying on a predefined framework.

The sentence now reads as follows (Line 193):

- ".... It examines how this integration is framed and how it responds to the complexity introduced by multiple interacting drivers. Instead of evaluating these strategies against a predefined framework, the analysis identifies recurring patterns and tensions within the broader context of FRM."
- Upon reading section 3.1., it is unclear to me the time period targeted by this literature review.

We acknowledge that the time period covered by the review was not clearly stated. The search did not impose a restriction on the starting year; all records available in the Web of Science (WoS) database up to September 2024 were considered. This clarification has been added to Section 3.1. The paragraph reads (Line 208):

- "... No start date limit was applied; all records available in the WoS database up to September 2024 were included in the review  $\dots$ "
- Please check Table 1 for typos.

Table 1 has been carefully reviewed, and all typos or inconsistencies have been corrected in the revised version.

• Why were studies on tsunamis ("disasters such as tsunamis and earthquakes, which were beyond the scope of this work") beyond the scope of this work, if they related to preparedness for such hazards?

The revised manuscript explicitly states that research on tsunamis were excluded because they fall outside the analytical scope of this review. As noted by Hendry, Alistair [2021], tsunamis are of geophysical origin

July 10, 2025 5 of 9

and do not result from the interaction of climate-driven processes, which is the core focus of CF events considered here. Their exclusion is not based on relevance to preparedness in general, but on the need for conceptual consistency: the review targets flood risks arising from the conjunction of meteorological, hydrological, and oceanographic drivers linked to climate variability and change. Including tsunamis would compromise the coherence of the framework and the comparability of the selected studies.

This clarification has been added to the revised manuscript as follows (Line240):

"Beyond the dominant themes aligned with flood preparedness, the word cloud also revealed peripheral clusters related to ecological studies—particularly those focused on seed banks, germination processes, and plant propagation—as well as hazards of tectonic origin, such as earthquakes and tsunamis. While thematically adjacent, these topics fall outside the scope of climate-related flood dynamics (Hendry, 2021). Our focus is on CF events arising from the interaction of meteorological, hydrological, and oceanographic drivers under climate variability and change, in coastal settings."

• Is the exclusion process described at lines 205–210 mainstream for literature reviews in flood preparedness? Is this method sound enough to correctly identify the papers that did not align with the objectives of the review? To me, the procedure sounds rather inconsistent and relevant studies may have been removed from the pool. Perhaps list this as a methodological limitation.

The exclusion process is now more explicitly justified and methodologically detailed. It combined topic modeling with expert judgment to refine the initial pool of articles. Using the Python-based tool Litstudy for trend visualization, we generated word clouds to identify prominent terms across the dataset. This strategy helped pinpoint thematic clusters that, despite matching the search strings, were conceptually misaligned with the scope of the review. For example, terms such as "oil" and "surfactant" were associated with studies on petroleum extraction, while others like "seed bank" and "germination" pertained to plant physiology research in coastal ecosystems. Upon further inspection, these terms were excluded as they did not address CF or preparedness strategies.

To ensure transparency, we now include the complete refined search query along with the list of terms excluded from the Topics (TS) field due to their lack of relevance to the review's objectives, as cited (Line 246):

"The following keywords were removed from the search in the Topic (TS) field: earthquake, species, tsunami, seed bank, habitat, germination, mangrove, irrigation, lake, soil, bank, food insecurity, organic matter, trees, sediment, dam, ice jam, drought, groundwater, energy."

The exclusion process is acknowledged as a methodological limitation, as it involved some interpretive judgment. This is always the case for scoping and systematic literature reviews. Moreover, we argue that tools like Litstudy are highly effective when managing large bibliographic datasets, helping to identify thematic inconsistencies that may not be easily detected through manual screening alone. This approach ensured that the analysis remained focused on the central themes of CF risk and preparedness strategies.

• What were the relevant and irrelevant records that served as the foundation for training the first machine learning model?

The relevant and irrelevant records used to train the first machine learning model in ASReview were initially identified through random selection, as built into the tool. For the training phase, 34 abstracts were manually labeled by the researchers. Only abstracts were shown—titles were intentionally withheld to ensure that classification was based on substantive content rather than potentially misleading or overly general titles.

Following this initial labeling, the model began suggesting additional texts for review based on active learning. As more abstracts were classified, the algorithm progressively improved its ability to distinguish relevant studies, enabling the screening process to focus on the most promising publications. Overall, approximately 40% of the records retrieved from WoS were screened through this iterative process.

July 10, 2025 6 of 9

**Results**

- Line 261: Social Sciences should also be written with capital letters.
  - Thank you for pointing that out. Social Sciences has been corrected to uppercase in Line 315.
- Figure 3: I recommend replacing this polar chart with another type of representation. Such charts are harder to read, and the same information can be conveyed in more classical and clearer ways.
  - The figure has been revised to replace the polar chart with a more accessible representation. The updated version presents the same information in a more straightforward and readable format to improve interpretability.
- I would like to see a more extensive explanation of this point: "This notable growth in scientific attention after 2012 aligns with a broader shift in natural hazard research paradigms, particularly following significant developments in climate risk frameworks."
  - As requested, this point has been elaborated to note that the post-2012 increase in scientific attention coincides with the introduction of *compound events* in the IPCC's SREX report, reflecting a broader shift in climate risk and hazard research paradigms.
  - This idea is expressed in the manuscript as follows (Line 356):
  - "Consistent with these trends, the post-2012 period is characterised not only by a quantitative expansion in CF and preparedness research, but also by a gradual diversification of its conceptual and methodological landscape. This growth aligns with a broader reconfiguration of natural hazard studies, catalysed by the formal introduction of compound events in the IPCC's SREX report (IPCC, 2012)..."
- "The surge in publications, particularly after 2015, coincides with the growing recognition of the need for integrated approaches that address the complexities of compound flooding and other interconnected hazards" this can be linked with the Sendai Framework.
  - Thank you for your comment. The link between the post-2015 surge in publications and the growing recognition of integrated approaches to compound hazards has been made explicit in the revised manuscript. This is captured in the revised text, which notes that (Line 359):
  - "...A notable consolidation of this trend is evident after 2015, coinciding with the adoption of the SFDRR, which marked a strategic shift from disaster management to disaster risk management. By prioritising anticipatory action, early warning, and systemic resilience, Sendai advanced a multi-hazard and risk-informed approach that aligns closely with the emerging discourse on CF. This convergence between policy and scientific agendas likely contributed to the increased academic focus on CF and preparedness as interdependent concerns..."
- I advise the authors to draw another timeline figure identifying the key trends discussed in section 4.1. The 0x is temporal, and the rest includes the emergence of key trends (start and end points). This figure can help the reader identify the diversification tendencies and the introduction of new terms (e.g., compound events, compound effects, multi-hazard) more readily, and it would make a valuable addition to the already rich and high-quality material in this paper. The figure can also include a similar design for the details in sections 4.2.
  - Thank you for the suggestion. Taking this advice into account, the figure has been revised to better reflect the emergence and evolution of key concepts discussed in Section 4.1, including shifts in terminology and thematic focus.
- Table 2: there is no need to separately provide the year. The reference alone looks neater. I also think the caption of the table should provide some details on the methodology of eliciting the key topics.
  - The inclusion of the year in the reference format has been corrected to follow the appropriate style. The caption of Table 2 has also been revised to include a brief description of the methodology used to elicit the key topics.

July 10, 2025 7 of 9

• Figure 6: I recommend replacing the pie chart with another type of chart. It is well known that pie charts are misleading and harder to read for most people. Also, on the bar chart, please replace the Count on Y with a more appropriate label.

The pie chart in Figure 6 has been replaced with a more suitable visualization. Additionally, the Y-axis label in the bar chart has been updated to *Number of Studies* to more accurately reflect the data.

• Figure 7: Please improve the readability of the text in this picture. Providing some contrasting background for the text would be beneficial to the reader.

The readability of Figure 7 has been improved by increasing the font size and adjusting the text formatting. The text is now clearly contrasted against the background to ensure better legibility.

**Conclusions**

• What is understood here by systemic vulnerability and systemic risk? The authors should clearly define these terms (also used in the Conclusions and throughout the text) in the introductory part.

Definitions of systemic vulnerability and systemic risk have been added to the Introduction (Line 46):

"At a more structural level, the concepts of systemic vulnerability and systemic risk offer a complementary lens. Systemic vulnerability refers to the susceptibility of interdependent systems—such as infrastructure networks, governance structures, or social services—to suffer disruption under external stress, due to the cascading effects that arise from their internal linkages (Weir et al., 2024). Systemic risk, in turn, captures the potential for these disruptions to propagate across sectors and scales, resulting in widespread and often unforeseen consequences (Armaş et al., 2025). This can further exacerbate systemic vulnerability as a persistent condition that can amplify future impacts or obstruct adaptive responses, even in the presence of mitigation efforts. Such a perspective situates compound risk within the broader dynamics of interdependence, where systemic conditions shape not only the onset of these impacts but their amplification and persistence."

• Line 551: complex interactions of what?

The original sentence—"Cascading impacts, non-linear climate feedback, and systemic vulnerabilities demand adaptive frameworks capable of anticipating complex interactions."—referred to the interplay among physical processes, socio-institutional dynamics, and evolving conditions within coupled human—natural systems. This formulation was removed in the revised version and replaced with a more explicit and distributed discussion of these interactions in the Section 7 (Future Research and reflections). The revised text now unpacks these dynamics through concrete examples, emphasizing how institutional fragmentation, behavioural responses, and technical constraints interact in shaping CF preparedness.

**Additional comments**

I recommend adding a dedicated Reflections section to consolidate the paper's key contributions. It can be placed after Results. This section should include clear answers to the two research questions and compare insights on CF preparedness with preparedness for other hazards influenced by climate change (in terms of frequency, intensity). By critically discussing these findings, this section would serve as the intellectual "heart" of the paper.

As suggested, Sections 5 (Discussion) and 7 (Future Research and Reflections) have been added. These sections revisit the two research questions and consolidate the main findings of the review, highlighting persistent challenges such as governance fragmentation, limited integration of behavioural dimensions, and the gap between conceptual frameworks and operational practice. They also outline implications for future research, including the need to develop more context-sensitive, participatory, and actionable preparedness strategies.

**References**

- Hendry, Alistair. Compound Flooding in the UK: Past, Present and Future Co-occurring Extreme Flooding Hazard Sources. PhD thesis, University of Southampton, 2021.
- Stina Ellevseth Oseland. Breaking silos: can cities break down institutional barriers in climate planning? Journal of Environmental Policy & Planning, 21(4):345-357, July 2019. ISSN 1523-908X, 1522-7200. doi: 10.1080/1523908X.2019.1623657. URL https://www.tandfonline.com/doi/full/10.1080/1523908X. 2019.1623657.
- Robert Sakic Trogrlic and Stefan Hochrainer-Stigler. Navigating multi-hazard risks: building resilience in a systemic risk landscape, October 2024. URL <a href="https://iiasa.ac.at/blog/oct-2024/">https://iiasa.ac.at/blog/oct-2024/</a> navigating-multi-hazard-risks-building-resilience-in-systemic-risk-landscape.
- Nicholas P. Simpson, Portia Adade Williams, Katharine J. Mach, Lea Berrang-Ford, Robbert Biesbroek, Marjolijn Haasnoot, Alcade C. Segnon, Donovan Campbell, Justice Issah Musah-Surugu, Elphin Tom Joe, Abraham Marshall Nunbogu, Salma Sabour, Andreas L.S. Meyer, Talbot M. Andrews, Chandni Singh, A.R. Siders, Judy Lawrence, Maarten Van Aalst, and Christopher H. Trisos. Adaptation to compound climate risks: A systematic global stocktake. *iScience*, 26(2):105926, February 2023. ISSN 25890042. doi: 10. 1016/j.isci.2023.105926. URL https://linkinghub.elsevier.com/retrieve/pii/S2589004223000032.
- Jakob Zscheischler, Olivia Martius, Seth Westra, Emanuele Bevacqua, Colin Raymond, Radley M. Horton, Bart Van Den Hurk, Amir AghaKouchak, Aglaé Jézéquel, Miguel D. Mahecha, Douglas Maraun, Alexandre M. Ramos, Nina N. Ridder, Wim Thiery, and Edoardo Vignotto. A typology of compound weather and climate events. Nature Reviews Earth & Environment, 1(7):333–347, June 2020. ISSN 2662-138X. doi: 10.1038/s43017-020-0060-z. URL https://www.nature.com/articles/s43017-020-0060-z.

July 10, 2025 9 of 9

**Reply to the Reviewer**

Re: Manuscript ID Preprint egusphere-2025-262

"Review article: Rethinking Preparedness for Coastal Compound Flooding: Insights from a Systematic Review"

Dina Vanessa Gómez-Rave, Anna Scolobig, Manuel del Jesus Natural Hazards and Earth System Sciences - NHESS

**Response to Reviewer 2**

We thank the reviewer for the thoughtful and detailed feedback on our manuscript. Below we provide point-by-point responses addressing each comment.

**1. General Comments**

The preprint titled "Rethinking Preparedness for Coastal Compound Flooding (CF): Insights from a Systematic Review" provides an insightful examination of strategies for managing compound flooding (CF) risks based on a structured literature review. The authors address the need to consider the multiple aspects of compound flooding risk including solutions that combine technical, environmental, and social dimensions, as well as the critical role of governance and multi-stakeholder collaboration.

Strengths of the paper include illustrating the evolution of CF research—from hazard-specific technical approaches to more holistic frameworks, while offering a critical lens on the shortcomings of current governance structures and participatory strategies. However, clear definitions and use of flood risk and disaster management terms are lacking. As a result, the paper framing lacks clarity and accurate use of terms which are well defined in the scientific literature. In particular the use of the term "preparedness" seems to be applied to more than just the preparedness phase of the disaster management cycle but rather flood risk and adaptation more broadly. The definition and use of this term, which also appears in the title should be clear.

Thank you for this valuable observation. To clarify the scope of the term *preparedness*, the revised manuscript now explicitly states (Line 69):

"Preparedness plays a central role in this shift. As defined by the UNDRR, preparedness refers to the knowledge and capacities developed by institutions, communities, and individuals to anticipate, respond to, and recover from likely, imminent, or ongoing hazard events UNDRR, United Nations Office for Disaster Risk Reduction [2017]. It includes early warning systems, contingency planning, and the institutional arrangements required to support timely and coordinated action."

Expanding on this definition, the manuscript further explains how the term is understood in this study, adopting a broader perspective (Line 118):

"We adopt a broad understanding of preparedness that goes beyond its conventional role in the DRR cycle—typically associated with EWS, contingency planning, and emergency readiness. Instead, it is framed as a multidimensional process encompassing anticipatory governance, infrastructural and ecosystem-based measures, and behavioural strategies aimed at reducing vulnerability prior to the manifestation of hazardous conditions. This perspective aligns not only with emerging literature on integrated FM Bark et al. [2021], [Konami et al. [2021], [De Silva et al. [2022], [Sánchez-García et al. [2024], but also firmly grounded in Priority 4 of the SFDRR, which advocates for preparedness actions that include inclusive governance, resilient infrastructure, public education, psychosocial support, and the incorporation of risk reduction into development planning and post-disaster reconstruction UNDRR, United Nations Office for Disaster Risk Reduction [2015]."

This definition supports the analytical focus of the study and ensures consistent use of the term throughout the manuscript.

Additionally, the integration of case studies based on the most relevant papers (e.g., China's Sponge City Program, the Netherlands' Delta Plan) adds depth to the analysis. However, the paper would be improved with a more explicit discussion of the limitations of the reviewed studies, particularly in terms of data availability and transferability. In addition, a more cohesive discussion section which distills and structures the findings for future research and practical applications would improve the impact of the paper.

Overall, this preprint makes a valuable contribution to the literature on disaster risk reduction and climate adaptation. With major revisions, it has the potential to contribute meaningfully to the scientific literature on compound flood risk management.

We have addressed this point by acknowledging the limitations of the reviewed case studies, particularly in terms of data availability and transferability. These aspects are now discussed in the revised manuscript, especially in the new section on Future Research. The final section was also reorganized to highlight key insights and reflect on their implications for research and practice.

**2. Specific Comments**

**Framing**

The flood risk and disaster management terms used are not defined and therefore the framing is unclear. For example, the stated focus is on preparedness, however, Blue & Green Infrastructure for example is more connected to adaptation or mitigation of hazards rather than preparedness.

I would encourage the authors to clearly define the risk equation they are using (hazard, exposure, vulnerability) and the disaster management cycle (preparedness, event/disaster, response, recovery, mitigation/adaptation) and cite relevant literature (for example Koks et al., 2015).

It seems that the intended focus is more risk reduction strategies across the disaster risk management cycle for compound floods in coastal areas. The conclusion does not mention the coastal context at all which is supposed to be the focus of this study. The findings should connect back to the focus area and provide an outlook related to that context.

We appreciate the reviewer's insights regarding conceptual framing and terminology. We agree that a clearer articulation of the underlying risk framework would strengthen the manuscript. In response, the revised version now incorporates a dedicated paragraph that defines the main components of risk and clarifies the distinction between drivers, hazards, exposure, and vulnerability. As included in the manuscript (Line 33):

"Risk is commonly conceptualised as the potential for adverse consequences for human or ecological systems resulting from the interaction between hazard, exposure, and vulnerability Intergovernmental Panel On Climate Change (Ipcc) [2023]. Within this framework, compound events are defined as the combination of climatic drivers and/or hazards that jointly contribute to societal or environmental risk Zscheischler et al. [2018]. Drivers encompass processes, variables, and phenomena in the climate and weather domain—such as precipitation, temperature, river flow, coastal water levels, atmospheric humidity, soil moisture or wind speed—that may operate across multiple spatial and temporal scales. Hazards, in contrast, denote the immediate physical phenomena—such as floods, heatwaves, or landslides—that may trigger impacts when they

July 10, 2025 2 of 8

coincide with exposure—the presence of people, infrastructure, or ecosystems in harm's way—and vulnerability—their propensity to suffer damage or loss due to limited capacity to anticipate, cope with, or recover from the event Koks et al. [2015], Zscheischler et al. [2020], Intergovernmental Panel On Climate Change (Ipcc) [2023]. The interplay among these components can result in compound risks, arising from single extremes or co-occurring events affecting critical systems or sectors Intergovernmental Panel On Climate Change (Ipcc) [2023]. This conceptual framing provides a basis for analysing how interacting climatic conditions can evolve into complex events—and how their consequences ripple through interconnected systems."

This addition helps keep the terminology consistent throughout the manuscript and places the review within a commonly used risk framework. It also contributes to linking preparedness with a broader understanding of CF risk in coastal areas.

**Methodology**

The use of ASReview and BERT model is innovative and the steps are clearly explained. It is mentioned that the ASReview model is based on their textual features to prevent author name and citation network biases. However, other biases can exist while using machine learning screening (e.g., keyword selection, training data). If these were addressed or at least identified this could be added.

Also, a clearer explanation of how subjective decisions were minimized would enhance reproducibility. The PRISMA flowchart (Figure 2) is clear but it would be helpful to add more detail on how the 49 articles were assessed to align with the research questions.

These aspects are now addressed in Section 3.2 and the limitations section. The screening process was structured to reduce subjectivity and enhance reproducibility by combining a fine-tuned BERT model for initial relevance scoring with active learning via ASReview. Predefined inclusion criteria guided the human-in-the-loop validation process, with iterative updates ensuring consistency in decision-making. To minimize bias, author names and citation data were excluded, and only titles and abstracts were used during the initial screening. Final inclusion decisions were based on full-text analysis using consistent criteria aligned with the research questions. Remaining sources of interpretative uncertainty—such as borderline cases and varying definitions of "compound"—are acknowledged as methodological limitations.

**Thematic Gaps**

While the paper acknowledges underrepresented themes like governance and behavioral dimensions, it stops short of proposing specific pathways for addressing them. The conclusion hints at the need for co-production and hybrid strategies but could be more explicit in offering guidance for implementation, especially in varied socio-political contexts.

The discussion on fragmented governance (Figure 8) and the challenges is valuable but could be strengthened by referencing mechanisms known in the literature to improve cross-sectoral coordination such as policy incentives or joint funding programs.

In addition, it may be helpful to look at the broader literature on several points. For example, it is mentioned that nature-based solutions are rather implemented in middle income countries but there are many projects that incorporate NbS in all income levels. For example, green dike and making room for the river projects in the Netherlands and Mangrove restoration in many countries globally. In addition, Indigenous Knowledge is integrated into preparedness and adaptation in high income countries (e.g. New Zealand, Australia, Canada). Perhaps rather than classification based on income, the approaches could be referenced (eg: NbS, Indigenous Knowledge) with some reference to regional strengths and challenges.

We revised the governance results section to address the reviewer's concerns by incorporating concrete coordination mechanisms—such as joint planning incentives, inter-agency funding schemes, and formal cooperation platforms—alongside a critical discussion of fragmentation and actor interactions (Lines 676 to

July 10, 2025 3 of 8

685). We also expanded the income-based framing with a regional analysis in Section 4.2 (from line 510 onward) that highlights institutional maturity, sociocultural dynamics, and environmental priorities, including examples of nature-based solutions and Indigenous Knowledge across diverse contexts.

**Figures and Visualizations**

Figures are generally helpful and relevant, however, the design of some visuals (e.g., Figures 6 and 7) are dense and would benefit from simplification or improved legends to enhance readability.

Figures 6 and 7 have been revised to improve readability. The visual structure has been simplified, font sizes increased where necessary, color contrasts adjusted (particularly for backgrounds and node/link elements), and legends clarified. These changes enhance accessibility while preserving the level of complexity needed to convey the information accurately.

**Integration of Social Dimensions**

The paper identifies a gap in social science research within the reviewed literature (Figure 3). It would be helpful to discuss why this gap exists and how it might impact the effectiveness of preparedness strategies. For example, are there biases in funding or publication trends that favor technical over social studies? Are there challenges with data collection or availability?

We have expanded the discussion around Figure 3 to briefly reflect on the limited presence of social science perspectives in the reviewed literature.

The added text reads as follows (Line 323):

"The observed asymmetry may reflect how research trajectories have developed over time, shaped by differing priorities as well as methodological, theoretical and disciplinary challenges. Historically, flood risk has been addressed through technical and hazard-centered frameworks, with a strong emphasis on hydrometeorological drivers, modelling, and structural measures, leaving less space for analysing how societies perceive, experience, and respond to flood events Lechowska [2022]. Socio-political dimensions are often treated as secondary, rather than central to how risks are understood and managed. Furthermore, inconsistent terminology and conceptual ambiguity, especially in definitions of multi-hazard and compound events, have contributed to the "fragmentation of the literature," generating redundancy and confusion that hinder interdisciplinary collaboration [Serinaldi et al. [2022]], Green et al. [2025]. Methodological constraints such as limited data availability, lack of standardization, and the context-dependence of social indicators also restrict their integration [Girons Lopez et al. [2017]], [Vanelli et al. [2022]]. Importantly, social and behavioural science research on these topics has been underfunded until the last decade. This undermined not only the theoretical but also the disciplinary development of risk perception, preparedness and communication studies. A more integrated approach is needed to inform preparedness strategies that reflect both the physical dynamics of CF and the ways in which societies experience and respond to them."

**Regional Disparities**

The analysis of high-, middle-, and low-income countries is useful but somewhat generalized. More nuanced comparisons (e.g., within middle-income countries) could reveal additional insights about contextual factors influencing preparedness.

We expanded Section 4.2 (from line 510 onward) to go beyond broad income categories, illustrating contextual differences within and across income groups through regionally specific examples of preparedness strategies.

July 10, 2025 4 of 8

**3. Technical Corrections**

| TC# | Line #                 | Comment and Response                                                                                                                                                                                                                                                                                                                                                                                                                                                                                                                                                                                                                                                                                                                                                                |
|-----|------------------------|-------------------------------------------------------------------------------------------------------------------------------------------------------------------------------------------------------------------------------------------------------------------------------------------------------------------------------------------------------------------------------------------------------------------------------------------------------------------------------------------------------------------------------------------------------------------------------------------------------------------------------------------------------------------------------------------------------------------------------------------------------------------------------------|
| 1   | Throughout             | Consider rephrasing long or complex sentences to improve readability, especially in the methods and discussion sections.  Long or complex sentences in the Methods and Discussion sections were revised to improve readability and flow.                                                                                                                                                                                                                                                                                                                                                                                                                                                                                                                                            |
| 2   | Throughout             | With the term "compound flooding" you sometimes abbreviate as "CF" and sometimes don't. This should be standardized throughout the paper.  The use of "compound flooding" and the abbreviation "CF" was made consistent throughout the manuscript.                                                                                                                                                                                                                                                                                                                                                                                                                                                                                                                                  |
| 3   | Line 8 to 22           | Abstract should mention the methods used and highlight key results. The abstract was updated to briefly describe the methods and highlight key findings, in line with the paper's focus (Line 8 to 21)                                                                                                                                                                                                                                                                                                                                                                                                                                                                                                                                                                              |
| 4   | Lines 55, 103, and 297 | Sendai Framework is introduced twice (Lines 55 and 103).  Mentions of the Sendai Framework were revised to avoid redundancy. It is now introduced only once, at Line 82, and referred to consistently thereafter.                                                                                                                                                                                                                                                                                                                                                                                                                                                                                                                                                                   |
| 5   | Lines 155–157          | You mention "storm surges, river flooding, and extreme rainfall" create heightened risk. These are all related to the hazard component of risk. If you only focus on hazard then this should be clearly stated. However, you later specify that you are looking at how strategies integrate technical, environmental, and social dimensions which suggests you look at drivers related to multiple components of risk. Be clear about how you define a use risk and hazard terminology. Key terms such as hazard, drivers, and risk were defined early in the manuscript (Lines 32 to 45). The scope of the analysis was clarified to reflect a focus on hazard-related drivers, while recognizing that preparedness strategies may touch on broader dimensions (Lines 118 to 125). |
| 6   | Line 194               | What is meant by "reflecting the diverse strategies employed to address flood risk and preparedness". Flood risk is something exists due to a combination of hazard, exposure, and vulnerability. Risk reduction measures can target each of these components. Actions for risk reduction can also be framed as targeting particular phases of the disaster management cycle including preparedness.  The sentence was rephrased to specify that the review focuses on strategies addressing hazard-related drivers during the preparedness phase of the disaster management cycle (Lines 225 and 226).                                                                                                                                                                             |
| 7   | Line 224               | Researcher-In-The-Loop (RITL) is mentioned in full twice with the abbreviation. Just include this once and then use the abbreviation. The term Researcher-In-The-Loop (RITL) is now written in full only once (Line 266), with the abbreviation used consistently throughout the rest of the manuscript.                                                                                                                                                                                                                                                                                                                                                                                                                                                                            |

(Continued on next page)

July 10, 2025 5 of 8

| TC# | Line #           | Comment and Response                                                                                                                                                                                                                                                                                                                                                                                                                                                                     |
|-----|------------------|------------------------------------------------------------------------------------------------------------------------------------------------------------------------------------------------------------------------------------------------------------------------------------------------------------------------------------------------------------------------------------------------------------------------------------------------------------------------------------------|
| 8   | Lines 273–276    | The two sentences starting with "In parallel, it is important to acknowledge" are a bit awkward. Consider rephrasing.  The entire paragraph was revised to improve phrasing and remove repetition.  The updated version appears in Lines 349 to 355.                                                                                                                                                                                                                                     |
| 9   | Table 2          | Clarify that the years listed are publication years, and ensure consistent formatting across entries.                                                                                                                                                                                                                                                                                                                                                                                    |
|     |                  | Possible double entry error for year with (Chan et al., 2023).                                                                                                                                                                                                                                                                                                                                                                                                                           |
|     |                  | The years and references with years are also somewhat redundant.                                                                                                                                                                                                                                                                                                                                                                                                                         |
|     |                  | Consider reformatting and perhaps only include the reference.                                                                                                                                                                                                                                                                                                                                                                                                                            |
|     |                  | Table 2 was adjusted. Formatting inconsistencies were corrected and the duplicate entry for Chan et al. (2023) was removed.                                                                                                                                                                                                                                                                                                                                                              |
| 10  | Line 307         | Reference to the literature would fit here at the end of the sentence. A reference was added at the end of the sentence, as suggested (Line 359).                                                                                                                                                                                                                                                                                                                                        |
| 11  | Lines 246–247    | "this nuanced aspect of preparedness" It is unclear what this refers to.  Revised to clarify that the phrase refers to preparedness for simultaneous or interacting flood drivers, and to explain why this topic is only recently gaining attention (Lines 286 to 294).                                                                                                                                                                                                                  |
| 12  | Line 544         | You mention "cognitive bias" here for the first time in the conclusions. While cognitive simplification is mentioned earlier in the article with regards to CF "cognitive bias" is not clearly addressed in the article. Be clear about what you mean in the conclusions and/or reference how you use the term earlier in the article. Revised to use "cognitive simplification" instead of "cognitive bias," ensuring terminological consistency with earlier sections (Lines 508, 706) |
| 15  | Line 422         | Typo with extra period. The extra period was removed.                                                                                                                                                                                                                                                                                                                                                                                                                                    |
| 16  | Lines 281 to 284 | Provide reference and quantification of increase in publications on natural hazard research.  Selected references and brief quantification were added to support the observed increase in hazard-related publications (Lines 336 to 344).                                                                                                                                                                                                                                                |

**References**

Rosalind H. Bark, Julia Martin-Ortega, and Kerry A. Waylen. Stakeholders' views on natural flood management: Implications for the nature-based solutions paradigm shift? *Environmental Science & Policy*, 115:91-98, January 2021. ISSN 14629011. doi: 10.1016/j.envsci.2020.10.018. URL <a href="https://linkinghub.elsevier.com/retrieve/pii/S1462901120313678">https://linkinghub.elsevier.com/retrieve/pii/S1462901120313678</a>.

Asitha De Silva, Dilanthi Amaratunga, and Richard Haigh. Green and Blue Infrastructure as Nature-Based Better Preparedness Solutions for Disaster Risk Reduction: Key Policy Aspects. Sustainability, 14(23): 16155, December 2022. ISSN 2071-1050. doi: 10.3390/su142316155. URL https://www.mdpi.com/2071-1050/14/23/16155.

M. Girons Lopez, G. Di Baldassarre, and J. Seibert. Impact of social preparedness on flood early warning sys-

- tems. Water Resources Research, 53(1):522-534, January 2017. ISSN 0043-1397, 1944-7973. doi: 10.1002/2016WR019387. URL https://agupubs.onlinelibrary.wiley.com/doi/10.1002/2016WR019387.
- Joshua Green, Ivan D. Haigh, Niall Quinn, Jeff Neal, Thomas Wahl, Melissa Wood, Dirk Eilander, Marleen De Ruiter, Philip Ward, and Paula Camus. Review article: A comprehensive review of compound flooding literature with a focus on coastal and estuarine regions. Natural Hazards and Earth System Sciences, 25 (2):747–816, February 2025. ISSN 1684-9981. doi: 10.5194/nhess-25-747-2025. URL https://nhess.copernicus.org/articles/25/747/2025/.
- Intergovernmental Panel On Climate Change (Ipcc). Climate Change 2022 Impacts, Adaptation and Vulnerability: Working Group II Contribution to the Sixth Assessment Report of the Intergovernmental Panel on Climate Change. Cambridge University Press, 1 edition, June 2023. ISBN 978-1-009-32584-4. doi: 10.1017/9781009325844. URL https://www.cambridge.org/core/product/identifier/9781009325844/type/book.
- E.E. Koks, B. Jongman, T.G. Husby, and W.J.W. Botzen. Combining hazard, exposure and social vulnerability to provide lessons for flood risk management. *Environmental Science & Policy*, 47:42-52, March 2015. ISSN 14629011. doi: 10.1016/j.envsci.2014.10.013. URL https://linkinghub.elsevier.com/retrieve/pii/S1462901114002056.
- Takahiro Konami, Hirohisa Koga, and Akihiko Kawatsura. Role of pre-disaster discussions on preparedness on consensus-making of integrated flood management (IFM) after a flood disaster, based on a case in the Abukuma River Basin, Fukushima, Japan. *International Journal of Disaster Risk Reduction*, 53:102012, February 2021. ISSN 22124209. doi: 10.1016/j.ijdrr.2020.102012. URL https://linkinghub.elsevier.com/retrieve/pii/S2212420920315144.
- Ewa Lechowska. Approaches in research on flood risk perception and their importance in flood risk management: a review. *Natural Hazards*, 111(3):2343–2378, April 2022. ISSN 0921-030X, 1573-0840. doi: 10.1007/s11069-021-05140-7. URL https://link.springer.com/10.1007/s11069-021-05140-7.
- Francesco Serinaldi, Federico Lombardo, and Chris G. Kilsby. Testing tests before testing data: an untold tale of compound events and binary dependence. Stochastic Environmental Research and Risk Assessment, 36(5):1373–1395, May 2022. ISSN 1436-3240, 1436-3259. doi: 10.1007/s00477-022-02190-6. URL https://link.springer.com/10.1007/s00477-022-02190-6.
- Carlos Sánchez-García, Óscar Corvacho-Ganahín, Albert Santasusagna Riu, and Marcos Francos. Nature-Based Solutions (NbSs) to Improve Flood Preparedness in Barcelona Metropolitan Area (Northeastern Spain). *Hydrology*, 11(12):213, December 2024. ISSN 2306-5338. doi: 10.3390/hydrology11120213. URL https://www.mdpi.com/2306-5338/11/12/213.
- UNDRR, United Nations Office for Disaster Risk Reduction. Sendai Framework for Disaster Risk Reduction 2015-2030. Technical report, 2015. URL <a href="https://www.undrr.org/publication/sendai-framework-disaster-risk-reduction-2015-2030">https://www.undrr.org/publication/sendai-framework-disaster-risk-reduction-2015-2030</a>.
- UNDRR, United Nations Office for Disaster Risk Reduction. The Sendai Framework Terminology on Disaster Risk Reduction. Technical report, 2017. URL https://www.undrr.org/terminology/preparedness.
- Franciele Maria Vanelli, Masato Kobiyama, and Mariana Madruga De Brito. To which extent are sociohydrology studies truly integrative? The case of natural hazards and disaster research. *Hydrology and Earth System Sciences*, 26(8):2301–2317, May 2022. ISSN 1607-7938. doi: 10.5194/hess-26-2301-2022. URL https://hess.copernicus.org/articles/26/2301/2022/.
- Jakob Zscheischler, Seth Westra, Bart J. J. M. Van Den Hurk, Sonia I. Seneviratne, Philip J. Ward, Andy Pitman, Amir AghaKouchak, David N. Bresch, Michael Leonard, Thomas Wahl, and Xuebin Zhang. Future climate risk from compound events. *Nature Climate Change*, 8(6):469–477, June 2018. ISSN 1758-678X, 1758-6798. doi: 10.1038/s41558-018-0156-3. URL https://www.nature.com/articles/s41558-018-0156-3.

July 10, 2025 7 of 8

Jakob Zscheischler, Olivia Martius, Seth Westra, Emanuele Bevacqua, Colin Raymond, Radley M. Horton, Bart Van Den Hurk, Amir AghaKouchak, Aglaé Jézéquel, Miguel D. Mahecha, Douglas Maraun, Alexandre M. Ramos, Nina N. Ridder, Wim Thiery, and Edoardo Vignotto. A typology of compound weather and climate events. Nature Reviews Earth & Environment, 1(7):333–347, June 2020. ISSN 2662-138X. doi: 10.1038/s43017-020-0060-z. URL https://www.nature.com/articles/s43017-020-0060-z.

---

## Referee Report (RR1)

**Second Review of egusphere-2025-262**

Recommendation: Minor revision

I appreciate the efforts deployed by the authors to improve the paper. The changes include the addition of hazard and hazard driver definitions, the number of reviewed papers, as well as clarifications on the aim, preparedness, etc. Moreover, almost all figures were edited and greatly improved in terms of expression capacity and readability. All of these additions or clarifications enhance the quality of this study.

However, there are still some issues to be addressed (the lines refer to the clean manuscript, not the one with track changes):

- Line 9: river discharge is not a hazard driver (river flow is, not the discharge itself), and storm surges are hazards themselves. Please rephrase.
- Line 27: I do not see the point of adding this (—often exceeding existing response capacities (Simpson et al., 2023) to the end of the phrase. It creates confusion.
- Lines 49-53 (The interplay among these components can result in compound risks, arising from single extremes or co-occurring events affecting critical systems or sectors (IPCC, 2023): the interplay of the stated elements makes up the risk, not the compounded risk. Compounded risks arise from multiple hazards (co-occurrent or sequential). Please read IPCC (2023) with greater attention and correct.
- Lines 90-95: The idea attributed to Armaş et al is not actually correct. Our study aims to analyse systemic vulnerability (not risk as indicated by the authors of the reviewed manuscript) using a new Systemic Vulnerability Model. The model relies on the Enhanced Impact Chains (EIC) introduced in Albulescu and Armaş (2024), so that the vulnerability dynamics tracked using the EICs are used as a key element of capturing systemic vulnerability. In short, I advise the author to cite both sources and modify the paragraph to really convey the results of the cited papers. Please see a suggestion below:

Systemic vulnerability refers to the susceptibility of interdependent systems—such as infrastructure networks, governance structures, or social services—to suffer disruption under external stress, due to the cascading effects that arise from their internal linkages (Weir et al., 2024). A recently proposed definition of systemic vulnerability is that related to the persistent core of vulnerability that endures over time despite mitigation efforts, societal and technological progress, leading to reinforced impacts (Armaş et al., 2025). This core can be depicted only by studying vulnerability dynamics across space and time, using new operational tools that can trace this dynamics (Enhanced Imapct Chains, as proposed by Albulescu and Armaş, 2024). Systemic risk, in turn, captures the potential for these disruptions to propagate across sectors and scales [find proper citation here], resulting in widespread and often unforeseen consequences (Armaş et al., 2025). This can further exacerbate systemic vulnerability as a persistent condition that can amplify future impacts or obstruct adaptive responses, even in the presence of mitigation efforts. Such a perspective situates compound risk within the broader dynamics of interdependence, where systemic conditions shape not only the onset of these impacts but their amplification and persistence.

If the authors do not wish to address these issues on the topic of systemic vulnerability, I kindly ask them to not refer to such concepts at all.

- I appreciate the detailed answer to this question in my first review report (What were the relevant and irrelevant records that served as the foundation for training the first machine learning model?). Your approach is indeed robust, but the manuscript's text does not leave the reader with this

impression. Please add more details from this answer into the methodology section in order to ensure clarity.

- Please check Table 2 and delete the extra commas .
- Line 703: delete the extra ) .
- I recommend including the Limitations as a subsection of the Discussion.

Finally, I commend the authors on their work. If the editor considers that I should review the implementation of the minor revision, I am happy to do so.

---

## Referee Report (RR2)

Journal Paper Reviewed: Gomez Rave, D. V., Scolobig, A., and del Jesus, M.: Review article: Rethinking Preparedness for Coastal Compound Flooding: Insights from a Systematic Review, EGUsphere [preprint], https://doi.org/10.5194/egusphere-2025-262, 2025.

Journal: Natural Hazards and Earth System Sciences (NHESS)

Second Round Review - 25.07.25

**1. General Comments**

The edited preprint titled "Rethinking Preparedness for Coastal Compound Flooding (CF): Insights from a Systematic Review" has been substantially improved based on the reviews and thanks to the effort of the authors. The edits made to date are appreciated and several additional comments are provided below.

The definitions and applications of the risk equation and the disaster management cycle are now more clearly articulated, with appropriate references, and the background has been significantly strengthened. While the paper provides a clear explanation for its use of the term "preparedness," this usage diverges from much of the existing literature. In this preprint, "preparedness strategies" is employed as an umbrella term encompassing both preparedness and adaptation strategies. This is primarily a matter of differing time scales: preparedness typically addresses response and recovery activities tied to specific hazard events, whereas adaptation refers to long-term strategies not linked to particular events. To avoid confusion, the authors could either use the more conventional phrasing "preparedness and adaptation strategies," or explicitly define their broadened use of "preparedness strategies," including clear inclusion and exclusion criteria.

This preprint on compound coastal flood risk makes a valuable contribution to the literature on disaster risk reduction and climate adaptation. With minor revisions it would be suitable for publication.

**2. Specific Comments**

**Coastal Focus**

It is stated that the study focuses on Coastal Compound Floods specifically. Sometimes "coastal CF" is used and sometimes just "CF" is used. It should be clear that the statements do not apply to all compound floods broadly, but rather that you superficially address coastal compound floods. Consider using the abbreviation coastal compound flood (CCF), which is used elsewhere in the scientific literature.

The term "coastal" could also be mentioned in the research questions. As they are currently stated the research questions could cover all compound flood types.

**Regional Analysis**

The integration of case studies is now well structured by timescale and region. The outline of the three phases helps to frame the shifts that have occurred in the field along with limitations. Some comments on specific regions are found below:

Europe: You could comment on the fact that in Europe there is a baseline for hazard mapping and use of certain technical tools with the EU Flood Directive.

Asia: You address the entire region of Asia but only mention examples from China. Your study list in Table 2 includes other countries such as Vietnam and Indonesia. They could also be mentioned here.

North America: Only the USA is covered here but in Table 2 you also have a study from Canada. Some relevant aspects of the Canadian context could be covered here.

**Methodology**

The methods are now very clearly explained including limitations. As mentioned, the fact that this method uses titles and abstracts is a key limitation and as a result, there are likely many relevant papers not included here. The paper would be strengthened if the authors could comment on what would be required to apply similar tools to the entire text of the scientific publications. It would be interesting to know what the key barriers would be (eg. computation time, less transparency etc.).

**3. Technical Corrections**

| TC# | Line #                | Comment                                                                                                                                                                                                                                                                                                                                             |
|-----|-----------------------|-----------------------------------------------------------------------------------------------------------------------------------------------------------------------------------------------------------------------------------------------------------------------------------------------------------------------------------------------------|
| 1   | Abstract (Lines 9–20) | "helps strengthening the link" Should be "helps strengthen the link"                                                                                                                                                                                                                                                                                |
| 2   | 65                    | Phrasing awkward: "In particular, FRM practices under occurrence of concurrent drivers must address the limitations of traditional single-hazard assumptions"                                                                                                                                                                                       |
| 3   | 263                   | You mention researcher and other information, is this the author of the article or the researcher using this meta-analysis method?                                                                                                                                                                                                                  |
| 4   | Table 2 (Line ~422)   | Netherland is used, but the country is "the Netherlands"                                                                                                                                                                                                                                                                                            |
|     |                       | In the caption: "research focused on CF and orange circles indicating those centred on coastal flooding preparedness"                                                                                                                                                                                                                               |
| 5   | Figure 6 (Line ~430)  | "Centred" uses UK spelling. Elsewhere you have "centered." Pick one spelling standard: UK or US English — NHESS typically accepts UK but consistency is key.                                                                                                                                                                                        |
|     |                       | It seems like Figure 6 is missing a legend for the map. Either the symbols on the map should correspond to the rest of the figure or a legend is needed.                                                                                                                                                                                            |
| 6   | 465                   | "preparedness campaigns mainly aimed at addressing conflicts (e.g. with NGOs or other organisations questioning ecological and environmental impacts of the programme)"                                                                                                                                                                             |
|     |                       | "Programme" vs "Program" — standardize spelling to UK or US.                                                                                                                                                                                                                                                                                        |
| 7   | 663 - Figure 8        | The style of this figure makes it hard to read with the "swirl" shape. For clarity, I would recommend a more straight branch diagram without the swirl effect.                                                                                                                                                                                      |
| 8   | 780                   | Lack of logical flow: "A vast majority of the analysed studies does not incorporate behavioural insights into preparedness frameworks. This omission is critical: if individuals—and institutions— simplify risk without including compound dynamics, then communication, EWS, and planning efforts must be adapted to counteract such tendencies." |
|     |                       | This section first suggests that behaviour needs to be considered but then mentioned the need to include compound dynamics. Clearly state the argument with a logical flow.                                                                                                                                                                         |

| 9  | 796      | Repetition: When local perspectives are sidelined, transformative change becomes unlikely         |
|----|----------|---------------------------------------------------------------------------------------------------|
| 10 | Multiple | Both the American spelling "modeling" and British spelling "modelling" is used in several places. |

---

## Author Response (AR2)

**Reply to the Reviewer**

Re: Manuscript ID Preprint egusphere-2025-262

"Review article: Rethinking Preparedness for Coastal Compound Flooding: Insights from a Systematic Review"

Dina Vanessa Gómez-Rave, Anna Scolobig, Manuel del Jesus Natural Hazards and Earth System Sciences - NHESS

**Response to Reviewer 1**

We thank Professor Cosmina Albulescu for her thoughtful and constructive second review. We have carefully considered all comments and revised the manuscript accordingly. The observations were particularly helpful in refining key concepts and improving the overall clarity and coherence of the text. Below, we provide a detailed point-by-point response.

**Review Report**

I appreciate the efforts deployed by the authors to improve the paper. The changes include the addition of hazard and hazard driver definitions, the number of reviewed papers, as well as clarifications on the aim, preparedness, etc. Moreover, almost all figures were edited and greatly improved in terms of expression capacity and readability. All of these additions or clarifications enhance the quality of this study.

However, there are still some issues to be addressed (the lines refer to the clean manuscript, not the one with track changes):

• Line 9: river discharge is not a hazard driver (river flow is, not the discharge itself), and storm surges are hazards themselves. Please rephrase.

We respectfully note that both storm surge and river discharge are widely recognized in the literature as key drivers of compound coastal flooding. Studies—including Bevacqua et al. [2020], Zscheischler et al. [2020], Latif and Simonovic [2023], and Green et al. [2025]—consistently describe compound flood events as resulting from the interaction of multiple source variables, including these two. While we acknowledge the reviewer's point regarding language use, our approach aligns with prevailing conventions in this research domain.

In particular, we follow the framework proposed by Zscheischler et al. [2020], which provides a well-established conceptual basis for compound event classification. This framework is widely adopted in the compound flooding literature. Given this broad and accepted usage, we suggest retaining the terms *river discharge* and *storm surge* as drivers in the revised manuscript. Nonetheless, we remain open to clarifying this choice—via a footnote or a definition box—should the reviewer or editors consider it necessary to prevent potential misunderstandings.

• Line 27: I do not see the point of adding this ("—often exceeding existing response capacities (Simpson et al., 2023)") to the end of the phrase. It creates confusion.

To improve clarity, we have revised the sentence so that the reference to Simpson et al. [2023] is more seamlessly integrated into the overall meaning. The updated version now reads (Lines 26-28):

"The greatest risks from a changing climate may not arise from single hazards, but from the interaction of multiple climatic drivers and/or hazards combined with diverse forms of exposure, intersectional socioeconomic and geopolitical vulnerabilities—often challenging the capacity of institutions and communities to respond effectively (Simpson et al., 2023)."

August 5, 2025 1 of 3

• Lines 49–53: ("The interplay among these components can result in compound risks, arising from single extremes or co-occurring events affecting critical systems or sectors (IPCC, 2023)"): the interplay of the stated elements makes up the risk, not the compounded risk. Compounded risks arise from multiple hazards (co-occurrent or sequential). Please read IPCC (2023) with greater attention and correct.

We acknowledge the comment and have carefully revisited the relevant definitions provided in IPCC (2023). The report defines compound risks as those "arising from the interaction of hazards, which may be characterised by single extreme events or multiple coincident or sequential events that interact with exposed systems or sectors." It further clarifies that compound events refer to the combination of multiple drivers and/or hazards.

Accordingly, in the revised manuscript, we now state (Lines 43-46):

The interplay of multiple drivers and/or hazards can lead to compound events—such as co-occurring extremes—and their intersection with exposed and vulnerable systems may result in compound risk (IPCC, 2023). This conceptual framing provides a basis for analysing how interacting climatic conditions can evolve into complex events—and how their consequences ripple through interconnected systems.

• Lines 90–95: The idea attributed to Armaş et al. is not actually correct. Our study aims to analyse systemic vulnerability (not risk as indicated by the authors of the reviewed manuscript) using a new Systemic Vulnerability Model. The model relies on the Enhanced Impact Chains (EIC) introduced in Albulescu and Armaş (2024), so that the vulnerability dynamics tracked using the EICs are used as a key element of capturing systemic vulnerability. In short, I advise the author to cite both sources and modify the paragraph to really convey the results of the cited papers. Please see a suggestion below:

Systemic vulnerability refers to the susceptibility of interdependent systems—such as infrastructure networks, governance structures, or social services—to suffer disruption under external stress, due to the cascading effects that arise from their internal linkages (Weir et al., 2024). A recently proposed definition of systemic vulnerability is that related to the persistent core of vulnerability that endures over time despite mitigation efforts, societal and technological progress, leading to reinforced impacts (Armaş et al., 2025). This core can be depicted only by studying vulnerability dynamics across space and time, using new operational tools that can trace this dynamics (Enhanced Impact Chains, as proposed by Albulescu and Armaş, 2024). Systemic risk, in turn, captures the potential for these disruptions to propagate across sectors and scales [find proper citation here], resulting in widespread and often unforeseen consequences (Armaş et al., 2025). This can further exacerbate systemic vulnerability as a persistent condition that can amplify future impacts or obstruct adaptive responses, even in the presence of mitigation efforts. Such a perspective situates compound risk within the broader dynamics of interdependence, where systemic conditions shape not only the onset of these impacts but their amplification and persistence.

If the authors do not wish to address these issues on the topic of systemic vulnerability, I kindly ask them to not refer to such concepts at all.

To address this point, the paragraph has been reworded in line with the reviewer's suggestion, incorporating the cited works to better reflect the intended conceptual distinctions (Lines 47-56).

• I appreciate the detailed answer to this question in my first review report ("What were the relevant and irrelevant records that served as the foundation for training the first machine learning model?"). Your approach is indeed robust, but the manuscript's text does not leave the reader with this impression. Please add more details from this answer into the methodology section in order to ensure clarity.

Section 3.2 Article screening and data analysis using Active Learning Process was revised and rewritten to incorporate these details, including the number of labeled abstracts, the use of random sampling, and the exclusion of titles to minimize bias (Lines 268-299).

August 5, 2025 2 of 3

• Please check Table 2 and delete the extra commas.

The extra commas in Table 2 have been removed.

• Line 703: delete the extra closing parenthesis.

The extra parenthesis has been corrected.

• I recommend including the Limitations as a subsection of the Discussion.

Thank you for the suggestion. We have incorporated the limitations into a dedicated subsection (5.3) under the Discussion section.

Finally, I commend the authors on their work. If the editor considers that I should review the implementation of the minor revision, I am happy to do so.

We appreciate the reviewer's engagement and constructive input throughout the review process. We would of course welcome any further feedback.

**References**

- E. Bevacqua, M. I. Vousdoukas, T. G. Shepherd, and M. Vrac. Brief communication: The role of using precipitation or river discharge data when assessing global coastal compound flooding. *Natural Hazards and Earth System Sciences*, 20(6):1765–1782, 2020. doi: 10.5194/nhess-20-1765-2020. URL https://nhess.copernicus.org/articles/20/1765/2020/.
- Joshua Green, Ivan D. Haigh, Niall Quinn, Jeff Neal, Thomas Wahl, Melissa Wood, Dirk Eilander, Marleen De Ruiter, Philip Ward, and Paula Camus. Review article: A comprehensive review of compound flooding literature with a focus on coastal and estuarine regions. Natural Hazards and Earth System Sciences, 25 (2):747–816, February 2025. ISSN 1684-9981. doi: 10.5194/nhess-25-747-2025. URL https://nhess.copernicus.org/articles/25/747/2025/.
- Shahid Latif and Slobodan P Simonovic. Compounding joint impact of rainfall, storm surge and river discharge on coastal flood risk: an approach based on 3d fully nested archimedean copulas. *Environmental Earth Sciences*, 82(2):63, 2023.
- Nicholas P. Simpson, Portia Adade Williams, Katharine J. Mach, Lea Berrang-Ford, Robbert Biesbroek, Marjolijn Haasnoot, Alcade C. Segnon, Donovan Campbell, Justice Issah Musah-Surugu, Elphin Tom Joe, Abraham Marshall Nunbogu, Salma Sabour, Andreas L.S. Meyer, Talbot M. Andrews, Chandni Singh, A.R. Siders, Judy Lawrence, Maarten Van Aalst, and Christopher H. Trisos. Adaptation to compound climate risks: A systematic global stocktake. *iScience*, 26(2):105926, February 2023. ISSN 25890042. doi: 10. 1016/j.isci.2023.105926. URL https://linkinghub.elsevier.com/retrieve/pii/S2589004223000032
- Jakob Zscheischler, Olivia Martius, Seth Westra, Emanuele Bevacqua, Colin Raymond, Radley M. Horton, Bart Van Den Hurk, Amir AghaKouchak, Aglaé Jézéquel, Miguel D. Mahecha, Douglas Maraun, Alexandre M. Ramos, Nina N. Ridder, Wim Thiery, and Edoardo Vignotto. A typology of compound weather and climate events. Nature Reviews Earth & Environment, 1(7):333–347, June 2020. ISSN 2662-138X. doi: 10.1038/s43017-020-0060-z. URL https://www.nature.com/articles/s43017-020-0060-z.

August 5, 2025 3 of 3

**Reply to the Reviewer**

Re: Manuscript ID Preprint egusphere-2025-262
"Review article: Rethinking Preparedness for Coastal Compound Flooding: Insights from a Systematic Review"

Dina Vanessa Gómez-Rave, Anna Scolobig, Manuel del Jesus Natural Hazards and Earth System Sciences - NHESS

**Response to Reviewer 2**

Reviewer 2 is acknowledged for the constructive and precise feedback, which has been carefully considered and has contributed to improving the manuscript. The following responses outline how each point has been addressed.

**1 General Comments**

The edited preprint titled "Rethinking Preparedness for Coastal Compound Flooding (CF): Insights from a Systematic Review" has been substantially improved based on the reviews and thanks to the effort of the authors. The edits made to date are appreciated and several additional comments are provided below.

The definitions and applications of the risk equation and the disaster management cycle are now more clearly articulated, with appropriate references, and the background has been significantly strengthened. While the paper provides a clear explanation for its use of the term "preparedness," this usage diverges from much of the existing literature. In this preprint, "preparedness strategies" is employed as an umbrella term encompassing both preparedness and adaptation strategies. This is primarily a matter of differing time scales: preparedness typically addresses response and recovery activities tied to specific hazard events, whereas adaptation refers to long-term strategies not linked to particular events. To avoid confusion, the authors could either use the more conventional phrasing "preparedness and adaptation strategies," or explicitly define their broadened use of "preparedness strategies," including clear inclusion and exclusion criteria.

This preprint on compound coastal flood risk makes a valuable contribution to the literature on disaster risk reduction and climate adaptation. With minor revisions it would be suitable for publication.

This is a relevant and appreciated remark, which we have addressed as follows. We clarified in the introduction and methods that our use of "preparedness strategies" is intentionally broad, encompassing both preparedness and adaptation actions relevant to CCF risk.

The Introduction now includes the following sentence to define the conceptual framing (Lines 124-128):

"We adopt a broad understanding of preparedness that goes beyond its conventional role in the DRR cycle—typically associated with EWS, contingency planning, and emergency readiness. Instead, it is framed as a multidimensional process encompassing anticipatory governance, infrastructural and ecosystem-based measures, and behavioural strategies aimed at reducing vulnerability prior to the manifestation of hazardous conditions. In this review, "preparedness strategies" are used in a broad sense to include both conventional preparedness activities (e.g., early warning systems, response planning) and longer-term adaptation measures (e.g., infrastructure upgrades, community capacity building). This expanded usage reflects the growing need for integrated and scalable responses to CCF risks, where the distinction between short-term and long-term interventions is often blurred in practice. This perspective aligns not only with emerging literature on integrated FM (Bark et al., 2021; Konami et al., 2021; De Silva et al., 2022; Sánchez-García et al., 2024),

August 5, 2025 1 of 5

but also firmly grounded in Priority 4 of the SFDRR, which advocates for preparedness actions that include inclusive governance, resilient infrastructure, public education, psychosocial support, and the incorporation of risk reduction into development planning and post-disaster reconstruction (UNDRR, 2015)".

In the Methodology, we also specify how this framing informed the inclusion criteria (Lines 238-242):

"The initial analysis of search results from the WoS database provided a broad perspective on flooding preparedness research, capturing diverse topics and approaches. A total of 874 articles met the defined criteria, addressing key themes such as disaster preparedness, resilience, and flood management across various environments, including coastal and estuarine regions. The use of the broader term "coastal flooding" was intended to capture studies published prior to the widespread adoption of the compound event framework. Consequently, the retrieved literature spans a wide range of disciplinary approaches and timeframes. Many of these contributions focus on the hazard dimension of flood risk, particularly through measures implemented during the preparedness phase of FRM. This broad scope reinforces the need to refine the analysis toward compound hazard configurations, ensuring coherence with the specific objectives of this review. In line with our broadened conceptualization of "preparedness strategies" as encompassing both short-term preparedness and long-term adaptation, we included studies that addressed either domain—provided they explicitly contributed to risk reduction in the context of CCF. This inclusive approach reflects the practical and temporal convergence between preparedness and adaptation, and guided the application of our inclusion and exclusion criteria."

**2 Specific Comments**

**2.1 Coastal Focus**

It is stated that the study focuses on Coastal Compound Floods specifically. Sometimes "coastal CF" is used and sometimes just "CF" is used. It should be clear that the statements do not apply to all compound floods broadly, but rather that you superficially address coastal compound floods. Consider using the abbreviation coastal compound flood (CCF), which is used elsewhere in the scientific literature.

The term "coastal" could also be mentioned in the research questions. As they are currently stated the research questions could cover all compound flood types.

The abbreviation "CCF" (Coastal Compound Flooding) is now used consistently throughout the manuscript, replacing the previous use of "CF" where appropriate. Additionally, this aspect has been explicitly incorporated into the research questions to better reflect the scope of the review.

**2.2 Regional Analysis**

The integration of case studies is now well structured by timescale and region. The outline of the three phases helps to frame the shifts that have occurred in the field along with limitations. Some comments on specific regions are found below:

- Europe: You could comment on the fact that in Europe there is a baseline for hazard mapping and use of certain technical tools with the EU Flood Directive.
- Asia: You address the entire region of Asia but only mention examples from China. Your study list in Table 2 includes other countries such as Vietnam and Indonesia. They could also be mentioned here.
- North America: Only the USA is covered here but in Table 2 you also have a study from Canada. Some relevant aspects of the Canadian context could be covered here.

The Asia section has been expanded to include Vietnam, Myanmar, Bangladesh, and Indonesia, reflecting the range of cases listed in Table 2. A reference to the EU Flood Directive has been added to the Europe

August 5, 2025 2 of 5

section to highlight the baseline for technical tools. The North America section now includes relevant aspects of the Canadian case, alongside the USA.

**2.3 Methodology**

The methods are now very clearly explained including limitations. As mentioned, the fact that this method uses titles and abstracts is a key limitation and as a result, there are likely many relevant papers not included here. The paper would be strengthened if the authors could comment on what would be required to apply similar tools to the entire text of the scientific publications. It would be interesting to know what the key barriers would be (e.g. computation time, less transparency etc.).

We thank the reviewer for highlighting this important limitation. Indeed, the reliance on titles and abstracts constrains the analytical depth and may exclude significant content embedded within the full texts. Our current approach was developed to balance analytical scope and computational feasibility; however, we fully agree that mining the full body of scientific publications could yield more comprehensive insights.

Recent developments in artificial intelligence — particularly the emergence of transformer-based Large Language Models (LLMs) — open new opportunities to address this gap. As reported in recent studies, LLMs have shown promise in supporting multiple stages of systematic review processes, including data extraction directly from full-text PDFs, not limited to titles or abstracts. In this regard, Hill et al. 2024 show how AI-powered tools can be effectively employed to extract targeted information from full-text documents with increasing efficiency and accuracy. These technologies can enhance granular extraction of methodological or result-related data (including figures and tables), the semantic understanding of nuanced text, enabling classification or synthesis that goes beyond keyword matching, and efficiency in systematic reviews by reducing manual screening workloads. Nonetheless, as Lieberum et al. 2025 emphasize, these applications remain largely exploratory, with variability in reproducibility and transparency, especially in complex tasks like bias assessment or full semantic interpretation. Moreover, concerns about hallucinations, prompt sensitivity, and lack of standard validation protocols persist. Scaling such approaches would require careful consideration of computational costs, access to full-text repositories, and the development of robust validation protocols to ensure interpretability and scientific rigor.

A specific note has been added in the Limitations section of the manuscript to explicitly acknowledge this potential future direction, along with the computational and methodological considerations required for safely expanding toward full-text mining (Lines 794-803):

"...Although this approach was designed to pursue methodological transparency and computational scalability, it inevitably limits the depth of the review. Recent advances in artificial intelligence—particularly in Natural Language Processing (NLP) and the development of transformer-based Large Language Models (LLMs)—have shown promise in enabling full-text mining and semantic extraction from scientific publications. These tools can enhance the identification of nuanced content and latent connections that may be overlooked when relying solely on metadata. For instance, (Hill et al., 2024) showed the potential of AI-powered tools to extract targeted methodological details from full texts, while (Lieberum et al., 2025) emphasized both the opportunities and the limitations of using LLMs in evidence synthesis, noting concerns related to reproducibility, hallucinations, and prompt sensitivity. Given these challenges, the decision to rely on abstracts and titles remains methodologically justified, though future applications of AI-supported full-text analysis may offer greater depth and coverage, provided robust validation frameworks are in place."

August 5, 2025 3 of 5

**3 Technical Corrections**

| TC# | Line #                | Comment                                                                                                                                                                                                                                                                                                                                                                                                                                                                                                                                                                                                                |
|-----|-----------------------|------------------------------------------------------------------------------------------------------------------------------------------------------------------------------------------------------------------------------------------------------------------------------------------------------------------------------------------------------------------------------------------------------------------------------------------------------------------------------------------------------------------------------------------------------------------------------------------------------------------------|
| 1   | Abstract (Lines 9–20) | "helps strengthening the link" Should be "helps strengthen the link"  The suggested correction has been made.                                                                                                                                                                                                                                                                                                                                                                                                                                                                                                          |
| 2   | 65                    | Phrasing awkward: "In particular, FRM practices under occurrence of concurrent drivers must address the limitations of traditional single-hazard assumptions."  The sentence has been revised to (Line 68): "In particular, FRM practices must address the limitations of traditional single-hazard assumptions when concurrent drivers occur."                                                                                                                                                                                                                                                                        |
| 3   | 263                   | You mention researcher and other information, is this the author of the article or the researcher using this meta-analysis method? We have updated the text to use the term 'user' instead of 'researcher' to make it clear that we are referring to the individual performing the meta-analysis, not the authors of the articles (Line 273).                                                                                                                                                                                                                                                                          |
| 4   | Table 2 (Line 422)    | "Netherland" is used, but the country is "the Netherlands." Corrected to "the Netherlands" in the table.                                                                                                                                                                                                                                                                                                                                                                                                                                                                                                               |
| 5   | Figure 6 (Line 430)   | In the caption: "research focused on CF and orange circles indicating those centred on coastal flooding preparedness" "Centred" uses UK spelling. Elsewhere you have "centered." Pick one spelling standard: UK or US English — NHESS typically accepts UK but consistency is key. It seems like Figure 6 is missing a legend for the map. Either the symbols on the map should correspond to the rest of the figure or a legend is needed. We have standardized the spelling to UK English ('centred') throughout the manuscript. We have also added a clear legend to the map in Figure 6(a) to address its symbols. |
| 6   | 465                   | "preparedness campaigns mainly aimed at addressing conflicts (e.g. with NGOs or other organisations questioning ecological and environmental impacts of the programme)"  "Programme" vs "Program" — standardize spelling to UK or US.  We have standardized the spelling throughout the manuscript, using UK English ('programme') for consistency.                                                                                                                                                                                                                                                                    |
| 7   | 663 – Figure 8        | The style of this figure makes it hard to read with the "swirl" shape. For clarity, I would recommend a more straight branch diagram without the swirl effect.  The figure has been updated to enhance readability.                                                                                                                                                                                                                                                                                                                                                                                                    |

(Continued on next page)

August 5, 2025 4 of 5

| TC# | Line #   | Comment                                                                                                                                                                                                                                                                                                                                                                                                                                                                                                                                                                                                                                                                                                                                                                                                                                                                                                                                                                                                                                                                                                                                                                                                 |
|-----|----------|---------------------------------------------------------------------------------------------------------------------------------------------------------------------------------------------------------------------------------------------------------------------------------------------------------------------------------------------------------------------------------------------------------------------------------------------------------------------------------------------------------------------------------------------------------------------------------------------------------------------------------------------------------------------------------------------------------------------------------------------------------------------------------------------------------------------------------------------------------------------------------------------------------------------------------------------------------------------------------------------------------------------------------------------------------------------------------------------------------------------------------------------------------------------------------------------------------|
| 8   | 780      | Lack of logical flow: "A vast majority of the analysed studies does not incorporate behavioural insights into preparedness frameworks. This omission is critical: if individuals—and institutions— simplify risk without including compound dynamics, then communication, EWS, and planning efforts must be adapted to counteract such tendencies." This section first suggests that behaviour needs to be considered but then mentioned the need to include compound dynamics. Clearly state the argument with a logical flow.  We have revised the paragraph to more explicitly connect the omission of behavioral insights with the simplification of risk and the neglect of compound dynamics. The new text now presents a more logical argument (Lines 826-829): "A vast majority of the analysed studies does not incorporate behavioural insights into preparedness frameworks. This is a critical omission because individuals—and institutions—tend to simplify complex risks, often failing to account for compound dynamics. Consequently, communication, EWS, and planning efforts must be adapted to counteract these tendencies and promote a more comprehensive understanding of risk." |
| 9   | 796      | Repetition: When local perspectives are sidelined, transformative change becomes unlikely. Repetition has been removed.                                                                                                                                                                                                                                                                                                                                                                                                                                                                                                                                                                                                                                                                                                                                                                                                                                                                                                                                                                                                                                                                                 |
| 10  | Multiple | Both the American spelling "modeling" and British spelling "modelling" is used in several places.  Spelling standardized throughout the manuscript.                                                                                                                                                                                                                                                                                                                                                                                                                                                                                                                                                                                                                                                                                                                                                                                                                                                                                                                                                                                                                                                     |

**References**

James Edward Hill, Catherine Harris, and Andrew Clegg. Methods for using bing's ai-powered search engine for data extraction for a systematic review. *Research synthesis methods*, 15(2):347–353, 2024.

Judith-Lisa Lieberum, Markus Toews, Maria-Inti Metzendorf, Felix Heilmeyer, Waldemar Siemens, Christian Haverkamp, Daniel Böhringer, Joerg J Meerpohl, and Angelika Eisele-Metzger. Large language models for conducting systematic reviews: on the rise, but not yet ready for use—a scoping review. *Journal of Clinical Epidemiology*, 181:111746, 2025.

UNDRR, United Nations Office for Disaster Risk Reduction. The Sendai Framework Terminology on Disaster Risk Reduction. Technical report, 2017. URL https://www.undrr.org/terminology/preparedness.

August 5, 2025 5 of 5